# Microbial fertilizers modulate tobacco growth and development through reshaping soil microbiome and metabolome

Xiaoyu Wang,[1] Mingming Sun,[2] Lei Tian,[3] Mingfeng Yang,[2] Qiang Gao,[2] LIli Wang,[3] Honghao Yan,[1] Long Yang,[1] Xin Hou,[1] Peng Liu,[1] Li Zhang[1]

**ABSTRACT**   To elucidate the mechanisms of microbial fertilizers in enhancing tobacco growth and quality, this greenhouse-based pot experiment conducted over 40 days post-transplanting employed integrated microbiomics and metabolomics approaches to conduct a comparative analysis among conventional chemical, organic, and microbial fertilizers. Plant agronomic traits were systematically assessed at 20, 30, and 40 days post-transplanting, while soil physicochemical parameters were analyzed at the experimental terminus (40 days). The findings underscored the remarkable potential of microbial fertilizers in augmenting soil's quick-release nutrient pool and bolstering soil enzymatic activity, surpassing both chemical and organic counterparts. The application of microbial fertilizers accelerated tobacco growth and development, and significantly elevated agronomic indices, including plant stature, stem girth, leaf extension, and the abundance of aromatic precursors, thereby facilitating a marked improvement in tobacco leaf quality. Furthermore, the microbial community composition underwent pronounced alterations subsequent to the application of microbial fertilizers, with the emergence of pivotal microorganisms such as *Rhodanobacter* and *Pseudolabrys* within the treatment group. These microorganisms emerged as vital players in nutrient cycling processes, fostered plant growth, and mitigated the incidence of plant diseases. Microbial fertilizers demonstrated a significantly superior capacity to stimulate metabolic vigor in tobacco plants compared to other treatments, concomitant with a substantial enrichment of several metabolites, such as 3-methylindole. These data collectively imply that microbial fertilizers represent a more efficacious means of ameliorating soil physicochemical attributes, thereby fostering superior tobacco growth, development, and quality enhancement.

**IMPORTANCE**   In recent years, there has been a surge in research examining the impacts of various fertilizers on the microbial composition of tobacco rhizosphere soils, and numerous studies have consecutively reported the growth-promoting mechanisms of diverse fertilizers on tobacco plants. However, despite these advancements, the existing body of literature remains inadequate in conclusively demonstrating the superiority of microbial fertilizers over traditional organic and inorganic fertilizers in tobacco cultivation. Consequently, our research aims to demonstrate the superiority of microbial fertilizers in enhancing plant growth by utilizing a comprehensive approach that integrates microbiomics and metabolomics techniques.

**KEYWORDS**   tobacco, multi-omics, metabolism, soil microorganism, soil properties, microbial fertilizers, aroma pre-cursors

Tobacco, as a significant economic crop, is extensively cultivated across the globe. The moderate application of chemical fertilizers has been demonstrated to promote the growth and development of tobacco, thus increasing the yield of tobacco leaves (1).

**Peer Reviewer** Zichao Mao, Yunnan Agricultural University, Kunming, Yunnan, China

Address correspondence to Li Zhang, zhanglili@sdau.edu.cn.

Xiaoyu Wang and Mingming Sun contributed equally to this article. Author order was determined by drawing straws.

The authors declare no conflict of interest.

Nonetheless, in recent times, tobacco farmers frequently over-apply chemical fertilizers in an attempt to maximize production, a practice that has not only given rise to profound soil health concerns but also adversely affected the normal growth and development of tobacco plants (2–4). Consequently, the adoption of environmentally friendly novel materials as alternatives to traditional chemical fertilizers is imperative for further augmenting tobacco leaf yield and improving its quality.

Soil microorganisms exhibit a profound correlation with soil quality and its physicochemical properties, which exert direct influences on crop yield and quality (5). A well-balanced soil microbial community structure facilitates the harmonious functioning of soil nutrient cycling, the decomposition of soil organic matter, the biodegradation of organic pollutants, and the enhancement of plant resilience to abiotic stress. These processes collectively contribute to the promotion of crop growth and development (6, 7). To fundamentally enhance soil nutrient status, microbial fertilization through the inoculation of exogenous microorganisms has emerged as a dual-functional strategy. Traditionally, the primary objective of this practice has been to enhance specific functional groups, such as introducing N2-fixing and P-solubilizing bacteria into the soil to improve plant uptake and utilization of N and P (8). Recent studies have demonstrated that these microbial inoculants can function as ecological engineers, reshaping the structure of soil microbial communities to establish a soil microbiome that promotes crop growth and development (9–11). These preparations are typically formulated on the basis of beneficial microorganisms ubiquitous in soil, namely plant growth-promoting bacteria (PGPB). PGPB establish unique symbiotic relationships with plant roots, enhancing soil nutrient availability through processes such as nitrogen fixation, phosphate solubilization, and potassium release. They secrete or induce the production and release of phytohormones, thereby directly stimulating plant growth (9, 12). For example, *Sphingomonas* and *Bacillus paralicheniformis* have been shown to enrich beneficial bacteria in the soil, thereby influencing the soil microbiome (13, 14). Additionally, microbial inoculants such as *Azotobacter*, *Enterobacter*, and *Rhizobium* can significantly enhance crop resistance to drought stress, demonstrating their direct benefits for plant growth and stress tolerance. Concurrently, select strains within these genera (e.g., *Bacillus*, *Pseudomonas*, *Acinetobacter*, *Serratia*, *Pantoea*, *Psychrobacter*, *Enterobacter*, and *Rahnella*) have been identified as significant PGPB, demonstrating potential as exogenous microorganisms for incorporation into microbial formulations (15).

Furthermore, microbial fertilizers have demonstrated favorable outcomes in practical applications. Specifically, plant growth-promoting rhizobacterial (PGPR) microbial fertilizers have altered the soil microbial composition, markedly enhanced the root vigor of tobacco plants, and contributed to a comprehensive enhancement of tobacco growth parameters (16). Compared to woody peat, microbial fertilizers significantly enhanced the diversity and composition of the tobacco rhizosphere soil microbiome, enriching beneficial microbial taxa such as plant growth-promoting bacteria and improving key soil functions. This included increased availability of potassium and higher organic matter content, which collectively supported a healthier soil microecology and promoted tobacco growth (17). Application of microbial fertilizers containing *Phanerochaete chrysosporium* and *Bacillus thuringiensis* adjusts soil nitrogen supply and leaf uptake, significantly reduces nicotine content of tobacco, and improves tobacco quality (18). Extending these findings, microbial fertilizers have also been successfully employed in cereal crop cultivation, where they promoted wheat and rice growth, optimized nutrient utilization efficiency, and improved overall soil health (19–21).

In recent years, there has been a surge in research examining the impacts of various fertilizers on the microbial composition of tobacco rhizosphere soils, and numerous studies have consecutively reported the growth-promoting mechanisms of diverse fertilizers on tobacco plants (22–24). However, despite these advancements, the existing body of literature remains inadequate in conclusively demonstrating the superiority of microbial fertilizers over traditional organic and inorganic fertilizers in tobacco

cultivation (25, 26). Consequently, the current investigation was focused on conducting a comparative analysis of tobacco plants treated with conventional chemical fertilizers, organic fertilizers, and microbial fertilizers, utilizing an integrative approach combining microbiomics and metabolomics techniques. Consequently, the present study aimed therefore (i) to demonstrate the significant advantages of microbial fertilizers over conventional fertilizers in enhancing both the yield and quality of tobacco, and (ii) to elucidate in detail the mechanisms by which microbial fertilizers influence tobacco quality through the modulation of metabolic activities.

## MATERIALS AND METHODS

### Experimental materials and water and fertilizer management

Tobacco plants used for potting experiments were placed in the tobacco greenhouse of the Dai Zong Campus of Shandong Agricultural University (36.17N, 117.16E).

Tobacco seeds (variety: Yunyan87) were surface-sterilized with 70% ethanol for 2 min, followed by 1% sodium hypochlorite solution for 5 min, and then rinsed three times with sterile distilled water. The sterilized seeds were placed on moist filter paper in Petri dishes and incubated in a growth chamber at 25 ± 1°C with a 16/8 h light/dark cycle. Germination was monitored daily, and seeds were considered germinated upon the emergence of the radicle. Germination rates were calculated after 7 days. Only batches with a germination rate of ≥90% were used for subsequent experiments to ensure uniformity and reliability of the seedlings. To prevent the occurrence of soil-borne diseases in continuous cropping, the potting soil was obtained from a large field that had not been planted with tobacco and was a sandy loam with the following basic physical and chemical properties: organic matter 2.15%, alkaline dissolved nitrogen 52.18 mg/kg, effective phosphorus 23.41 mg/kg, quick-acting potassium 121.98 mg/kg, pH 6.00.

When seedlings grow to three to five true leaves, select seedlings with consistent growth, transplant them into 24 cm × 25.5 cm pots with 5 kg of soil, make sure each pot has the same growing conditions, and keep the same management with the local farmland. The pot experiment comprised four treatments: (i) a non-fertilized control (CK); (ii) chemical fertilization (NPK) with inorganic fertilizers; (iii) organic fertilization (OF) combining equivalent NPK doses with organic fertilizer; and (iv) microbial fertilization (MF) pairing matched NPK inputs with microbial inoculants. The composition of fertilizers in each group is shown in Table 1. The conventional recommended field fertilizer rate was 900 kg/hm$^2$, and potting rates were converted based on the field rates. Ensuring that the same total N input was obtained for each treatment, an N:P:K (1:1.5:3) application ratio, i.e., the same as that used in local on-farm crop management, was applied between the control and treatment groups to ensure that any differences observed between these groups could not be attributed to these nutrients (27). Differences in application rates, potassium sulfate, and calcium superphosphate were used to balance the nutrient levels, and the detailed fertilizer rates are shown in Table 2.

Use 70% of the total fertilizer application as a base fertilizer and mix it well with the soil before transplanting, and use the remaining 30% as a follow-up fertilizer by sprinkling the granules on the soil surface 7 days after transplanting (after the seedling

**TABLE 1** Fertilizers used in the experiment

| Type of fertilizer | Formulations | Technical indicators |
| --- | --- | --- |
| Organic fertilizer | Granulated | The main raw materials are straw, mushroom dregs, wine lees, grain scraps. |
| Microbial fertilizer | Granulated | Effective number of live bacteria ≥ 1 billion/g, mainly containing *Bacillus subtilis*, *Bacillus licheniformis*. |
| Fertilizers | Granulated | N:P:K = 10:10:20 |
| Potassium sulfate | Granulated | N:P:K = 0:0:50 |
| Calcium superphosphate | Granulated | N:P:K = 0:12:0 |

TABLE 2 Fertilizer application rate (g/kg) for each treatment in a pot experiment

| Fertilizer type | CK | NPK | OF | MF |
|---|---|---|---|---|
| Organic fertilizer | 0 | 0 | 2 | 0 |
| Microbial fertilizer | 0 | 0 | 0 | 2 |
| Fertilizers | 0 | 2 | 2 | 2 |
| Potassium sulfate | 0 | 0.4 | 0.4 | 0.4 |
| Calcium superphosphate | 0 | 0.83 | 0.83 | 0.83 |

resting period). In the first 7 days after transplanting, 400 mL of water was evenly poured into each pot every day, during which the state of the seedlings was observed, and water was replenished promptly if wilting occurred; after 7 days, 400 mL of water was evenly poured into each pot every other day.

## Soil sample collection and analysis

Soil samples were collected 40 days after transplanting. Three representative tobacco plants of uniform growth were selected among 15. The plants were pulled out to remove the rhizosphere soil. Three samples of tobacco rhizosphere soil were collected from randomly selected plants in the field. The remaining rhizosphere soil was used to determine soil physicochemical properties and enzyme activity.

## Determination of soil chemical properties and enzyme activity

Following the treatment of the retrieved soil samples, the concentration of alkaline dissolved nitrogen within the soil matrix was quantitatively assessed employing the alkaline diffusion technique (28). To determine the availability of phosphorus in the soil, the sodium bicarbonate leaching procedure coupled with the molybdenum-antimony anti-staining method was utilized. Additionally, the potassium content of the soil was precisely measured through the application of the ammonium acetate leaching method, subsequently analyzed via flame photometry. The organic matter content of the soil was determined adopting the potassium dichromate dilution-heating methodology (29). Lastly, the soil's pH value was accurately measured utilizing the potentiometric method, providing a comprehensive characterization of the soil's physicochemical properties. For each treatment, three independent soil samples were collected from the rhizosphere of three representative tobacco plants, with each sample analyzed in three technical replicates. The pot test was conducted 40 days after transplanting the tobacco plants. For each treatment, rhizosphere soil samples were collected from three representative tobacco plants, with each sample analyzed independently. Each soil sample was subjected to three technical replicates for the measurement of soil urease, sucrase, and peroxidase activities. Soil urease activity was determined using a soil urease kit (Solarbio, BC0120), soil sucrase activity was measured using a soil sucrase kit (Solarbio, BC0240), and soil peroxidase activity was assessed using a soil peroxidase kit (Solarbio, BC0100).

## Determination of agronomic traits and aroma precursors in plants

At 20, 30, and 40 days after transplanting, three plants exhibiting similar growth vigor were individually selected for measurement of plant height, stem circumference, and the length and width of their mid-section leaves (count the fourth to sixth leaves from top to bottom).

The content of aroma precursors serves as one of the crucial criteria for assessing the quality of tobacco leaves (30). At 40 days after transplanting, the leaves of the middle part of three tobacco plants with uniform growth were selected and stored in self-sealing bags at low temperature in an ultra-low-temperature refrigerator at −80°C for the determination of the content of aroma precursors. Plasmalemma pigment content was determined by colorimetric extraction with 95% ethanol grinding. Soluble sugar content was determined by using the kit (Solarbio, BC0030) and starch content by using

the kit (Solarbio, BC0700). For the determination of polyphenolic compounds in tobacco, please refer to YC/T 202-2006 "Determination of Polyphenolic Compounds Chlorogenic Acid, Scopoletin and Rutin in Tobacco and Tobacco Products."

## Soil DNA extraction and high-throughput sequencing

### DNA extraction and quality assessment

Total genomic DNA was extracted from soil samples using the E.Z.N.A. Soil DNA Kit (Omega Bio-tek, Norcross, GA, USA) following the manufacturer's protocol. DNA integrity was verified via 1% agarose gel electrophoresis, while concentration and purity were quantified using a NanoDrop2000 spectrophotometer (Thermo Fisher Scientific, USA).

### PCR amplification, library preparation, and sequencing

The V3-V4 hypervariable region of the bacterial 16S rRNA gene was amplified using primer pairs 338F (5′-ACTCCTACGGGAGGCAGCAG-3′) and 806R (5′-GGAC-TACHVGGGTWTCTAAT-3′). Each 20 µL PCR reaction contained 4 µL of 5×TransStart FastPfu buffer, 2 µL of 2.5 mM Deoxyribonucleoside Triphosphates (dNTPs), 0.8 µL each of forward and reverse primers (5 µM), 0.4 µL TransStart FastPfu DNA polymerase, and 10 ng template DNA. Thermal cycling conditions were 95℃ for 3 min; 27 cycles of 95℃ for 30 s, 55℃ for 30 s, and 72℃ for 30 s; followed by a final extension at 72℃ for 10 min. Amplified products were purified using a PCR Clean-Up Kit (YuHua, China) after separation on a 2% agarose gel and quantified with a Qubit 4.0 fluorometer (Thermo Fisher Scientific, USA). Sequencing libraries were constructed using the NEXTFLEX Rapid DNA-Seq Kit, involving adapter ligation, magnetic bead-based size selection, PCR enrichment, and final library purification. Paired-end sequencing (PE250/PE300) was performed on the Illumina platform by Shanghai Majorbio Bio-pharm Technology Co., Ltd. (The raw data are deposited in the SRA database: PRJNA1238203).

### Bioinformatics analysis

Raw reads were quality-filtered using fastp (v.0.19.6) to trim low-quality bases (Phred score < 20), remove reads shorter than 50 bp, and discard reads containing ambiguous nucleotides. Overlapping paired-end reads were merged with FLASH (v.1.2.11) using a minimum overlap of 10 bp and a maximum mismatch ratio of 0.1. Chimeric sequences were identified and removed via reference-based filtering against the SILVA database (v.138) using UPARSE (v.7.1). Operational taxonomic units (OTUs) were clustered at 97% similarity, and sequences classified as chloroplast or mitochondrial origin were excluded. To standardize sequencing depth, all samples were rarefied to 20,000 reads, achieving a Good's coverage of 99.09%. Taxonomic annotation was performed using the RDP Classifier (v.2.11) against the SILVA database with a 70% confidence threshold. Functional profiles were predicted using PICRUSt2 (v.2.2.0) based on KEGG pathways.

## Metabolome analysis

The Liquid Chromatography-Tandem Mass Spectrometry (LC-MS/MS) analysis of the sample was conducted on a Thermo UHPLC-Q Exactive HF-X system equipped with an ACQUITY HSS T3 column (100 mm × 2.1 mm i.d., 1.8 µm; Waters, USA) at Majorbio Bio-Pharm Technology Co., Ltd. (Shanghai, China). The mobile phases consisted of 0.1% formic acid in water:acetonitrile (95:5, vol/vol) (solvent A) and 0.1% formic acid in acetonitrile:isopropanol:water (47.5:47.5, vol/vol) (solvent B). MS conditions were as follows: The flow rate was 0.40 mL/min and the column temperature was 40℃. The injection volume was 3 µL. The mass spectrometric data were collected using a Thermo UHPLC-Q Exactive HF-X Mass Spectrometer equipped with an electrospray ionization source operating in positive mode and negative mode. The optimal conditions were set as follows: aux gas heating temperature at 425℃; capillary temp at 325℃; sheath gas flow rate at 50 psi; aux gas flow rate at 13 psi; ionspray voltage floating at −3,500 V in

negative mode and 3,500 V in positive mode, respectively; normalized collision energy, 20-40-60 eV rolling for MS/MS. Full MS resolution was 60,000, and MS/MS resolution was 7,500. Data acquisition was performed with the data-dependent acquisition mode. The detection was carried out over a mass range of 70 m/z–1,050 m/z.

## Data processing

Soil physicochemical properties and soil enzyme activities were analyzed by one-way analysis of variance (ANOVA) using SPSS 25.0 software at $P \leq 0.05$. Biological statistical analysis was performed on bacterial and fungal OTUs clustered at 97% sequence similarity using Uparse (version 7.0.1090). Alpha diversity was calculated using Mothur (version 1.30.2), including Chao 1, Shannon, Simpson, ACE, and Coverage. To analyze the differences in alpha diversity between groups, the Wilcoxon rank-sum test was employed. Principal coordinate analysis (PCoA) based on the Bray-Curtis distance algorithm was conducted to examine the similarity of microbial community structures among samples. The significance of differences in microbial community structure between sample groups was further assessed using permutational multivariate analysis of variance (PERMANOVA). Species differences were analyzed using the Kruskal-Wallis rank-sum test (Kruskal-Wallis H test). Bacterial and fungal taxa with significantly different abundances from phylum to genus level between groups were identified using LEfSe analysis (linear discriminant analysis [LDA] effect size) (LDA score > 3, $P < 0.05$). Variance analysis was performed on the preprocessed metabolomics matrix file ($\log_2$-transformed and normalized using Pareto scaling) using one-way ANOVA ($P < 0.05$, FDR-corrected) and partial least squares-discriminant analysis (PLS-DA) in MetaboAnalyst 5.0. Orthogonal Partial Least Squares Discriminant Analysis (OPLS-DA) of the metabolomics data was performed using the R software package 'ropls' (version 1.6.2), with a total of seven cycles of cross-validation to assess the stability of the model. The metabolites with variable importance in projection (VIP) > 1, $P < 0.05$, were determined as significantly different metabolites based on the VIP obtained by the OPLS-DA model and the $P$-value generated by Student's $t$-test. Differential metabolites among two groups were mapped into their biochemical pathways through metabolic enrichment and pathway analysis based on the KEGG database (http://www. genome.jp/kegg/). These metabolites could be classified according to the pathways they were involved in or the functions they performed. Enrichment analysis was used to analyze a group of metabolites in a function node, whether they appear or not. The principle was that the annotation analysis of a single metabolite develops into an annotation analysis of a group of metabolites. Python package "scipy.stats" (https://docs.scipy.org/doc/scipy/) was used to perform enrichment analysis to obtain the most relevant biological pathways for experimental treatments.

## RESULTS

### Effect of fertilization on soil chemical properties and enzyme activities

The investigation into the impact of various treatments on soil chemical composition revealed pronounced effects, as evidenced by the substantial enhancement in the concentrations of readily available nitrogen, phosphorus, potassium, and soil organic matter in the NPK, OF, and MF treatments (Fig. 1), relative to the control group. Notably, within the treatment group, the MF and OF treatments exhibited comparable levels of readily available nitrogen, phosphorus, and potassium, with no significant differences observed between them. Both MF and OF significantly outperformed the NPK treatment in terms of readily available nitrogen and potassium, while for readily available phosphorus, MF and OF showed similar concentrations, both significantly higher than NPK. Regarding soil organic matter content, MF and OF treatments did not significantly differ from each other, but both significantly surpassed the NPK treatment. Furthermore, the MF treatment significantly elevated soil pH, aligning it closer to neutrality compared to the other treatment groups.

Soil enzyme activity, a pivotal metric for assessing soil fertility, was examined in this study, encompassing the activities of catalase, urease, and sucrase across the

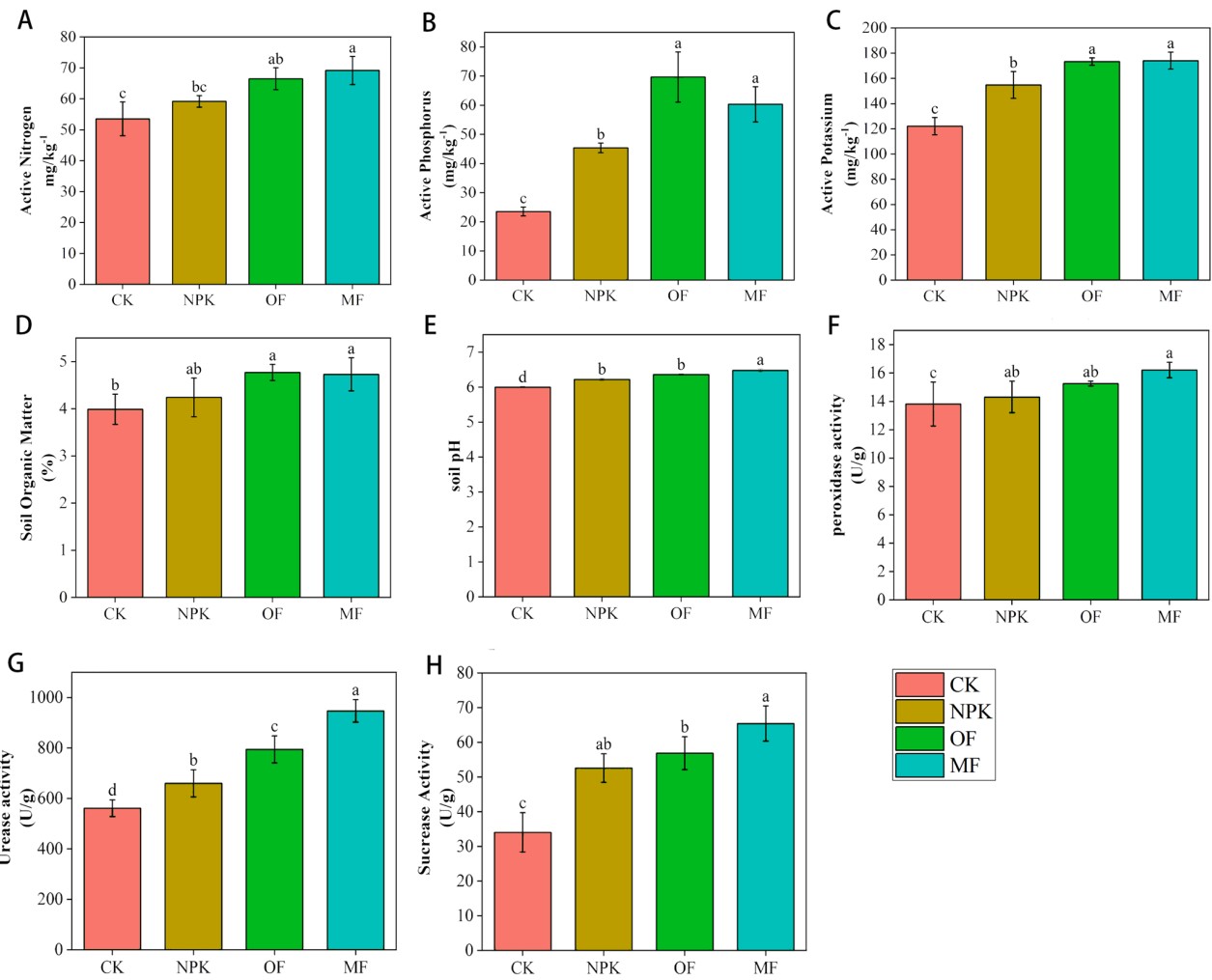

**FIG 1** Effects of different fertilization treatments on soil physicochemical properties and enzyme activities: (A) active nitrogen, (B) active phosphorus, (C) active potassium, (D) organic matter, (E) soil pH, (F) catalase, (G) urease, (H) sucrose synthase. Data are presented as mean ± SD (*n* = 3). For each treatment, three independent soil samples were collected from the rhizosphere of three representative tobacco plants, with each sample analyzed in three technical replicates. Statistical analysis was performed using one-way ANOVA followed by Tukey's *post hoc* test (*P* < 0.05). Different lowercase letters indicate significant differences among treatments.

control and treatment groups (Fig. 1). The findings underscored a marked elevation in enzyme activities for all treatment groups in comparison to the control, indicative of enhanced soil biological processes. Specifically, in terms of urease and sucrase activities, the observed differences among the treatment groups were statistically significant and demonstrated a discernible hierarchical pattern, with MF > OF > NPK. Similarly, for catalase activity, while the trend of MF > OF > NPK persisted, the distinctions among the treatment groups failed to reach statistical significance.

## Effect of fertilization on plant agronomic traits and aroma precursors

The agronomic traits of plants, which serve as direct indicators of their growth and developmental status, were comprehensively evaluated in this experiment by measuring height, stem circumference, and leaf length and width across various growth stages for each group.(Fig. 2) The outcomes revealed remarkable superiority in the agronomic indices of the treatment groups over the control group. Among the treatment cohorts, a consistent trend of MF > OF > NPK was discernible, with similar patterns observed at every measurement point. Notably, the plant growth profiles of MF and OF were closely

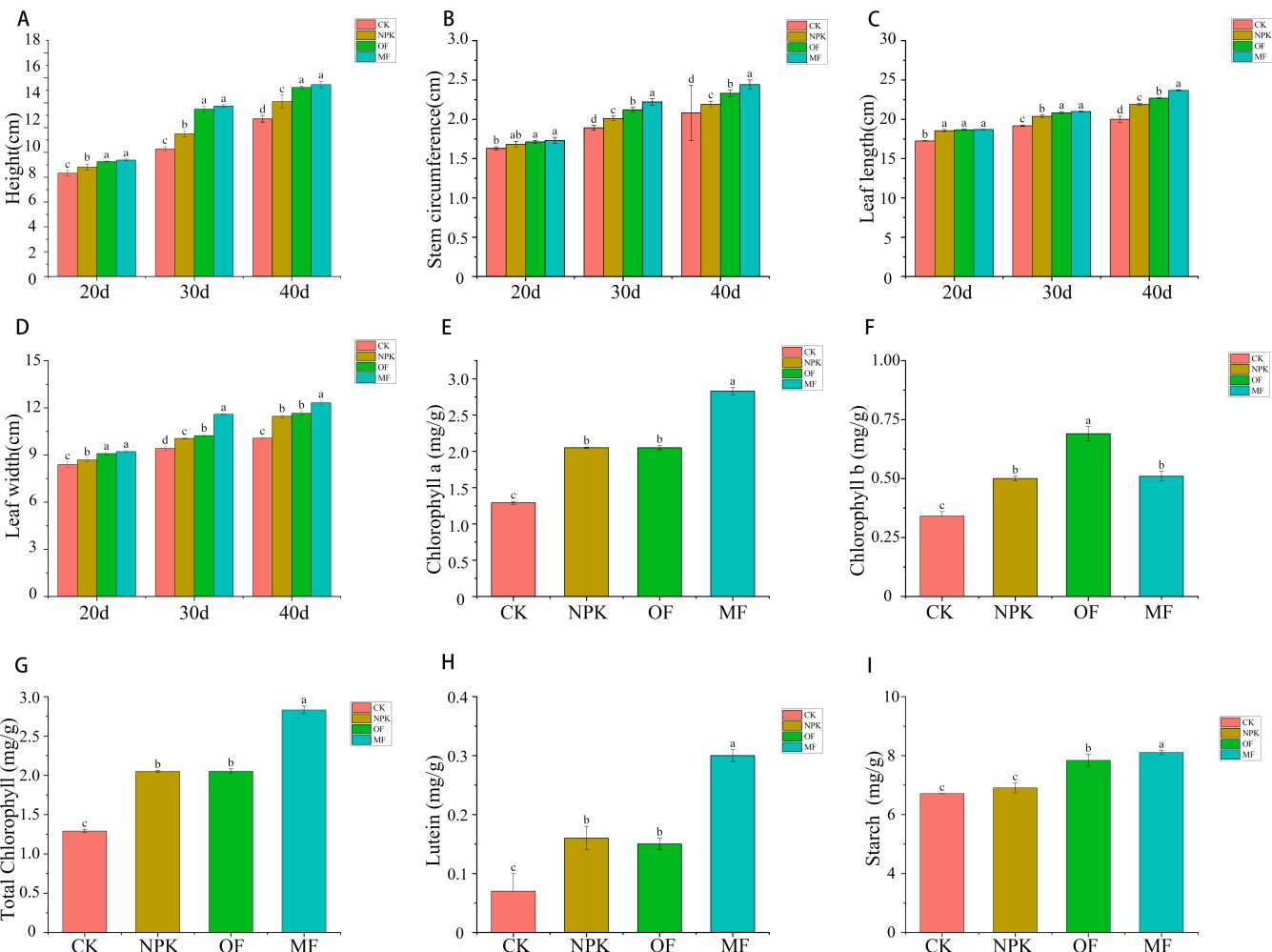

**FIG 2** Effects of different fertilization treatments on plant agronomic traits and aroma precursors: (A) height, (B) stem circumference, (C) leaf length, (D) leaf width, (E) chlorophyll a, (F) chlorophyll b, (G) total chlorophyll, (H) lutein, (I) starch. Data are presented as mean ± SD ($n$ = 3). For each treatment, three independent soil samples were collected from the rhizosphere of three representative tobacco plants, with each sample analyzed in three technical replicates. Statistical analysis was performed using one-way ANOVA followed by Tukey's *post hoc* test ($P < 0.05$). Different lowercase letters indicate significant differences among treatments.

aligned, particularly in terms of height and leaf length, exhibiting minimal significant differences.

Additionally, the content of aroma precursors, a crucial quality parameter in tobacco, was examined by quantifying chlorophyll a, chlorophyll b, total chlorophyll, lutein, and starch levels in samples from distinct groups. Compared to the control (CK), a marked increase in aroma precursor content was observed in the treatment groups. Specifically, in chlorophyll and lutein content, a distinct hierarchy emerged, with OF > MF > NPK, and these differences among treatments were statistically significant. Conversely, in starch content, a similar trend was observed, albeit with MF > OF > NPK, where the discrepancy between MF and OF was relatively minor.

## Changes in rhizospheric microbial diversity of tobacco

The Venn diagram illustrates the shared and unique OTUs among different fertilization treatments, based on 97% similarity clustering of high-quality 16S rRNA and Internal Transcribed Spacer (ITS) gene sequences. A total of 1,760,579 raw reads were obtained from high-throughput sequencing, with 1,662,187 high-quality reads retained after

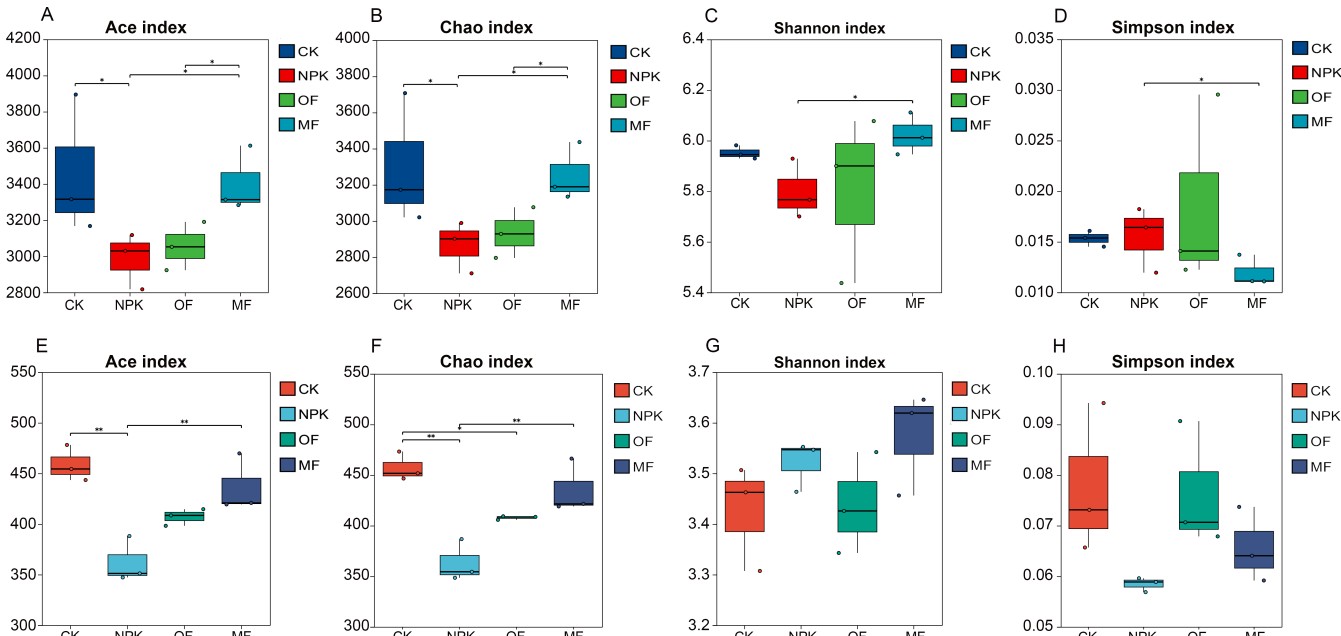

**FIG 3** Effects of fertilization treatments on rhizospheric soil alpha diversity. (A–D) Bacterial communities: (A) Ace index, (B) Chao index, (C) Shannon index, (D) Simpson index. (F–H) Fungal communities: (E) Ace index, (F) Chao index, (G) Shannon index, (H) Simpson index. Significant differences were analyzed by one-way ANOVA (*$P < 0.01$, **$P < 0.001$).

quality filtering. These reads were clustered into OTUs, resulting in 2,214 bacterial shared OTUs and 296 fungal shared OTUs. In the bacterial community, 843, 630, 582, and 677 OTUs were specific to the CK, NPK, OF, and MF treatments, respectively (Fig. 3A). In the fungal community, 172, 106, 129, and 175 OTUs were specific to each treatment, respectively (Fig. 3B). Among the bacterial OTUs, the CK treatment exhibited the highest number of unique OTUs (843), followed by MF (677), NPK (630), and OF (582). In contrast, for fungal OTUs, the MF treatment showed the highest number of unique OTUs (175), followed by CK (172), OF (129), and NPK (106).

## Characterization of microbial communities

Alpha diversity metrics, including Shannon, Simpson, Chao, Ace, and Coverage indices, were employed to characterize microbial community richness, diversity, and evenness. The coverage index (>97% across all groups) confirmed sufficient sequencing depth (Good's estimator < 3%), ensuring the detected taxa reliably represented true biological communities (Fig. S2). Consistent patterns between Chao and Ace indices (Fig. 3A and B) demonstrated robust estimation of bacterial richness: MF maintained comparable richness to CK, while both NPK and OF exhibited significant reductions ($P < 0.01$). Notably, NPK unexpectedly showed lower richness than CK, revealing distinct impacts of fertilization regimes. Subsequent Shannon and Simpson analyses (Fig. 3C and D) revealed elevated bacterial diversity in MF compared to NPK ($P < 0.01$), with Simpson indices ranking MF < OF < CK <NPK. In contrast to bacterial responses, fungal communities exhibited divergent patterns. While fungal Chao/Ace indices (Fig. 3E and F) similarly indicated universal richness declines under treatments (MF > OF > NPK in preservation efficiency), fungal diversity trends (Shannon: MF > NPK > CK >OF; Simpson: inverse pattern) showed no statistical significance ($P > 0.05$)(Fig. 3G and H). This decoupling between bacterial and fungal diversity responses suggests taxa-specific sensitivities to fertilization practices. Collectively, MF uniquely preserved soil microbial richness and mitigated diversity loss in bacteria, whereas conventional NPK and OF induced substantial alterations in community structure.

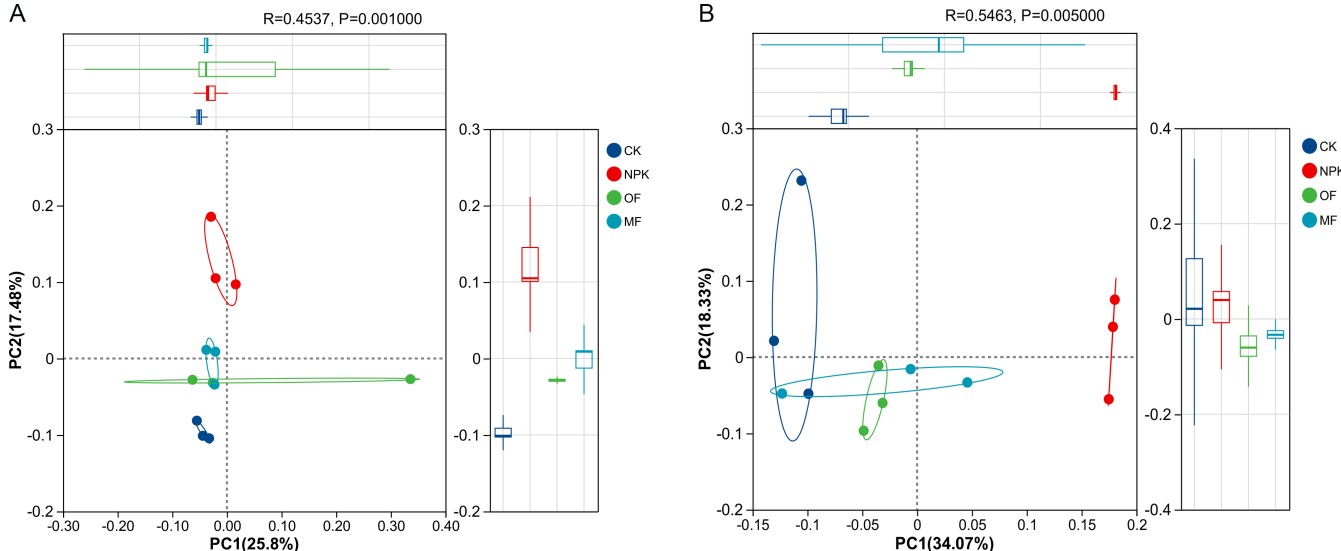

**FIG 4** Beta diversity analysis of rhizospheric soil bacterial (A) and fungal (B) communities based on OTU levels. PERMANOVA with 999 permutations was performed to assess group differences (bacteria: $R = 0.4573$, $P = 0.001$. fungi: $R = 0.5463$, $P = 0.005$).

PCoA analyses based on Bray-Curtis dissimilarity revealed distinct clustering patterns in soil bacterial and fungal beta diversity across treatments. For bacterial communities (Fig. 4A), PERMANOVA confirmed significant separation among treatment groups ($R = 0.4573$, $P = 0.001$, 999 permutations), with each treatment cohort exhibiting unique architectural signatures and greater intra-group convergence compared to the control. Similarly, fungal communities (Fig. 4B) showed pronounced segregation between fertilized and control groups ($R = 0.5463$, $P = 0.005$), underscoring the profound impact of fertilization on restructuring soil microbial assemblages. Notably, the larger $R$ value for fungi (54.6% variance explained vs 45.7% for bacteria) suggests that fertilization elicited stronger differentiation in fungal community composition, despite their lower overall perturbation susceptibility relative to bacterial communities.

## Effect of fertilization on the structural composition of microbial communities

The microbial community structure of each experimental group was profiled using 16S rRNA gene amplicon sequencing for bacteria and ITS amplicon sequencing for fungi on Illumina PE300/PE250 platform. For downstream analysis, we focused on the top 10 most abundant bacterial and fungal phyla, along with the top 15 most abundant genera. (Fig. 5). For bacterial phylum-level species abundance, results showed that the dominant phyla identified in the four treatments were Actinobacteria, Proteobacteria, Chloroflexi, Firmicutes, and Bacteroidota (Fig. 5A). We further examined the abundance of species at the genus level of the bacterial community, and the dominant genera were *Kirbbella*, *Streptomyces*, *Lysobacter*, *Bacillus*, and *Devosia* (Fig. 5B).

For fungal phylum-level species abundance, it was found that Ascomycota was the most abundant fungal community followed by Basidiomycota and Mortierellomycota (Fig. 5C). At the genus level, the most abundant fungi genus is *Arthrobotrys* followed by *Unclassified_p_Ascomycota*, *Myceliopthora*, *Unclassified_f_stephanosporaceae*, and *Chaetomium* (Fig. 6D).

Significant differences in microbial taxa abundance across treatments were assessed using the Kruskal-Wallis H test ($P < 0.05$), followed by LEfSe analysis (LDA score > 3.0) to identify biomarkers driving group differentiation (Fig. S3). The Kruskal-Wallis test revealed that the MF treatment harbored distinct bacterial genera (*Rhodanobacter*, *Pseudolabrys*, *Gemmatimonas*, and *Terrabacter*) and a fungal genus (*Microascus*) with significant abundance variations compared to other groups (Fig. 6). LEfSe further validated these findings, highlighting *Rhodanobacter* (LDA = 3.861, $P = 0.01879$),

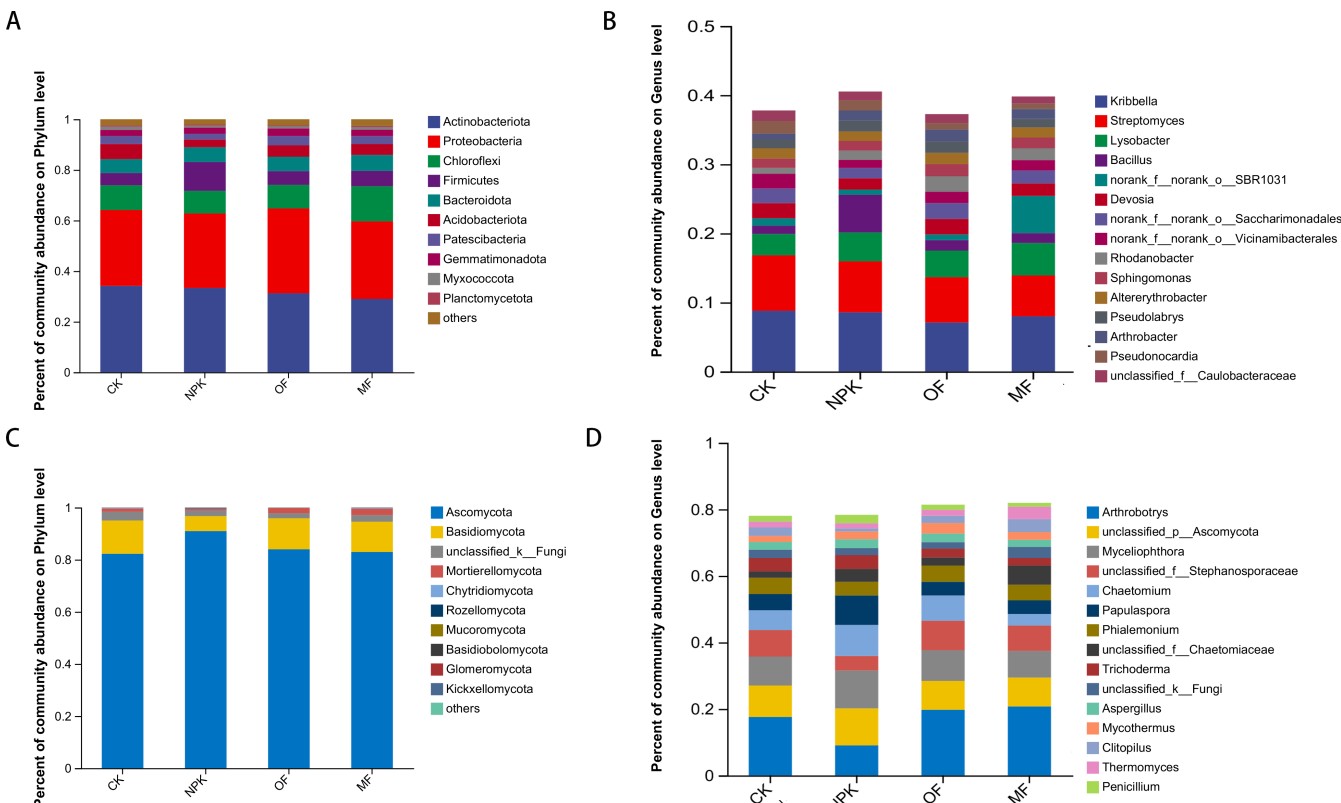

**FIG 5** Analysis of soil microbial composition. Bacteria: (A) phylum level, (B) genus level; fungi: (C) phylum level, (D) genus level.

*Pseudolabrys* (R = LDA = 3.378, *P* = 0.03446), and *Microascus* (LDA = 4.274, *P* = 0.03879) as key discriminative taxa in the MF group (Fig. S4).

## Association between microorganisms and soil properties and plant traits

In order to elucidate the pivotal role of prevalent bacterial groups in modulating plant growth and development, a comprehensive analysis was undertaken correlating the dominant bacterial genera within the top 15 category with various soil properties and plant phenotypic traits. This analysis yielded a correlation heatmap (Fig. 7), which provided insights into the intricate interrelationships.

The findings revealed a discernible pattern, wherein five dominant bacterial genera exhibited significant and positive correlations with soil attributes. Notably, *Rhodanobacter* demonstrated a robust association with all soil indicators except for soil organic matter (SOM), whereas *Bacillus* was significantly linked to SOM. *Pseudolabrys*, *Sphingomonas*, and *Arthrobacter* showed significant positive correlations with soil sucrose synthase, available potassium (AK), and soil pH, respectively. Conversely, four bacterial genera manifested significant negative correlations with soil properties. *Pseudonocardia* stood out, displaying a pronounced negative relationship with ammonium nitrogen, available phosphorus (AP), AK, and pH. This was followed by *Streptomyces*, *unclassified_f__Caulobacteraceae*, and *Kirbbella*, each exhibiting negative associations with varying degrees of significance, particularly with AP, AK, Soil Urease (S-UE), and soil pH.

Regarding plant traits, four dominant bacterial genera exhibited significant positive correlations. *Rhodanobacter* stood out with a strong positive correlation across all traits, followed by *Arthrobacter*, *Sphingomonas*, and *Pseudotabry*s, which displayed significant positive relationships with the majority of plant indicators. In contrast, four bacterial genera, including *Pseudonocardia*, demonstrated significant negative correlations with plant traits. Specifically, *Pseudonocardia* had a notable negative impact on plant height, stem circumference, leaf dimensions (length and width), chlorophyll content (a, b, and

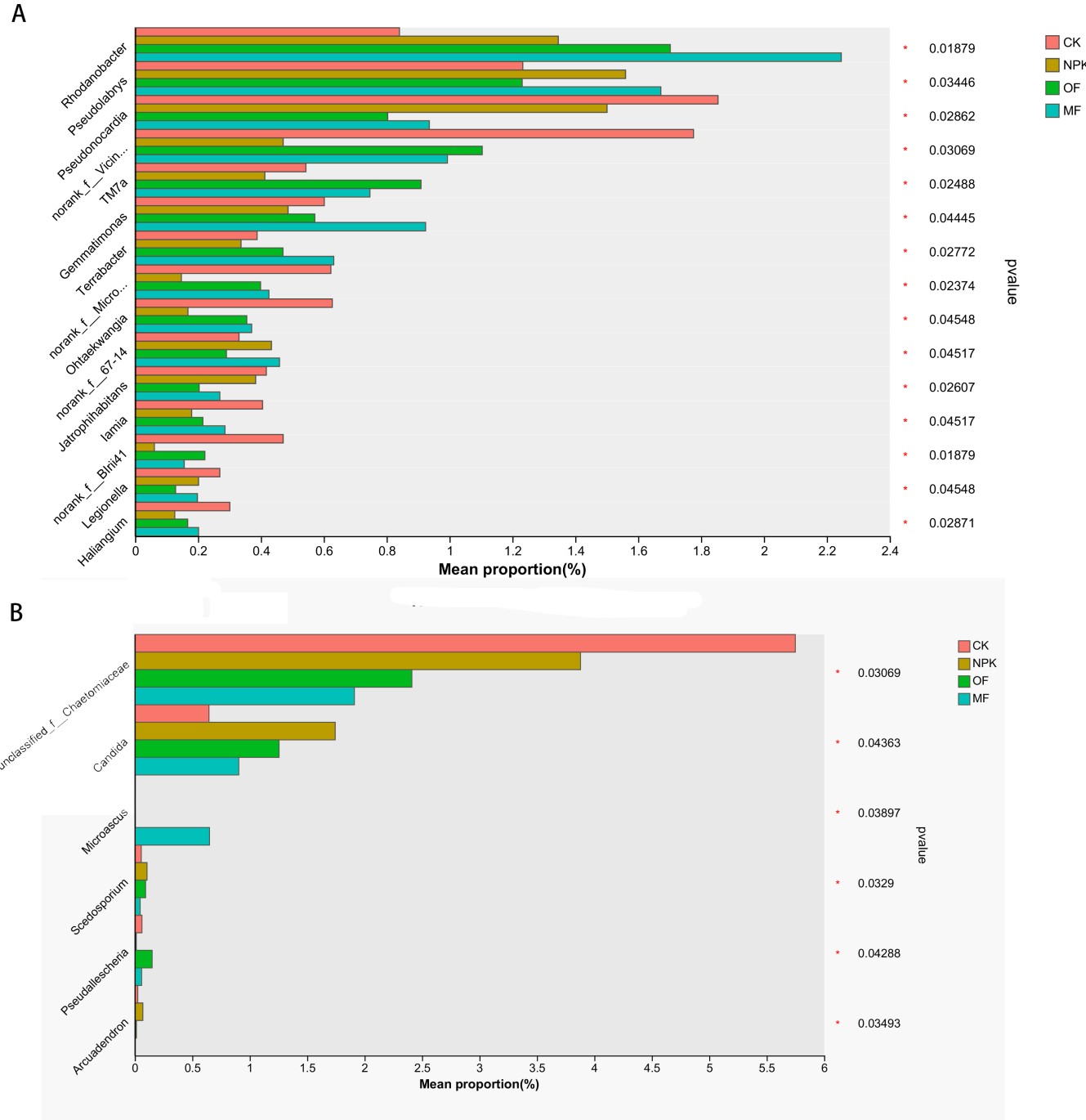

**FIG 6** Significantly different microorganisms between the control and treatment groups: (A) bacteria, (B) fungi.

total), and starch levels. *Streptomyces* and *unclassified_f__Caulobacteraceae* also showed negative correlations, but their significance was confined to select plant metrics.

A notable observation was that, while the majority of the dominant fungal genera exhibited positive correlations with all indicators, these associations failed to reach statistical significance. In contrast, a distinct pattern emerged with *unclassified_f__Chaetomiacea*, which displayed a significant negative correlation with the majority of the investigated metrics. Furthermore, *unclassified_k__Fungi* also manifested a negative correlation with most indicators, particularly evincing a statistically significant negative relationship with chlorophyll b, a crucial photosynthetic pigment. Thus, compared to

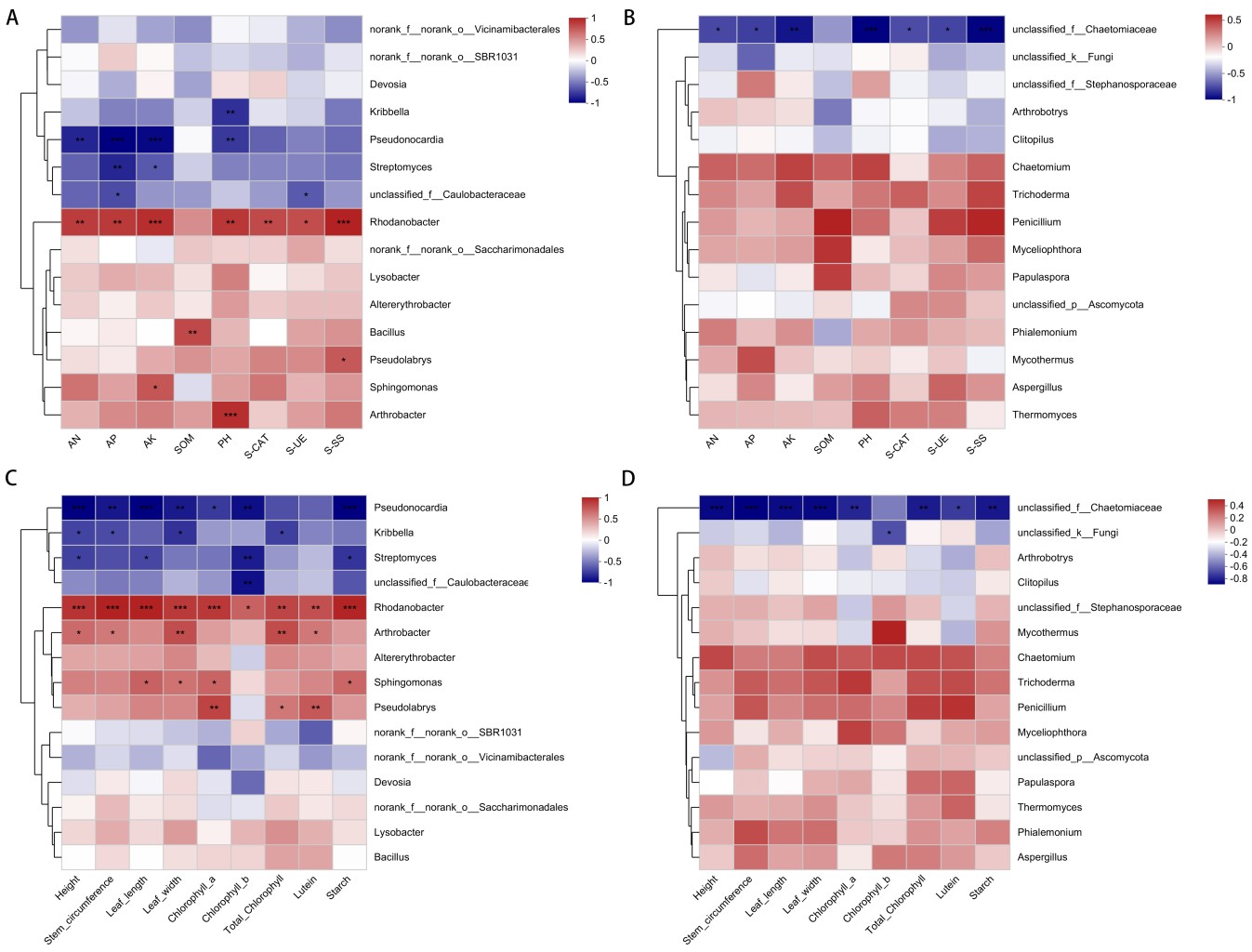

**FIG 7** Association between microorganisms and soil chemical properties and enzyme activities: (A) bacteria, (B) fungi; association between microorganisms and plant agronomic traits and aroma precursors: (C) bacteria, (D) fungi.

fungi, the genus bacteria are more closely related to the growth and development process of plants.

## Effect of different fertilization treatments on metabolites of tobacco leaves

All small molecule metabolites in the samples were detected and analyzed using LC-MS analysis, and a total of 404 plant secondary metabolites were identified, including 198 terpenoids, 56 phenolic acids, 36 steroids, 33 organic acids, 22 flavonoids, 18 coumarins, 14 alkaloids, 12 indole, 5 quinones, 4 lignans, 4 stilbenes, and 2 tannins. Partial Least Squares Discriminant Analysis (PLS-DA) was performed on the intergroup samples to compare the metabolite profiles between different groups (Fig. 8A). The experiment proved that the metabolite separation was obvious between the treated group and the control group, and the metabolite separation was obvious between MF and OF and NPK. In order to further demonstrate the effect of MF on plant metabolic processes, the changes of tobacco metabolites among different groups were analyzed by differential volcano mapping (Fig. 8B through D). A total of 323 differential metabolites were identified in MF vs CK (up = 306, down = 17), 328 differential metabolites in MF vs NPK (up = 250, down = 78), and 151 differential metabolites in MF vs OF (up = 180, down = 43).

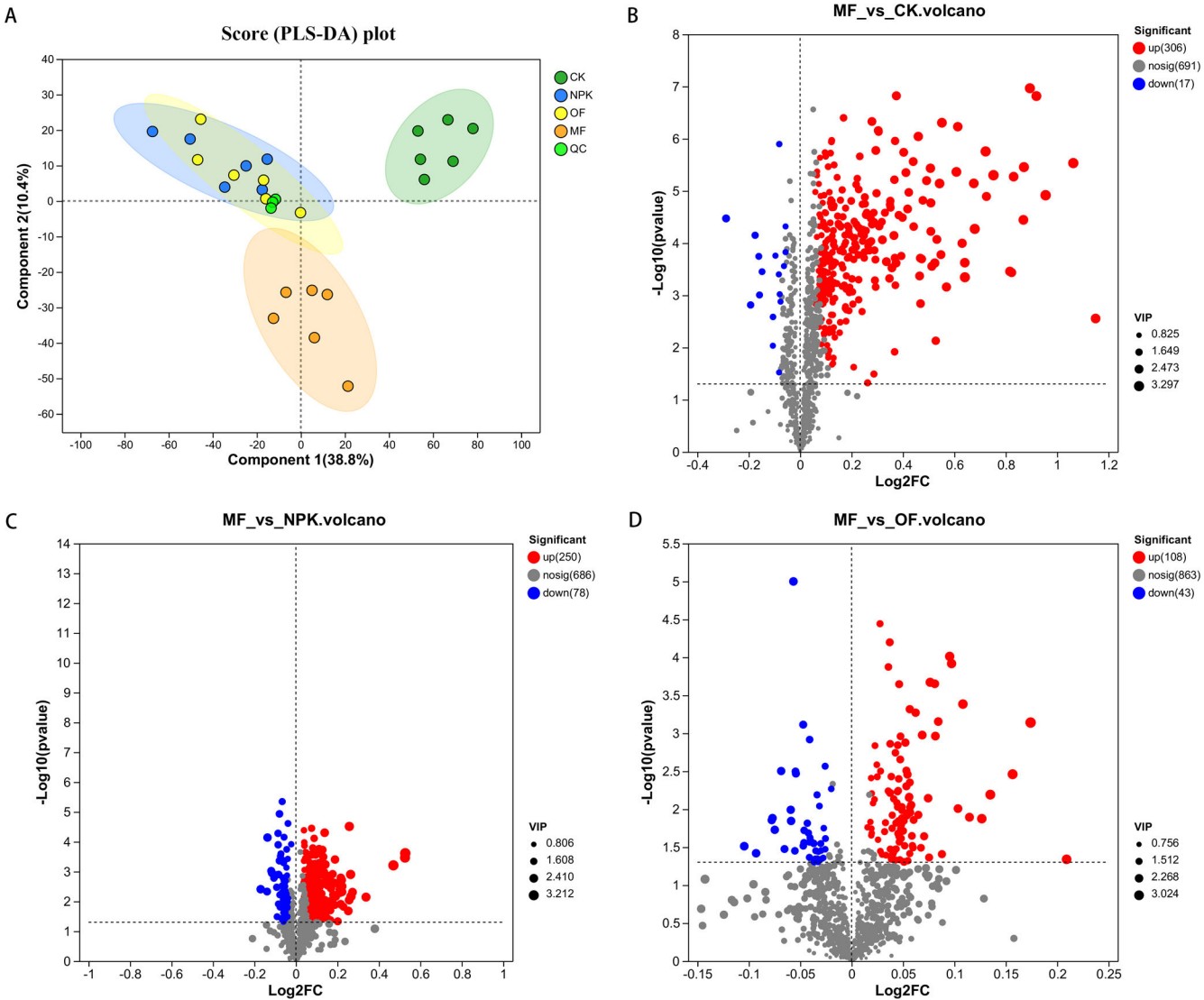

**FIG 8** (A) PLS-DA on metabolites between different groups; volcano plot of differential metabolites between groups: (B) MF vs CK, (C) MF vs NPK, (D) MF vs OF.

The utilization of the KEGG database facilitated the enhancement and identification of critical metabolic pathways within the observed differential metabolome (Fig. 9). A two-by-two comparison with CK, NP, OF, using MF as a baseline, showed that the metabolic pathways significantly enriched in MF vs CK were "linoleic acid metabolism," "alpha-linolenic acid metabolism," "plant hormone signal transduction," "arachidonic acid metabolism," and "arginine biosynthesis." The metabolic pathways that were significantly enriched in MF vs NPK were "alpha-linolenic acid metabolism," "plant hormone signal transduction," "linoleic acid metabolism," "arachidonic acid metabolism," and "cutin, suberine and wax biosynthesis." The metabolic pathways significantly enriched in MF vs OF were "plant hormone signal transduction," "phenylalanine metabolism," "alpha-linolenic acid metabolism," and "linoleic acid metabolism."

Given the predominantly convergent nature of the enriched metabolic routes across the various groups, a deeper interrogation into the intergroup variations in metabolic dynamics was pursued. We systematically identified the top 20 metabolites exhibiting statistically significant differences across each comparative group, subsequently subjecting these to cluster analysis. The results showed 18 up-regulated metabolites in MF vs CK (six lipids, three terpenes, one phenolic acid, one coumarin, one carbohydrate,

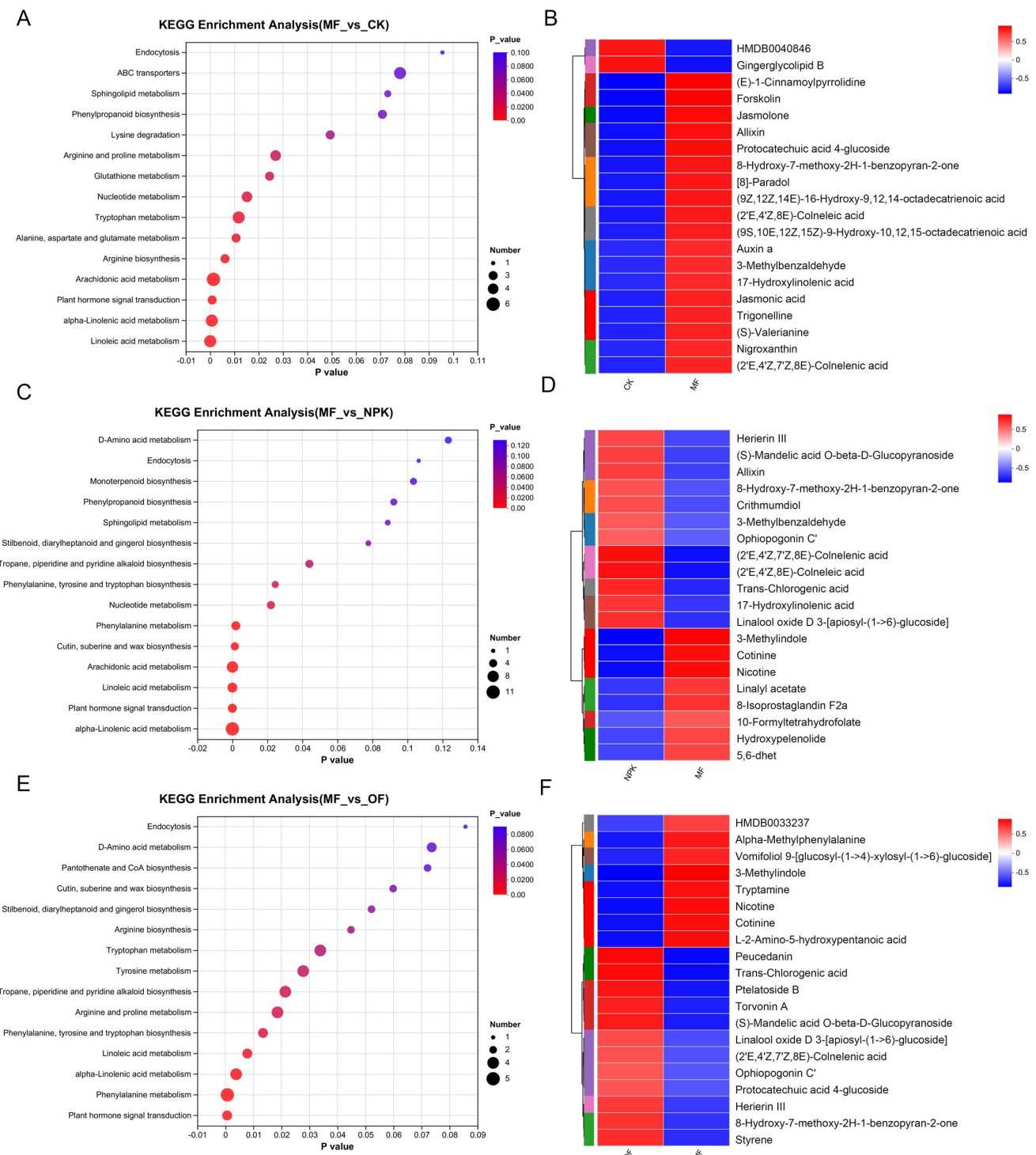

**FIG 9** KEGG metabolic pathway enrichment between groups (A) MF vs CK, (C) MF vs NPK, (E) MF vs OF. The horizontal coordinates are the *P*-value of the significance of the enrichment, and the smaller *P*-value is statistically significant, and a *P*-value of less than 0.05 is generally considered to be a function of a significantly enriched term; the vertical coordinate is KEGG pathway. The size of the bubbles in the graph represents how much of the pathway is enriched to the metabolic set of compounds; heatmap of clustering of up- and down-regulated differential metabolites between groups: (B) MF vs CK, (D) MF vs NPK, (F) MF vs OF.

one alkaloid, and five others) and two down-regulated (one carbohydrate and one lipid); eight up-regulated in MF vs NPK (two terpenes, two lipids, one indole, and

three others) and 12 down-regulated (four lipids, two carbohydrates, one coumarin, one steroid, one phenolic acid, and three others); eight up-regulated in MF vs OF (two carbohydrates, two indoles, one amino acid, one phenolic acid, and two others) and 12 down-regulated (four carbohydrates, two steroids, two coumarins, one phenolic acid, one lipid, and two others). Notably, in both MF vs NPK and MF vs OF comparisons, we identified nine overlapping metabolites, which included two carbohydrates (linalool oxide D 3-[apiosyl-(1→6)-glucoside], (S)-mandelic acid O-β-D-glucopyranoside), one lipid ((2'E,4'Z,7'Z,8E)-colnelenic acid), one coumarin (8-hydroxy-7-methoxy-2H-1-benzopyran-2-one), one steroid (Ophiopogonin C'), one indole (3-methylindole), and three others (cotinine, nicotine, herierin III). These nine metabolites were extracted separately from the control group and three treatment groups (CK, NPK, OF, MF) and visualized using stacked bar charts to investigate their quantitative differences across treatments (Fig. S5). The results demonstrated significant differences between the CK group and the treatment groups, further confirming that fertilization treatments markedly altered the metabolic profiles of tobacco. Among the treatment groups, MF exhibited notable distinctions from both NPK and OF. Specifically, the contents of 3-methylindole, nicotine, and cotinine in MF were significantly higher than those in OF and NPK, whereas two glycosides ((S)-mandelic acid O-beta-D-glucopyranoside and linalool oxide D 3-[apiosyl-(1→6)-glucoside]) were significantly lower in MF compared to NPK and OF.

## DISCUSSION

### Effect of fertilizer application on soil properties and plant traits

Soil available nutrients serve as critical indicators for assessing soil fertility and underpinning agricultural productivity. In this study, we conducted a comparative analysis of microbial fertilizer, conventional chemical fertilizer, and organic fertilizer on tobacco growth, with a particular focus on their impacts on soil nutrient dynamics. Our results demonstrated that both MF and OF exhibited superior capacity in enhancing soil available nutrient levels compared to NPK. The efficacy of MF in improving soil nutrient availability aligns with previous findings (31–35), while its similarity to OF in this regard has also been documented (36). The application of MF significantly ameliorated soil acidity, a phenomenon likely attributable to enhanced microbial activity in decomposing soil organic matter. This possible process accelerates the degradation of humus and other acidic compounds, thereby modulating soil pH toward conditions more favorable for plant growth (37, 38). Soil enzyme activities, which are instrumental in regulating soil fertility and fostering plant growth and development, were also examined. Aligning with previous research, the application of MFs led to a marked enhancement in the activities of key enzymes, including sucrase, urease, and catalase. This observation confirmed that the beneficial microorganisms inherent in MF actively participate in the decomposition of soil organic matter, secreting a diverse array of enzymes during their proliferation. These enzymes contribute to an overall elevation in soil enzyme levels, facilitating nutrient transformations such as nitrogen and phosphorus fixation, thereby enhancing soil nutrient availability and promoting plant health (39–41).

The assessment of agronomic traits in plants is paramount for quantifying their growth and developmental progress, as evidenced by extensive literature (42–44). Prior research has underscored the remarkable potential of microbial fertilizers in enhancing plant height, spike length, and yield in wheat (45), as well as leaf length and width in tobacco (46). Our current study aligns with these findings. During the early growth stage (20 days), no statistically significant differences ($P > 0.05$) were observed in agronomic traits—including plant height, stem circumference, leaf length, and leaf width—across the MF, OF, and NPK treatments. However, as the trial progressed (30 days, 40 days), MF-treated plants progressively exhibited superior performance in these parameters compared to both OF and NPK groups. This temporal divergence was most evident in stem circumference and leaf width. This superiority can be attributed to MF's more effective enhancement of the soil nutrient profile, thereby augmenting nutrient availability to plants, and microbial-mediated nutrient mobilization and root-microbe

interactions may require extended periods to substantially influence plant morphological development. Chlorophyll, a vital aroma precursor in tobacco, undergoes degradation during tobacco processing to yield crucial aroma compounds like neophytadiene and phytofuran. Attia et al. (47) have reported that plant growth-promoting rhizobacteria can significantly elevate total chlorophyll content and the specific fractions of chlorophyll a and b in tomato. Our experimental results echo this observation, revealing a notable increase in chlorophyll content in tobacco plants fertilized with MFs compared to NPKs and OFs. This suggests that MF can effectively increase the content of aroma components in tobacco leaves by boosting chlorophyll levels. Furthermore, the presence of certain plant growth-promoting microorganisms within MF has been shown to mitigate both biotic and abiotic stresses, thereby fortifying plant resilience. This mechanism may represent a pivotal avenue through which MFs exert their beneficial effects, as supported by existing literature (48, 49). Meanwhile, we emphasize that 40 day data just elucidate developmental mechanisms, and future validation with mature-stage sampling is essential to link early patterns to final quality parameters.

## Microorganisms in the soil

The diversity and composition of soil microbial communities hold paramount importance in sustaining the health and functional integrity of soil ecosystems, as emphasized in previous studies (50–52). Shan et al. (53) have highlighted the contribution of microbial inoculation to enhancing soil microbial diversity and fostering the reorganization of microbial community structures. Our current experimental findings concur with these observations, revealing that MF uniquely preserved soil microbial richness and mitigated diversity loss in bacteria, whereas conventional NPK and organic fertilizers induced substantial alterations in community structure. This enhancement can be attributed to the improved nutrient availability conferred by MFs, which fosters the proliferation and swift colonization of beneficial bacteria within the soil matrix (54–56). Despite the application of varying fertilizer types, the species composition of major soil microbial genera remained unaltered; however, their abundance was notably influenced. Specifically, *Kirbbella*, *Streptomyces*, *Lysobacter*, *Bacillus*, and *Devosia* emerged as the dominant bacterial genera, whereas *Arthrobotrys*, *Myceliopthora*, and *Chaetomium* constituted the preeminent fungal genera in the soil. To assess the impact of fertilization on microbial community composition, PCoA based on Bray-Curtis dissimilarity was performed. For bacterial communities (Fig. 5A), PERMANOVA revealed significant structural divergence across treatments ($R = 0.4573$, $P = 0.001$), with MF forming a distinct cluster separate from NPK, OF, and CK, indicating that microbial fertilizer drove distinct bacterial community assembly. In contrast, fungal communities (Fig. 5B) exhibited stronger overall separation ($R = 0.5463$, $P = 0.005$), yet partial overlap persisted between MF and conventional fertilizer groups. This aligns with previous observations that fungal communities display greater compositional flexibility under fertilization regimes compared to bacterial communities (57–59).

Kruskal-Wallis H test and LEfSe analysis identified *Rhodanobacter* (LDA = 3.861, $P = 0.01879$) and *Pseudolabrys* ($R = $ LDA = 3.378, $P = 0.03446$) as discriminative taxa in the MF treatment. While these genera were present across all treatments, their abundance was significantly higher in MF (Fig. 7). *Rhodanobacter* has been implicated in soil catalase and oxidase enzyme production (60), and the observed positive correlations between its abundance and soil enzyme activity, plant agronomic traits, and aroma precursor indices may suggest a potential functional linkage. This aligns with prior studies reporting associations between *Rhodanobacter* proliferation and crop yield improvements in Chinese cabbage systems, hinting at its possible role in enhancing nutrient mobilization and phytohormone regulation (61). Similarly, *Pseudolabrys*—a taxon linked to nitrogen fixation—showed correlations with soil available nutrients, potentially reflecting its rhizosphere colonization and root nodulation capabilities (62, 63). The relationship between *Pseudolabrys* and aroma precursors (e.g., chlorophyll) could indicate microbial modulation of leaf quality, though mechanistic validation remains necessary. The two

dominant bacterial genera (*Rhodanobacter* and *Pseudolabrys*) likely exhibit synergistic relationships with the Ascomycota fungus *Microascus: Rhodanobacter* degrades lignin derivatives and other complex aromatic compounds into small-molecule substrates (e.g., vanillic acid) for utilization by *Microascus*, while the fungus secretes laccase to pretreat recalcitrant lignin, facilitating bacterial degradation (64, 65). Simultaneously, the nitrogen-fixing actinobacterium *Pseudolabrys* converts atmospheric $N_2$ into $NH_4^+$ to support fungal amino acid synthesis, whereas Ascomycota fungi (e.g., *Fusarium*) reciprocate by secreting phytase to mobilize organic phosphorus, thereby promoting *Pseudolabrys* growth in nitrogen-enriched microzones (64, 66).

In this study, we introduced *Bacillus subtilis* and *Bacillus licheniformis* as well-characterized PGPR. Previous research has demonstrated that *B. subtilis* suppresses soil-borne pathogens through the secretion of lipopeptide antibiotics (e.g., surfactin and iturin) while enhancing root development and alleviating abiotic stress via the production of indole-3-acetic acid and 1-aminocyclopropane-1-carboxylic acid (ACC) deaminase (67). Complementarily, *B. licheniformis* exhibits robust phosphate solubilization and nitrogen fixation capabilities, with its secreted organic acids and extracellular polysaccharides not only mobilizing mineral-bound phosphorus but also improving soil aggregate stability to enhance nutrient retention (68). In our experiment, the co-inoculation of these two strains significantly increased tobacco stem diameter and leaf area index, accompanied by elevated activities of soil urease and sucrose synthase, aligning with prior reports of their growth-promoting and soil-modifying functions (46, 69–71). Despite the functional advantages of the exogenous strains, their rhizosphere colonization did not induce a significant increase in the overall abundance of *Bacillus*. This phenomenon may be attributed to resource competition and allelopathic inhibition between the introduced strains and native *Bacillus* populations—antibiotics such as bacillaene secreted by *B. subtilis*, while effective against pathogens, may concurrently suppress the proliferation of phylogenetically related microorganisms (72). In contrast, NPK-treated soils lacking exogenous antimicrobial compounds permitted unrestricted growth of native microbes, which likely failed to promote plant growth due to the absence of specialized functional traits (e.g., antibiotic synthesis), representing a "high-abundance, low-functionality" community structure. Future investigations should integrate time-series monitoring (e.g., qPCR-based quantification) with metabolic activity profiling of introduced strains (e.g., metatranscriptomics) to precisely delineate their dynamic functional contributions.

## Effect of different fertilizers on plant metabolism

The PLS-DA indicated that fertilizer application, particularly MF, significantly modulated plant metabolic activities. Further comparative analysis using volcano plots revealed a preponderance of up-regulated metabolites in MF-treated samples compared to both control and other treatment groups. The magnitude of up-regulation was substantially greater than that of down-regulation, suggesting that MF application significantly boosts plant metabolic levels. Differential metabolite enrichment analysis revealed certain convergences among enriched metabolic pathways across treatment groups (Fig. 8) were enriched in "linoleic acid metabolism," "alpha-Linolenic acid metabolism," and "plant hormone signal transduction," likely stemming from the similar impacts of different fertilizer treatments on plant basal metabolic processes (73, 74). In our analysis of the top 20 differential metabolite clusters derived from the comparisons of MF vs OF and MF vs NPK, we identified nine overlapping metabolites of particular interest. Notably, 3-methylindole (skatole) emerged as a significantly up-regulated metabolite in both comparative groups, echoing its potential positive contribution to flavor quality, as suggested by previous studies (75, 76). Specifically, moderate concentrations of skatole impart rich fruity and floral nuances to tea aroma, underscoring its significance as a key aroma source.

Conversely, we observed a notable down-regulation of two glucosides, namely (S)-mandelic acid O-beta-D-glucopyranoside and linalool oxide D 3-[apiosyl-(1→6)-glucoside], in both MF vs OF and MF vs NPK comparisons. This finding is consistent

with the recognition that degradation of volatile metabolites of glycosides bound to fatty acids, amino acids, and other substances can produce aromatic components (77). Furthermore, within the MF vs OF comparison, we detected a significant up-regulation of the oligosaccharide vomifoliol 9-[glucosyl-(1→4)-xylosyl-(1→6)-glucoside], indicative of MF's pronounced effect on carbohydrate metabolic pathways and its potential to foster the production of aroma-contributing substances in tobacco. Our findings demonstrate that MF treatment significantly elevated nicotine and its primary oxidative metabolite cotinine compared to other treatments and controls. Nicotine, a pyridine alkaloid unique to Solanaceae plants, typically accumulates at elevated concentrations during the mid-growth phase (30–60 Days Post Transplanting [DPT]) in tobacco (78), functioning as a potent defense compound against lepidopteran herbivores through neurotoxic activity (79, 80). Notably, the observed cotinine enrichment aligns with its established role in stress adaptation, as CYP82E4-mediated nicotine conversion to cotinine enhances drought and salinity tolerance through redox homeostasis modulation (81). These coordinated metabolic shifts suggest that MF application may amplify tobacco's innate chemical defense system while reinforcing abiotic stress resilience, potentially through microbiome-induced up-regulation of P450 enzyme networks and secondary metabolite biosynthesis.

Furthermore, our investigation revealed that in MF vs NPK, MF demonstrated a markedly decreased lipid content. This observation may stem from the microbial metabolism of lipids into an array of esters within tobacco leaves (82). This metabolic transformation enhances the tobacco's bouquet with floral and fruity nuances (83). Additionally, we observed significant up-regulation in the concentrations of several crucial compounds in MF vs NPK. Notably, linalyl acetate, a pivotal terpene and primary aroma constituent in lavender essential oils, was elevated. This compound is renowned for imparting a rich, sweet aroma (84). Another notable compound, hydroxypelenolide, a sesquiterpene lactone exhibiting vital biological activities, including antiviral, anti-inflammatory, and anti-cytotoxic properties, was also found to be up-regulated. Moreover, studies have underscored its potent antifeedant effects on insects (85, 86). In MF vs OF, our analysis further indicated a pronounced increase in indole derivatives, particularly tryptamine. As a biogenic amine, tryptamine is widely acknowledged for its contribution to plant growth and development processes. It is also capable of bolstering tobacco plants' resilience against pests and diseases (87).

## Conclusions

The present study has demonstrated that microbial fertilizers, through their ability to modify soil microbial community structure and tobacco leaf metabolic pathways, effectively promote tobacco growth, development, and quality enhancement, surpassing traditional chemical and organic fertilizers. Application of microbial fertilizers significantly increased the abundance of beneficial soil bacteria such as *Rhodanobacter* and *Pseudolabrys*. Furthermore, metabolomic analyses revealed that microbial fertilizers markedly enhanced tobacco's metabolic activity, elevating levels of multiple quality-enhancing metabolites. Consequently, we recommend the adoption of microbial fertilizers in tobacco field production as a strategy to minimize the reliance on conventional inorganic fertilizers, thereby ensuring stable soil nutrient levels and maintaining a productive soil microbial community structure conducive to optimal tobacco yield and quality.

## Highlight

1. MFs significantly boost abundance of beneficial bacterial genera (*Rhodanobacter* and *Pseudolabrys*), enhancing soil quality and plant growth and development.
2. Compared to traditional fertilizers, MF markedly elevates plant metabolic activity, especially in sugar, lipid, and amino acid production.

3. MF drastically enriches key aroma components in tobacco (e.g., linalyl acetate), enhancing the aromatic quality of tobacco leaves.

## ACKNOWLEDGMENTS

Thanks to all authors for their contribution to the manuscript. Thanks to Prof. Yu Zhang for his invaluable guidance and intellectual mentorship throughout the thesis writing process. Her critical insights on research methodology and constructive feedback during multiple revisions profoundly shaped the academic rigor of this work. Special thanks are due to Researcher Jiaxin Liu for her meticulous assistance in manuscript proofreading and technical corrections.

This work was supported by the Foundation of Taishan brand cigarette high-quality core raw material development and application in Shandong (202102004), and analysis of the characteristic styles and mellowing characteristics of American functional Roubaix tobaccos (202302004) and Construction and Application of Quality Management Control Model for Tobacco Production and Acquisition in Shandong Province (202301001), Research and application of efficient cultivation technology for integrated tobacco and wheat production (2024371300260411).

X.W. and M.S.: conceptualization, methodology, data curation, and writing - original draft. L.T. and M.Y.: formal analysis and writing – review and editing. Q.G. and L.W.: software and investigation. H.Y. and L.Y.: visualization and supervision. X.H. and P.L.: validation and resources. L.Z.: project administration and funding acquisition. All authors have read and agreed to the published version of the manuscript.

Lei Tian and Lili Wang are employed by Shandong Linyi Tobacco Co., Ltd. Mingming Sun, Mingfeng Yang, and Qiang Gao are employed by China Tobacco Shandong Industrial Co., Ltd. The remaining authors declare that the research was conducted in the absence of any commercial or financial relationships that could be construed as a potential conflict of interest.

## AUTHOR AFFILIATIONS

[1]College of Plant Protection, Shandong Agricultural University, Tai'an, Shandong, China
[2]China Tobacco Shandong Industrial Co., Ltd., Qing'dao, China
[3]Shandong Linyi Tobacco Co., Ltd., Linyi, China

## AUTHOR ORCIDs

Li Zhang http://orcid.org/0000-0001-7725-9369

## AUTHOR CONTRIBUTIONS

Xiaoyu Wang, Conceptualization, Data curation, Methodology, Writing – original draft | Mingming Sun, Conceptualization, Data curation, Methodology, Writing – original draft | Lei Tian, formal analysis, writing - review and editing | Mingfeng Yang, formal analysis, writing - review and editing | Qiang Gao, Investigation, Software | Llli Wang, Investigation, Software | Honghao Yan, Supervision, Visualization | Long Yang, Supervision, Visualization | Xin Hou, Resources, Validation | Peng Liu, Resources, Validation | Li Zhang, Funding acquisition, Project administration

## DATA AVAILABILITY

The datasets generated during and/or analyzed during the current study are available from the corresponding author on reasonable request.

## ADDITIONAL FILES

The following material is available online.

## Supplemental Material

**Supplemental material (Spectrum02605-24-s0001.docx).** Fig. S1 to S5.

## Open Peer Review

**PEER REVIEW HISTORY (review-history.pdf).** An accounting of the reviewer comments and feedback.

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
