## [Reviewer comments · Microbiology Spectrum]

Microbiology Spectrum

Microbial Fertilizers Modulate Tobacco Growth and Development through Reshaping Soil Microbiome and Metabolome

Xiaoyu Wang, Mingming Sun, Lei Tian, Mingfeng Yang, Qiang Gao, Lili Wang, Honghao Yan, Long Yang, Xin Hou, Peng Liu, and Li Zhang

Corresponding Author(s): Li Zhang, Shandong Agricultural University

Review Timeline:

Submission Date:	October 20, 2024
Editorial Decision:	January 28, 2025
Revision Received:	April 1, 2025
Accepted:	April 12, 2025

Editor: Frédérique Reverchon

Reviewer(s): Disclosure of reviewer identity is with reference to reviewer comments included in decision letter(s). The following individuals involved in review of your submission have agreed to reveal their identity: Zichao Mao (Reviewer #1)

Transaction Report:

DOI: <https://doi.org/10.1128/spectrum.02605-24>

Re: Spectrum02605-24 (Microbial Fertilizers Modulate Tobacco Growth, Development, and Quality through Reshaping Soil Microbiome and Metabolome)

Dear Dr. Li Zhang:

Thank you for the privilege of reviewing your work. Below you will find my comments, instructions from the Spectrum editorial office, and the reviewer comments.

I have reviewed the manuscript myself as I was not able to secure a second reviewer for your article.

You will find below my comments and those from another independent reviewer. We both recommend a thorough revision of your manuscript.

My comments are based on the fact that the discussion and conclusions are not sustained by your results. In several parameters, no significant difference is found between the MF treatment and the OF treatment. Moreover, Kruskal-Wallis to assess differences in microbiome composition is not suitable so you would need to perform another statistical test. A PERMANOVA must complement your PCoA, and crucial Methods details (such as the bioinformatic analyses) should be provided.

Minor comments:

Abstract: a brief method overview should be added: field or greenhouse experiment? For how long did it last?

Introduction:

L51: "Therefore, to fundamentally improve the soil nutrient status, the incorporation of exogenous microorganisms emerges as a viable strategy for modulating the community structure of soil microbiota." This idea should be clearly introduced. The incorporation of exogenous microbes has been traditionally aimed at increasing the relative abundance of taxa with beneficial traits, such as N fixation or P solubilization. The idea that these microbial fertilizers could modulate the soil microbiota is more recent and it is still unclear whether this modulation 1) always occurs and 2) is always beneficial for the plant.

L57: "For example, *Sphingomonas* and *Bacillus paralicheniformis* are able to enrich beneficial bacteria in the soil (Wang, Wei et al. 2022, Xu, Qin et al. 2023), and *Azotobacter*, *Enterobacter*, and *Rhizobium* can significantly improve crop resistance to drought stress". Not clear whether these examples are aimed at showing how microbial inoculation influences the soil microbiome or at enlisting some of the benefits of microbial inoculation for the plant. Please rephrase.

L59: "Concurrently, *Bacillus*, *Pseudomonas*, *Acinetobacter*, *Serratia*, *Pantoea*, *Psychrobacter*, *Enterobacter*, and *Rahnella* are recognized as significant PGPB, with potential as exogenous microorganisms for incorporation into microbial formulations. (Singh, Sharma et al. 2023)." Some members of these bacterial genera are PGPB, but not all of them, they contain pathogenic species as well (e.g., *Pseudomonas syringae*, *Enterobacter cloacae*...). Please rephrase.

L66: what do you mean by "improved rhizosphere soil microecology". In terms of diversity, enrichment of beneficial taxa, desirable functions?

L67: scientific names in italics

L76: More examples should be given from the literature to introduce your hypotheses that microbial fertilisers will have a stronger positive effect on tobacco growth than other types of fertilisers.

Methods

L90: what about the description of seed germination?

L95-98: confusing. You mean that the differences between the fertilizer treatment and the other treatments will be attributed to N, right, as P and K fertilisers are applied in all treatments? But P and K are also present in the applied fertilizer. Could you please clarify?

L109: "Three representative tobacco plants of uniform growth were selected". How many plants to start with?

L125: Enzymatic assays: how many plants were selected? The three same individuals as mentioned before? Then you mixed the three samples, meaning you only have one composite sample for treatment? Is that correct?

There is no description of the bioinformatic pipeline. How did you perform sequence filtering, taxonomic affiliation, which database did you use, etc. The SRA project number where you deposited your data should be mentioned as well.

Results

Fig. 1: what is your n? For enzymatic activities, how did you perform statistical analysis with one composite sample per treatment? Did you mean "available" N, P etc, or what is "active" N?

L181: nutrient levels in the microbial fertilizer treatments are not significantly different from those of the organic fertilizer treatment.

Fig 2: please include the number of replicates and statistical tests performed in the figure caption.

Section 3.3. Please start by indicating the number of reads, and then the number of OTUs.

L217: which statistical analyses did you perform to detect these "significant differences" in the number of OTUs? In which treatment did you find the largest number of OTUs? Please explain.

Fig. 3: "Venn diagrams" instead of "Homogeneous maps"? This Figure could go to Supplementary Materials as you already present many figures.

L226: what does the "coverage index" indicate in terms of community?

L227-239: there is no significant differences in terms of alpha diversity metrics for bacterial nor fungal community, so stating that

there is a trend is misleading. This section should be rewritten accordingly.

Figure 6: a PERMANOVA should be applied to test for significant differences in beta-diversity.

L254: highly hypothetical (and should not be in the Results section). My take is that you have a large variation in the MF treatment and this hides possible differences between treatments, this is not an indicator of fungal community resilience.

L273: "Kruskal-Wallis H test with One-way ANOVA" may not be suitable for microbiome composition data. See for example: <https://www.tandfonline.com/doi/full/10.3402/mehd.v26.27663> and many other articles on the subject.

Fig. 10a: what is QC?

Discussion

In general, the discussion is not based on the actual results. For example, you mention that the MF treatment is superior to the other fertilisers in terms of soil nutrients, which is not true based on your statistics (L357-360). You mention that "Our findings (...) corroborating the outcomes of numerous prior studies on microbial fertilizers (Basak, Mandal et al. 2017, Ali, Liu et al. 2024). Only 2 cited studies is not "numerous". Also, you could cite other studies which did not find an effect on microbial inoculation on soil nutrients.

L361: hypothetical, you did not assess humus degradation.

L376: "MFs significantly optimized various agronomic traits of plants compared to NPK and OF". Be precise. Which traits?

Because not all measured parameters were enhanced by microbial inoculation.

L395: no, your differences in diversity metrics were not significant.

Paragraph L390-409 is not based on actual results. Also, you did not perform the proper statistical tests to assess differences in terms of taxa composition (see my comment above) or for PCoA. So this paragraph should be rewritten accordingly.

L410: these genera were not unique, they were present in all treatments.

The role of these genera cannot be "confirmed" by correlation analysis, be careful in your statements. You would need additional experiment to determine a causal effect between Rhodanobacter and the measured traits.

Why do you think that Bacillus was not significantly enriched in your MF treatment?

Revision Guidelines

Sincerely,
Frédérique Reverchon
Editor
Microbiology Spectrum

Reviewer #1 (Comments for the Author):

The integrated amplicon metagenomic strategy and metabolomic approach to conduct a comparative analysis among conventional chemical (NPK), organic (OF), and microbial fertilizers (MF) in this study. The results showed that application MF accelerated tobacco growth and development with change the metabolism of tobacco as well as the microbial community composition alterations in rhizosphere. The results is very interesting, however after carefully read the manuscript, I reviewed some comments:

Major comments

1. 40 days pot planting tobacco leaves is too earlier to evaluating the evaluation of the quality of the harvesting commercial leaves, it should more focus on the effects on metabolic changes as well as growth and development of tobacco
2. Based on the experimental designing, the NPK, OF and MF treatments did not remove the nutrients (NPK and other miner elements) difference
3. Microbial Fertilizer mainly containing *Bacillus subtilis* and *Bacillus licheniform*, it is should to be monitored the variations during the applying process, otherwise it will left question that whether the living organisms, their metabolite or even their culture medial components play roles to the changes of rhizosphere microbiota and growth and development of tobacco.
4. The microbiota results (Fig 7) demonstrated that in Phylum level Firmicutes, and Genus level *Bacillus* are top 4 abundant microorganisms, however it's hard to understand that the NPK treatment resulted in both Firmicutes and *Bacillus* had highest abundant microbiota even the MF with mainly containing *Bacillus subtilis* and *Bacillus licheniform* were applied.
5. In Fig 8 the applied MF (with mainly *Bacillus subtilis* and *Bacillus licheniform* dose of billion/g) with total 10g applications (Table 1 and 2) in pot planting tobacco, did not showed in the lists of significantly different microorganisms.

Miner comments

1. Langrage need to improve
2. Discussing should including the function of *Bacillus subtilis* and *Bacillus licheniform*, the discussing part also need including correlation between the significant bacteria and fungi.
3. Heatmaps of fig11 BDF can be merged together to show all those metabolites changed in all treatments(CK, NPK, OF and MF)

Microbial Fertilizers Modulate Tobacco Growth, Development, and Quality through Reshaping Soil Microbiome and Metabolome

Xiaoyu Wang^{1†}, Mingming Sun^{2†}, Lei Tian³, Mingfeng Yang², Qiang Gao², Lili Wang³, Honghao Yan¹, Long Yang¹, Xin Hou¹, Peng Liu¹, Li Zhang^{1*}

¹ College of Plant Protection, Shandong Agricultural University, Taian 271000, China; 18660071383@163.com(X.W.), 17842443663@163.com(H.Y.), lyang@sdau.edu.cn(L.Y.); houxin@sdau.edu.cn(X.H.); liupeng2003@sdau.edu.cn(P.L.); zhanglili@sdau.edu.cn(L.Z.);

² China Tobacco Shandong Industrial Co.,Ltd., Qing' dao 266300, China; sunmingming1994@126.com(M.S.), mingfeng618@163.com(M.Y.);

³ Shandong Linyi Tobacco Co., Ltd., Linyi 276826, China; tl_liushui@126.com(T.L.), lyysgq@139.com(Q.G.), 13954934881@163.com(L.W.);

* Correspondence: zhanglili@sdau.edu.cn(L.Z.);

† These authors contributed equally to this work.

12

Highlight:

1. Microbial Fertilizers (MF) Significantly Boost Abundance of Beneficial Bacterial Genera (*Rhodanobacter* and *Pseudolabrys*), Enhancing Soil Quality and Plant Growth and Development.

2. Compared to Traditional Fertilizers, MF Markedly Elevates Plant Metabolic Activity, Especially in Sugar, Lipid, and Amino Acid Production.

3. MF Drastically Enriches Key Aroma Components in Tobacco (e.g., Linalyl Acetate), Enhancing the Aromatic Quality of Tobacco Leaves.

Graphical Abstracts:

21

Abstract: To elucidate the mechanisms of microbial fertilizers in enhancing tobacco growth and quality, this study employed integrated microbiomics and metabolomics approaches to conduct a comparative analysis among conventional chemical, organic, and microbial fertilizers. The findings underscored the remarkable potential of microbial fertilizers in augmenting soil's quick-release

nutrient pool and bolstering soil enzymatic activity, surpassing both chemical and organic counterparts. The application of micro-
bial fertilizers accelerated tobacco growth and development significantly elevated agronomic indices, including plant stature, stem
girth, leaf extension, and the abundance of aromatic precursors, thereby facilitating a marked improvement in tobacco leaf quality.
Furthermore, the microbial community composition underwent pronounced alterations subsequent to the application of microbial
fertilizers, with the emergence of pivotal microorganisms such as *Rhodanobacter* and *Pseudolabrys* within the treatment group. These
microorganisms emerged as vital players in nutrient cycling processes, fostering plant growth, and mitigating the incidence of plant
diseases. Microbial fertilizers demonstrated a significantly superior capacity to stimulate metabolic vigor in tobacco plants com-
pared to other treatments, concomitant with a substantial enrichment of several metabolite, such as 3-Methylindole. These data
collectively imply that microbial fertilizers represent a more efficacious means of ameliorating soil physicochemical attributes,
thereby fostering superior tobacco growth, development, and quality enhancement.

**Keywords:** Tobacco; Multi-omics; Metabolism; Soil microorganism; Soil properties; Microbial fertilizers; Aroma pre-cursors

1. Introduction

Tobacco, as a significant economic crop, is extensively cultivated across the globe. The moderate application of
chemical fertilizers has been demonstrated to promote the growth and development of tobacco, thus increasing the
yield of tobacco leaves (Rashidzadeh, Olad et al. 2015). Nonetheless, in recent times, tobacco farmers frequently
over-apply chemical fertilizers in an attempt to maximize production, a practice that has not only given rise to
profound soil health concerns but also adversely affected the normal growth and development of tobacco plants
(Chakraborty, Tiwari et al. 2017, Mao, Lu et al. 2017, El Khattabi, Louche et al. 2018). Consequently, the adoption of
environmentally friendly novel materials as alternatives to traditional chemical fertilizers is imperative for further
augmenting tobacco leaf yield and improving its quality.

Soil microorganisms exhibit a profound correlation with soil quality and its physicochemical properties, which
exert direct influences on crop yield and quality (Liu, Liu et al. 2023). A well-balanced soil microbial community
structure facilitates the harmonious functioning of soil nutrient cycling, the decomposition of soil organic matter, the
biodegradation of organic pollutants, and the enhancement of plant resilience to abiotic stress, these processes
collectively contribute to the promotion of crop growth and development (Chaudhary, Sindhu et al. 2023, Li, Wang et
al. 2023). Therefore, to fundamentally improve the soil nutrient status, the incorporation of exogenous
microorganisms emerges as a viable strategy for modulating the community structure of soil microbiota (Lai, Duan et
al. 2024). These preparations are typically formulated on the basis of beneficial microorganisms ubiquitous in soil,
namely plant growth-promoting bacteria (PGPB). PGPB establish unique symbiotic relationships with plant roots,
enhancing soil nutrient availability through processes such as nitrogen fixation, phosphate solubilization, and
potassium release, they secrete or induce the production and release of phytohormones, thereby directly stimulating
plant growth (Lai, Duan et al. 2024, Yan, Liu et al. 2024). For example, *Sphingomonas* and *Bacillus paralicheniformis* are
able to enrich beneficial bacteria in the soil (Wang, Wei et al. 2022, Xu, Qin et al. 2023), and *Azotobacter*, *Enterobacter*,
and *Rhizobium* can significantly improve crop resistance to drought stress. Concurrently, *Bacillus*, *Pseudomonas*,
*Acinetobacter*, *Serratia*, *Pantoea*, *Psychrobacter*, *Enterobacter*, and *Rahnella* are recognized as significant PGPB , with
potential as exogenous microorganisms for incorporation into microbial formulations. (Singh, Sharma et al. 2023).

Furthermore, microbial fertilizers have demonstrated favorable outcomes in practical applications. Specifically,
plant growth-promoting rhizobacterial (PGPR) microbial fertilizers have altered the soil microbial composition,
markedly enhanced the root vigor of tobacco plants, contributed to a comprehensive enhancement of tobacco growth
parameters. (Shang, Fu et al. 2023). Compared to woody peat, microbial fertilizers significantly improved tobacco
rhizosphere soil microecology and increased soil fast-acting potassium and organic matter content (Yu, Zhang et al.
2023). Application of microbial fertilizers containing *Phanerochaete chrysosporium* and *Bacillus thuringiensis* adjusts

soil nitrogen supply and leaf uptake, significantly reduces nicotine content of tobacco and improves tobacco quality
(Shang, Chen et al. 2017). Extending these findings, microbial fertilizers have also been successfully employed in
cereal crop cultivation, where they promoted wheat and rice growth, optimized nutrient utilization efficiency, and
improved overall soil health (Tahir, Khalid et al. 2018, Naher, Biswas et al. 2021, Ma, He et al. 2022).

In recent years, there has been a surge in research examining the impacts of various fertilizers on the microbial
composition of tobacco rhizosphere soils, and numerous studies have consecutively reported the growth-promoting
mechanisms of diverse fertilizers on tobacco plants (Yan, Zhao et al. 2020, Shang, Fu et al. 2023, Wang, Xiao et al. 2023).
However, despite these advancements, the existing body of literature remains inadequate in conclusively
demonstrating the superiority of microbial fertilizers over traditional organic and inorganic fertilizers in tobacco
cultivation. Consequently, the current investigation was focused on conducting a comparative analysis of tobacco
plants treated with conventional chemical fertilizers, organic fertilizers, and microbial fertilizers, utilizing an
integrative approach combining microbiomics and metabolomics techniques. Consequently, the present study aimed,
therefore: (i) to demonstrate the significant advantages of microbial fertilizers over conventional fertilizers in
enhancing both the yield and quality of tobacco, and (ii) to elucidate in detail the mechanisms by which microbial
fertilizers influence tobacco quality through the modulation of metabolic activities.

2. Materials and Methods

2.1. Experimental materials and water and fertilizer management

Tobacco plants used for potting experiments were placed in the tobacco greenhouse of the Dai Zong Campus of
Shandong Agricultural University (36.17N, 117.16E).

The experimental tobacco variety for potting was NC55. To prevent the occurrence of soil-borne diseases in
continuous cropping, the potting soil was obtained from a large field that had not been planted with tobacco and was
a sandy loam with the following basic physical and chemical properties: organic matter 2.15%, alkaline dissolved
nitrogen 52.18 mg/kg, effective phosphorus 23.41 mg/kg, quick-acting potassium 121.98 mg/kg, pH 6.00.

When seedlings grow to 3-5 true leaves, select seedlings with consistent growth, transplant them into
24cm*25.5cm pots with 5kg of soil, make sure each pot has the same growing conditions, and keep the same
management with the local farmland. The experimental protocol was as follows: treatment were CK (no fertilizer),
NPK (chemical fertilizer), OF (organic fertilizer), and MF (microbial fertilizer), the composition of fertilizers in each
group is shown in Table 1. The conventional recommended field fertilizer rate was 900 kg/hm² and potting rates were
converted based on the field rates. To ensure that the same total N input was obtained for each treatment, a N:P:K
(1:1.5:3) application ratio, i.e., the same as that used in local on-farm crop management, was applied between the
control and treatment groups to ensure that any differences observed between these groups could not be attributed to
these nutrients. Differences in application rates, Potassium sulphate and calcium superphosphate were used to balance
the nutrient levels and the detailed fertilizer rates are shown in Table 2.

Use 70% of the total fertilizer application as a base fertilizer and mix it well with the soil before transplanting,
and use the remaining 30% as a follow-up fertilizer by sprinkling the granules on the soil surface 7 days after
transplanting (after the seedling resting period). In the first 7 days after transplanting, 400mL of water was evenly
poured into each pot every day, during which the state of the seedlings was observed, and water was replenished
promptly if wilting occurred; after 7 days, 400mL of water was evenly poured into each pot every other day.

**Table 1.** Fertilizers used in the experiment

Type of fertilizer	Formulations	Technical indicators
--------------	----------------------

[revised manuscript text omitted]

*2.5. Soil DNA extraction and high-throughput sequencing*

Three rhizosphere soil samples were selected for each treatment, for a total of 12 samples. The collected soil
samples were extracted from DNA by Shanghai Meiji Bio-medical Technology Co. Ltd. and the extracted DNA was
detected using 1% agarose gel electrophoresis assay, which showed clear bands and good-quality DNA. 16S rRNA
universal primers 338F (5' -ACTCCTACGGGAGGCAGCAG-3') and 806R (5' -
GGAC-TACHVGGGTWTCTAAT-3') for 16S rRNA gene in bacteria, and ITS1F(5' -
-CTTGGTCATTTAGAGGAAGTAA-3') and ITS2R (5' -GCTGCGTTCATCGATGC-3') in fungi. The PCR
products of the same sample were mixed and detected by 2% agarose gel electrophoresis, and the PCR products were
cut and recovered by using the AxyPreDNA Gel Recovery Kit (AXYGEN), and the PCR products were detected and
quantified by using the QuanyiFluorTM-ST Blue Fluorescent Quantification System (Promega), taking into account
the results of the preliminary quantification by electro-phoresis. Final fungal and bacterial amplicon sequencing was
performed using the Illumina MiSeq system.

*2.6. Metabolom analysis*

The LC-MS/MS analysis of sample was conducted on a Thermo UHPLC-Q Exac-tive HF-X system equipped with
an ACQUITY HSS T3 column (100 mm × 2.1 mm i.d., 1.8 μm; Waters, USA) at Majorbio Bio-Pharm Technology Co.
Ltd. (Shanghai, China). The mobile phases consisted of 0.1% formic acid in water:acetonitrile (95:5, v/v) (sol-vent A)
and 0.1% formic acid in acetonitrile:isopropanol:water (47.5:47.5, v/v) (solvent B). MS conditions: The flow rate was
0.40 mL/min and the column temperature was 40°C. The injection volume was 3 μL. The mass spectrometric data
were collected using a Thermo UHPLC-Q Exactive HF-X Mass Spectrometer equipped with an electrospray ionization
(ESI) source operating in positive mode and negative mode. The optimal conditions were set as followed: Aux gas

heating temperature at 425°C; Capillary temp at 325°C; sheath gas flow rate at 50 psi; Aux gas flow rate at 13 psi;
ionspray voltage floating (ISVF) at -3500V in negative mode and 3500V in positive mode, respectively; Normalized
collision energy , 20-40-60 eV rolling for MS/MS. Full MS resolution was 60000, and MS/MS resolution was 7500. Data
acquisition was performed with the Data Dependent Acquisition (DDA) mode. The detection was carried out over a
mass range of 70-1050 m/z.

2.7. Data processing

Soil physicochemical properties and soil enzyme activities were analyzed by one-way ANOVA using SPSS 25.0
software at $P \leq 0.05$. Biological statistical analysis of 97% similarity OTU clustering using Uparse (version 7.0.1090).
Alpha diversity was calculated using Mothur (version v.1.30.2), including Chao 1, Shanno, Simpson, ACE, and
Coverage. Beta analyzed differences in microbial community composition between treatments based on PCoA with
Bray-Curtis distance and principal coordinates were analyzed using R (version 3.3.1). Species differences were
analyzed using the Kruskal-Wallis rank sum test (Kruskal-Wallis h test). Correlations between soil microorganisms
and soil environmental factors were analyzed using redundancy analysis (RDA). RDA was performed using the R
vegan package (version 2.4.3).

3. Results

3.1. Effect of fertilization on soil chemical properties and enzyme activities

The investigation into the impact of various treatments on soil chemical composition revealed pronounced effects,
as evidenced by the substantial enhancement in the concentrations of readily available nitrogen, phosphorus, potas-
sium, and soil organic matter in the NPK, OF, and MF treatments (Figure 1), relative to the control group. Notably,
within the treatment group, the MF treatment exhibited notably elevated levels of readily available nitrogen and po-
tassium, surpassing those observed in NPK and OF. Conversely, the MF treatment manifested a marginally lower
concentration of readily available phosphorus, albeit this disparity was statistically insignificant, with both MF and OF
significantly outperforming NPK. Regarding soil organic matter content, MF and OF treatments did not significantly
differ from each other but both significantly surpassed the NPK treatment. Furthermore, the MF treatment significantly
elevated soil pH, aligning it closer to neutrality compared to the other treatment groups.

Soil enzyme activity, a pivotal metric for assessing soil fertility, was examined in this study, encompassing the
activities of catalase, urease, and sucrase across the control and treatment groups (Figure 1). The findings underscored
a marked elevation in enzyme activities for all treatment groups in comparison to the control, indicative of enhanced
soil biological processes. Specifically, in terms of urease and sucrase activities, the observed differences among the
treatment groups were statistically significant and demonstrated a discernible hierarchical pattern, with MF > OF >
NPK. Similarly, for catalase activity, while the trend of MF > OF > NPK persisted, the distinctions among the treatment
groups failed to reach statistical significance.

 Figure 1. Effects of different fertilization treatments on soil physicochemical properties and enzyme activities:
 (A)Active Nitrogen, (B)Active Phosphorus, (C)Active potassium, (D)Organic Matter, (E)Soil pH, (F)Catalase,
 (G)Urease, (H)Sucrose synthase.

**3.2. Effect of fertilization on plant agronomic traits and aroma precursors**

The agronomic traits of plants, which serve as direct indicators of their growth and developmental status, were
 comprehensively evaluated in this experiment by measuring height, stem circumference, and leaf length and width
 across various growth stages for each group. The outcomes revealed a remarkable superiority in the agronomic indices
 of the treatment groups over the control group. Among the treatment cohorts, a consistent trend of MF > OF > NPK was
 discernible, with similar patterns observed at every measurement point. Notably, the plant growth profiles of MF and
 OF were closely aligned, particularly in terms of height and leaf length, exhibiting minimal significant differences.

Additionally, the content of aroma precursors, a crucial quality parameter in tobacco, was examined by quanti-
 fying chlorophyll a, chlorophyll b, total chlorophyll, lutein, and starch levels in samples from distinct groups. Com-
 pared to the control (CK), a marked increase in aroma precursor content was observed in the treatment groups. Spe-
 cifically, in chlorophyll and lutein content, a distinct hierarchy emerged, with OF > MF > NPK, and these differences
 among treatments were statistically significant. Conversely, in starch content, a similar trend was observed, albeit with
 MF > OF > NPK, where the discrepancy between MF and OF was relatively minor.

 Figure 2. Effects of different fertilization treatments on plant agronomic traits and aroma precursors: (A)Height,
 (B)Stem circumference, (C)Leaf length, (D)Leaf width, (E)Chlorophyll a, (F)Chlorophyll b, (G)Total Chlorophyll,
 (H)Lutein, (I)Starch.

3.3. Changes in rhizospheric microbial diversity of tobacco

The Venn diagram shows, based on 97.0% similarity clustering, the number of OTUs that reflect the commonality
 between microbial populations in soil. Significant differences in the number of rhizospheric OTUs under different fer-
 tilization treatments. In the soil bacterial community, there were a total of 2214 OTUs in the different fertilization
 treatments, of which 843, 630, 582, and 677 OTUs were specific to the CK, NPK, OF, and MF treatments, respectively
 (Figure. 3A). In the fungal community, a total of 296 OTUs were obtained, of which 172, 106, 129, and 175 were specific
 to each treatment, respectively (Figure 3B).

Figure 3. Homogeneous maps of bacterial (A) and fungal (B) based on OTU levels.

3.4. Characterization of microbial communities

Alpha diversity reflects the abundance and diversity of the samples, mainly the Shannon index, Simpson index,
 Chao index Ace index, and coverage index. Simpson index and Shannon index mainly respond to the diversity of the
 soil community, and Chao, ACE, and coverage respond to the abundance of the community. The results of α -diversity
 analyses showed that the Shannon index of soil bacterial community showed an increasing trend after MF treatment
 compared to the control, while the NPK treatment showed a decreasing trend compared to the control (Figure 4A). The
 Simpsons index was significantly lower in MF treatment compared to CK and increased in NPK and OF treatments
 (Figure 4B). We further evaluated the Chao and Ace indices and showed that they were significantly lower in all ferti-
 lization treatments, with the MF treatment being close to CK in terms of community richness (Figure 4C, E). Finally, we
 assessed the microbial coverage of each sample and found that the coverage values of all samples were close to 1, in-
 dicating the highest level of species coverage of the soil samples (Figure 4D). For fungi, Shannon index was signifi-
 cantly higher in the MF treatment group and significantly lower in the other treatments as compared with the CK
 treatment (Figure 5A), while Simpson index was reduced to some extent. Chao index and Ace index were somewhat
 improved and close to the CK control (Figure 5C, E); the final assessment revealed a significant increase in the coverage
 index and some improvement in the other treatments (Figure 5D). In summary, the application of microbial increased
 soil bacterial and fungal diversity and abundance to some extent.

 Figure 4. The effects of different fertilization treatments on the α -diversity of rhizospheric soil bacterial communities:
 (A)Shannon index, (B)Simpson index, (C)Chao index, (D)Coverage, (E)ACE index.

Figure 5. The effects of different fertilization treatments on the α -diversity of rhizospheric soil fungal communities:
(A)Shannon index, (B)Simpson index, (C)Chao index, (D)Coverage, (E)ACE index.

PCA analyses based on Bray-Curtis variation reflected the diversity of soil bacterial and fungal β between treat-
ment. In the context of bacterial community architecture, a distinct segregation was evident among samples belonging
to disparate treatment cohorts (Figure 6A). Each treatment group exhibited unique signatures, whereas the β -diversity
among these groups manifested a greater degree of convergence when compared to the control cohort. Turning our
attention to the fungal community composition, a pronounced segregation was likewise observed between the treat-
ment and control groups (Figure 6B). This finding underscores the potency of different fertilizer applications in sub-
stantially altering the soil microbial makeup and fostering a refined fungal community structure. Notably, the fungal
community exhibited a notably lesser degree of perturbation subsequent to fertilizer application, relative to the bacte-
rial community. This observation implies a heightened stability and resilience within the fungal community structure,
rendering it less prone to modifications compared to its bacterial counterpart.

Figure 6. Beta diversity analysis of rhizospheric soil bacterial (A) and fungal (B) communities based on OTU levels.

3.5. Effect of fertilization on the structural composition of microbial communities

The microbial community structure of each group was analyzed by 16S and ITS amplicon sequencing, and the
bacterial and fungal phyla with the top 10 relative abundances, and the bacterial and fungal genera with the top 15
relative abundances were selected for subsequent analysis (Figure 7). For bacterial phylum level species abundance
results showed that the dominant phyla identified in the four treatments were Actinobacteria, Proteobacteria, Chlor-
oflexi, Firmicutes, and Bacteroidota (Figure 7A). We further examined the abundance of species at the genus level of the
bacterial community, and the dominant genera was Kribblella, Streptomyces, Lysobacter, Bacillus, and Devosia (Figure
7B).

For fungal phylum level species abundance, it was found that Ascomycota was the most abundant fungal com-
munity followed by Basidiomycota and Mortierellomycota (Figure 7C). At the genus level, the most abundant fungi
genus is Arthrotrrys followed Unclassified_p_Ascomycota, Myceliophthora, Unclassified_f_stephanosporaceae,
Chaetomiun (Figure 7D).

Figure 7. Analysis of soil microbial composition. Bacteria: (A) phylum level, (B) genus level; Fungi: (C) phylum level,
 (D) genus level.

The test of significant difference between groups was conducted using Kruskal-Wallis H test with One-way
 ANOVA to assess the level of significance of the differences in species abundance between different groups of micro-
 bial communities to obtain significant differential species between groups (Figure 8). The results showed that the dif-
 ferential bacterial genera present in the MF group were Rhodanobacter, Pseudolabrys, Gemmatimonas, and Ter-
 rabacter, and the differential fungal genus was Microascus. In addition to this, CK possessed most of the differential
 genera that showed significant differences with the samples from the other three treatment groups, further indicating
 that the application of fertilizers significantly altered the microbial community structure of the soil.

Figure 8. Significantly different microorganisms between the control and treatment groups: (A)bacteria, (B)fungi.

3.6. Association between microorganisms and soil properties and plant traits

In order to elucidate the pivotal role of prevalent bacterial groups in modulating plant growth and development, a
comprehensive analysis was undertaken correlating the dominant bacterial genera within the Top15 category with
various soil properties and plant phenotypic traits. This analysis yielded a correlation heatmap (Figure 9), which pro-
vided insights into the intricate interrelationships.

The findings revealed a discernible pattern, wherein five dominant bacterial genera exhibited significant and
positive correlations with soil attributes. Notably, Rhodanobacter demonstrated a robust association with all soil in-
dicators except for soil organic matter (SOM), whereas Bacillus was significantly linked to SOM. Psedolabrys, Sphin-
gomonas, and Arthrobacter showed significant positive correlations with soil sucrose synthase (S-SS), available potas-

sium (AK), and soil pH respectively. Conversely, four bacterial genera manifested significant negative correlations
 with soil properties. Pseudonocardia stood out, displaying a pronounced negative relationship with ammonium ni-
 trogen (AN), available phosphorus (AP), AK, and pH. This was followed by Streptomyces, unclassi-
 fied_f_Caulobacteraceae, and Kribbella, each exhibiting negative associations with varying degrees of significance,
 particularly with AP, AK, S-UE and soil pH.

Regarding plant traits, four dominant bacterial genera exhibited significant positive correlations. Rhodanobacter
 stood out with a strong positive correlation across all traits, followed by Arthrobacter, Sphingomonas, and
 Pseudotabrys, which displayed significant positive relationships with the majority of plant indicators. In contrast, four
 bacterial genera, including Pseudonocardia, demonstrated significant negative correlations with plant traits. Specifi-
 cally, Pseudonocardia had a notable negative impact on plant height, stem circumference, leaf dimensions (length and
 width), chlorophyll content (a, b, and total), and starch levels. Streptomyces and unclassified_f_Caulobacteraceae also
 showed negative correlations, their significance was confined to select plant metrics.

A notable observation was that, while the majority of the dominant fungal genera exhibited positive correlations
 with all indicators, these associations failed to reach statistical significance. In contrast, a distinct pattern emerged with
 unclassified_f_Chaetomiaceae, which displayed a significant negative correlation with the majority of the investigated
 metrics. Furthermore, unclassified_k_Fungi also manifested a negative correlation with most indicators, particularly
 evincing a statistically significant negative relationship with chlorophyll b, a crucial photosynthetic pigment. Thus,
 compared to fungi, the genus bacteria is more closely related to the growth and development process of plants.

 Figure 9. Association between microorganisms and soil chemical properties and enzyme activities: (A)bacteria,
 (B)fungi; association between microorganisms and plant agronomic traits and aroma precursors: (C)bacteria, (D)fungi.

*3.7. Effect of different fertilization treatments on metabolites of tobacco leaves*

All small molecule metabolites in the samples were detected and analyzed using LC-MS analysis, and a total of
 404 plant secondary metabolites were identified, including 198 Terpenoids, 56 Phenolic acids, 36 Steroids, 33 Organic

acids, 22 Flavonoids, 18 Coumarins, 14 Alkaloids, 12 Indole, 5 Quinones, 4 Lignans, 4 Stilbenes and 2 Tannins. Partial
 Least Squares Discriminant Analysis (PLS-DA) was performed on the intergroup samples to compare the metabolite
 profiles between different groups (Figure 10A). The experiment proved that the metabolite separation was obvious
 between the treated group and the control group, and the metabolite separation was obvious between MF and OF and
 NPK. In order to further demonstrate the effect of MF on plant metabolic processes, the changes of tobacco metabolites
 among different groups were analyzed by differential volcano mapping (Figure 10). 323 differential metabolites were
 identified in MFvsCK (up=306, down=17), 328 differential metabolites in MFvsNPK (up=250, down=78), and 151 dif-
 ferential metabolites in MFvsOF (up=180, down=43).

Figure 10. (A) PLS-DA on metabolites between different groups; Volcano plot of differential metabolites between
 groups: (B) MFvsCK, (C) MFvsNPK, (D) MFvsOF.

The utilization of the KEGG database facilitated the enhancement and identification of critical metabolic pathways
 within the observed differential metabolome (Figure 10). A two-by-two comparison with CK, NP, OF, using MF as a
 baseline, showed that the metabolic pathways significantly enriched in MFvsCK were: "linoleic acid metabolism",
 "alpha-Linolenic acid metabolism", "plant hormone signal transduction", "arachidonic acid metabolism", "arginine
 biosynthe". The metabolic pathways that were significantly enriched in MFvsNPK were: "alpha-Linolenic acid me-
 tabolism", "plant hormone signal transduction", "linoleic acid metabolism", "arachidonic acid metabolism", "cutin,
 suberine and wax biosynthesis". The metabolic pathways significantly enriched in MFvsOF were: "plant hormone
 signal transduction", "phenylalanine metabolism", "alpha-Linolenic acid metabolism", "Linoleic acid metabolism".

Given the predominantly convergent nature of the enriched metabolic routes across the various groups, a deeper
 interrogation into the intergroup variations in metabolic dynamics was pursued. We systematically identified the top

20 metabolites exhibiting statistically significant differences across each comparative group, subsequently subjecting
 these to cluster analysis. The results showed that the 18 up-regulated metabolites in MFvsCK included 6 lipids, 3 ter-
 penes, 1 phenolic acid, 1 coumarin, 1 carbohydrate, 1 alkaloid, 5 others, and the 2 down-regulated metabolic species
 included 1 carbohydrate, 1 lipid; the 8 up-regulated metabolites in MFvsNPK included 2 terpenes, 2 lipids, 1 indole,
 3 others, 12 down-regulated metabolites including 4 lipids, 2 carbohydrates, 1 coumarin, 1 steroid, 1 phenolic acid,
 3 others; 8 up-regulated metabolites in MFvsOF including 2 carbohydrates, 2 indoles, 1 amino acid, 1 phenolic acid,
 2 others, and 12 down-regulated metabolites including 4 carbohydrates, 2 steroids, 2 coumarins, 1 phenolic acid, 1 lipid,
 2 others.

Figure 11. KEGG metabolic pathway enrichment between groups (A)MFvsCK, (C)MFvsNPK, (E)MFvsOF, the hori-
zontal coordinates are the p-value of the significance of the enrichment, and the smaller p-value is statistically signifi-
cant, and a p-value of less than 0.05 is generally considered to be a function of a significantly enriched term; vertical
coordinate is KEGG pathway. The size of the bubbles in the graph represents how much of the pathway is enriched to
the metabolic set of compounds; Heatmap of clustering of up- and down-regulated differential metabolites between
groups: (B)MFvsCK, (D)MFvsNPK, (E)MFvsOF.

4. Discussion

4.1. Effect of Fertilizer Application on Soil Properties and Plant trait

Soil quick-release nutrients serve as pivotal indicators for evaluating soil fertility, underpinning agri-
cultural productivity. In the present investigation, we conducted a comparative analysis of microbial fertiliz-
ers (MF), conventional chemical fertilizers (NPK), and organic fertilizers (OF) on tobacco crop growth, with
a focus on their impact on soil nutrient dynamics. Our findings revealed that MFs exhibited a superior ca-
pacity in augmenting soil quick-release nutrient levels compared to the other two fertilizer types, corrobo-
rating the outcomes of numerous prior studies on microbial fertilizers (Basak, Mandal et al. 2017, Ali, Liu et
al. 2024). The application of MF significantly ameliorated soil acidity, a phenomenon attributed to the in-
tensified microbial activity in decomposing soil organic matter, this process accelerated the degradation of
humus and other acidic compounds, thereby modulating soil pH towards more favorable conditions for plant
growth (Hu, Liu et al. 2018, Wang, Cheng et al. 2022). Soil enzyme activities, which are instrumental in
regulating soil fertility and fostering plant growth and development, were also examined. Aligns with pre-
vious research, the application of MFs led to a marked enhancement in the activities of key enzymes, in-
cluding sucrase, urease, and catalase, this observation confirming that the beneficial microorganisms inher-
ent in MF actively participate in the decomposition of soil organic matter, secreting a diverse array of en-
zymes during their proliferation. These enzymes contribute to an overall elevation in soil enzyme levels, fa-
cilitating nutrient transformations such as nitrogen and phosphorus fixation, thereby enhancing soil nutrient
availability and promoting plant health (Zhao, Ni et al. 2016, Ren, Liu et al. 2021, He, Zhang et al. 2024).

The assessment of agronomic traits in plants is paramount for quantifying their growth and develop-
mental progress, as evidenced by extensive literature (Ambati, Phuke et al. 2020, Varshney and Majee 2022,
Chandra, Jaiswal et al. 2023). Prior research has underscored the remarkable potential of microbial fertiliz-
ers (MF) in enhancing plant height, spike length, and yield in wheat (Yang, Gong et al. 2020), as well as leaf
length and width in tobacco (Zhai, Hu et al. 2020). Our current study aligns with these findings, demon-
strating that MFs significantly optimized various agronomic traits of plants compared to NPK and OF. This
superiority can be attributed to MF' more effective enhancement of the soil nutrient profile, thereby aug-
menting nutrient availability to plants. Chlorophyll, a vital aroma precursor in tobacco, undergoes degrada-
tion during tobacco processing to yield crucial aroma compounds like neophytadiene and phytofurane. Attia
et al. (Attia, El-Sayyad et al. 2020) have reported that plant growth-promoting rhizobacteria can significant-
ly elevate total chlorophyll content and the specific fractions of chlorophyll a and b in tomato. Our experi-
mental results echo this observation, revealing a notable increase in chlorophyll content in tobacco plants
fertilized with MFs compared to NPKs and OFs. This suggests that MF can effectively enhance the quality
of tobacco leaves by boosting chlorophyll levels. Furthermore, the presence of certain plant
growth-promoting (PGP) microorganisms within MF has been shown to mitigate both biotic and abiotic
stresses, thereby fortifying plant resilience. This mechanism may represent a pivotal avenue through which
MFs exert their beneficial effects, as supported by existing literature (Liu, Zhong et al. 2020, Gamit and
Amaresan 2023).

4.2. Microorganisms in the soil

The diversity and composition of soil microbial communities hold paramount importance in sustaining
the health and functional integrity of soil ecosystems, as emphasized in previous studies (Babin, Vogel et al.
2014, Moreira-Grez, Muñoz-Rojas et al. 2019, Chen, Wang et al. 2022). Shan et al. (Shan, Wei et al. 2023)
have highlighted the contribution of microbial inoculation to enhancing soil microbial diversity and foster-
ing the reorganization of microbial community structures. Our current experimental findings concur with
these observations, revealing that the application of microbial fertilizers (MFs) significantly augmented the
diversity of microbial communities. This enhancement can be attributed to the improved nutrient availability
conferred by MFs, which fosters the proliferation and swift colonization of beneficial bacteria within the soil
matrix (Liang, Hou et al. 2020, Yun, Yan et al. 2021, Zhang, Wu et al. 2023). Despite the application of
varying fertilizer types, the species composition of major soil microbial genera remained unaltered; howev-
er, their abundance was notably influenced. Specifically, *Kribbellla*, *Streptomyces*, *Lysobacter*, *Bacillus*, and
*Devosia* emerged as the dominant bacterial genera, whereas *Arthrobotrys*, *Myceliophthora*, and *Chaetomium*
constituted the preeminent fungal genera in the soil. To further elucidate the effects of different fertilizers on
soil microbial communities, principal coordinate analysis (PCoA) was employed. In the context of soil bac-
terial communities, PCoA revealed a pronounced segregation between MF and both NPK, OF, and CK, in-
dicating that MF significantly refined the soil bacterial community structure and impacted the reorganization
of microbial community dynamics. Conversely, PCoA of soil fungi demonstrated some overlap among the
three fertilizer treatments, potentially reflecting the greater resilience of fungal communities to changes in
fertilizer type compared to bacterial communities, as suggested by previous research (Gong, Liu et al. 2019,
Mayer, Csorbainé et al. 2021, Zhang, Li et al. 2022).

The application of microbial fertilizers (MF) introduced two unique bacterial genera, *Rhodanobacter*
and *Pseudolabrys*, into the soil ecosystem. *Rhodanobacter* is recognized as a primary contributor to catalase
and oxidase enzymes in soil (Madhaiyan, Poonguzhali et al. 2014), and the strong positive correlation ob-
served between *Rhodanobacter* abundance and soil enzyme activities, plant agronomic traits, and aroma
precursor indices in this study underscores this functional role. This finding aligns with previous research
demonstrating a significant positive correlation between *Rhodanobacter* abundance and Chinese cabbage
yield, further corroborating its role in enhancing multiple agronomic traits. *Rhodanobacter*, as a plant
growth-promoting rhizobacterium, facilitates plant growth and development by enhancing nutrient mobiliza-
tion, promoting phytohormone production, and bolstering the plant root system's resistance to soil pathogens
(Xu, Lv et al. 2023). Concurrently, *Pseudolabrys*, a nitrogen-fixing bacterium prevalent in soil, exhibited a
notable positive correlation with various rapidly available nutrients in the soil. This can be attributed to
*Pseudolabrys*' ability to associate with plant roots, forming rhizomes that contribute to nitrogen fixation,
thereby enriching the soil with essential nutrients (Lee, Oh et al. 2020, Zheng, Wang et al. 2024). Mean-
while, the significant correlation observed between *Pseudolabrys* and the content of various aroma precur-
sors, particularly chlorophyll, underscores the pivotal role of *Pseudolabrys* in enhancing the quality of to-
bacco leaves.

4.3. Effect of different fertilizers on plant metabolism

The PLS-DA analysis indicated that fertilizer application, particularly MF, significantly modulated
plant metabolic activities. Further comparative analysis using volcano plots revealed a preponderance of
up-regulated metabolites in MF-treated samples compared to both control and other treatment groups. The
magnitude of up-regulation was substantially greater than that of down-regulation, suggesting that MF

application significantly boosts plant metabolic levels. Differential metabolite enrichment analysis revealed
certain convergences among enriched metabolic pathways across treatment groups (Figure 9): were enriched
in “linoleic acid metabolism”, “alpha-Linolenic acid metabolism” and “plant hormone signal transduction”,
likely stemming from the similar impacts of different fertilizer treatments on plant basal metabolic processes
(Xu, Chu et al. 2020, Fatima, Fatima et al. 2024). In our analysis of the top 20 differential metabolite
clusters derived from the comparisons of MFvsOF and MFvsNPK, we identified nine overlapping
metabolites of particular interest. Notably, 3-Methylindole(skatole) emerged as a significantly up-regulated
metabolite in both comparative groups, echoing its potential positive contribution to flavor quality, as
suggested by previous studies (Farmer, McConnell et al. 1995, Zhu, Niu et al. 2021). Specifically, moderate
concentrations of skatole impart rich fruity and floral nuances to tea aroma, underscoring its significance as
a key aroma source.

Conversely, we observed a notable down-regulation of two glucosides , namely (S)-Mandelic acid
O-beta-D-Glucopyranoside and Linalool oxide D 3-[apiosyl-(1->6)-glucoside], in both MFvsOF and
MFvsNPK comparisons. This finding is consistent with the recognition that degradation of volatile
metabolites of glycosides bound to fatty acids, amino acids, and other substances can produce aromatic
component(Chen, Zhu et al. 2022). Furthermore, within the MFvsOF comparison, we detected a significant
up-regulation of the oligosaccharide Vomifoliol 9-[glucosyl-(1->4)-xylosyl-(1->6)-glucoside], indicative of
MF's pronounced effect on carbohydrate metabolic pathways and its potential to foster the production of
aroma-contributing substances in tobacco. Our study also revealed a marked enhancement in the content of
nicotine and its primary metabolite cotinine following the application of microbial fungal fertilizers. While
these compounds evoke controversy due to their psychoactive properties, they have been shown to bolster
tobacco's resistance mechanisms against pests and diseases during its initial and intermediate growth stages
(Lee, Joo et al. 2021).

Furthermore, our investigation revealed that in MFvsNPK, MF demonstrated a markedly decreased lipid
content. This observation may stem from the microbial metabolism of lipids into an array of esters within
tobacco leaves (Giménez-Bañón, Moreno-Olivares et al. 2022), this metabolic transformation enhances the
tobacco's bouquet with floral and fruity nuances (Liu, Gong et al. 2023). Additionally, we observed
significant upregulation in the concentrations of several crucial compounds in MFvsNPK. Notably, Linalyl
acetate, a pivotal terpene and primary aroma constituent in lavender essential oils, was elevated. This
compound is renowned for imparting a rich, sweet aroma (Xiao, Li et al. 2017). Another notable compound,
Hydroxypelenolide, a sesquiterpene lactone exhibiting vital biological activities, including antiviral,
anti-inflammatory, and anti-cytotoxic properties, was also found to be upregulated. Moreover, studies have
underscored its potent antifeedant effects on insects (Marina, Ana et al. 2017, Fraga, Díaz et al. 2021). In
MFvsOF, our analysis further indicated a pronounced increase in indole derivatives, particularly Tryptamine.
As a biogenic amine, Tryptamine is widely acknowledged for its contribution to plant growth and
development processes, it is also capable of bolstering tobacco plants' resilience against pests and diseases
(Gill, Ellis et al. 2003).

5. Conclusions

The present study has demonstrated that microbial fertilizers, through their ability to modify soil
microbial community structure and tobacco leaf metabolic pathways, effectively promote tobacco growth,
development, and quality enhancement, surpassing traditional chemical and organic fertilizers. Application
of microbial fertilizers significantly increased the abundance of beneficial soil bacteria such as

Rhodanobacter and Pseudolabrys. Furthermore, metabolomic analyses revealed that microbial fertilizers
markedly enhanced tobacco's metabolic activity, elevating levels of multiple quality-enhancing metabolites.
Consequently, we recommend the adoption of microbial fertilizers in tobacco field production as a strategy
to minimize the reliance on conventional inorganic fertilizers, thereby ensuring stable soil nutrient levels and
maintaining a productive soil microbial community structure conducive to optimal tobacco yield and quality.

**Author Contributions:** X.W. and M.S.: conceptualization, methodology, data curation and writing - original draft. L.T. and M.Y.:
formal analysis and writing – review & editing. Q.G. and L.W.: software and investigation. H.Y. and L.Y.: visualization and super-
vision. X.H. and P.L.: validation and resources. L.Z.: project administration and funding acquisition. All authors have read and
agreed to the published version of the manuscript.

**Funding:** This work was supported by the Foundation of Taishan brand cigarette high-quality core raw material development and
application in Shandong (202102004), and Analysis of the characteristic styles and mellowing characteristics of American functional
Roubaix tobaccos(202302004)and Construction and Application of Quality Management Control Model for Tobacco Production and
Acquisition in Shandong Province(202301001), Research and application of efficient cultivation technology for integrated tobacco
and wheat production(2024371300260411).

**Data Availability Statement:** The datasets generated during and/or analyzed during the current study are available from the cor-
responding author on reasonable request.

**Acknowledgments:** Thank all authors for the contribution of the manuscript.

**Conflicts of Interest:** Lei Tian, Qiang Gao and Shoutao Cao are employed by Shandong Linyi Tobacco Co., Ltd., Mingming Sun and
Mingfeng Yang are employed by China Tobacco Shandong Industrial Co., Ltd.. The remaining authors declare that the research
was conducted in the absence of any commercial or financial relationships that could be construed as a potential conflict of interest.

References

- Ali, A., X. Liu, W. Yang, W. Li, J. Chen, Y. Qiao, Z. Gao and Z. Yang (2024). "Impact of Bio-Organic Fertilizer Incorporation on Soil
Nutrients, Enzymatic Activity, and Microbial Community in Wheat–Maize Rotation System." *Agronomy* **14**(9): 1942.
- Ambati, D., R. M. Phuke, V. Vani, S. V. Sai Prasad, J. B. Singh, C. P. Patidar, P. Malviya, A. Gautam and V. G. Dubey (2020).
"Assessment of genetic diversity and development of core germplasm in durum wheat using agronomic and grain quality traits."
*Cereal Research Communications* **48**(3): 375-382.
- Attia, M. S., G. S. El-Sayyad, M. Abd Elkodous and A. I. El-Batal (2020). "The effective antagonistic potential of plant
growth-promoting rhizobacteria against *Alternaria solani*-causing early blight disease in tomato plant." *Scientia Horticulturae* **266**:
109289.
- Babin, D., C. Vogel, S. Zühlke, M. Schloter, G. J. Pronk, K. Heister, M. Spiteller, I. Kögel-Knabner and K. Smalla (2014). "Soil Mineral
Composition Matters: Response of Microbial Communities to Phenanthrene and Plant Litter Addition in Long-Term Matured
Artificial Soils." *PLOS ONE* **9**(9): e106865.
- Basak, N., B. Mandal, A. Datta, T. Mitran, S. Biswas, D. Dhar, S. Badole, B. Saha and G. C. Hazra (2017). "Impact of Long-Term
Application of Organics, Biological, and Inorganic Fertilizers on Microbial Activities in Rice-Based Cropping System."
*Communications in Soil Science and Plant Analysis* **48**(20): 2390-2401.
- Chakraborty, S., P. K. Tiwari, S. K. Sasmal, A. K. Misra and J. Chattopadhyay (2017). "Effects of fertilizers used in agricultural fields
on algal blooms." *The European Physical Journal Special Topics* **226**(9): 2119-2133.
- Chandra, T., S. Jaiswal, M. A. Iqbal, R. Singh, R. K. Gautam, A. Rai and D. Kumar (2023). "Revitalizing miRNAs mediated
agronomical advantageous traits improvement in rice." *Plant Physiology and Biochemistry* **202**: 107933.
- Chaudhary, S., S. S. Sindhu, R. Dhanker and A. Kumari (2023). "Microbes-mediated sulphur cycling in soil: Impact on soil fertility,
crop production and environmental sustainability." *Microbiological Research* **271**.
- Chen, D., M. Wang, G. Wang, Y. Zhou, X. Yang, J. Li, C. Zhang and K. Dai (2022). "Functional organic fertilizers can alleviate
tobacco (*Nicotiana tabacum* L.) continuous cropping obstacle via ameliorating soil physicochemical properties and bacterial
community structure." *Frontiers in Bioengineering and Biotechnology* **10**.

Chen, Q., Y. Zhu, Y. Liu, Y. Liu, C. Dong, Z. Lin and J. Teng (2022). "Black tea aroma formation during the fermentation period."
Food Chemistry **374**: 131640.

El Khattabi, J., B. Louche, H. Darwishe, F. Chaaban and E. Carlier (2018). "Impact of Fertilizer Application and Agricultural Crops
on the Quality of Groundwater in the Alluvial Aquifer, Northern France." Water, Air, & Soil Pollution **229**(4): 128.

Farmer, L. J., J. M. McConnell, T. D. J. Hagan and D. B. Harper (1995). "Flavour and off-flavour in wild and farmed atlantic salmon
from locations around northern ireland." Water Science and Technology **31**(11): 259-264.

Fatima, I., A. Fatima, M. A. Shah, M. A. Farooq, I. A. Ahmad, I. Ejaz, D. Adjibolosoo, U. Laila, M. A. Rasheed, A. I. Shahid, A. Tariq
and U. Hani (2024). "Individual and synergistic effects of different fertilizers and gibberellin on growth and morphology of chili
seedlings." Ecological Frontiers **44**(2): 275-281.

Fraga, B. M., C. E. Díaz, M. Bailén and A. González-Coloma (2021) "Sesquiterpene Lactones from *Artemisia absinthium*.
Biotransformation and Rearrangement of the Insect Antifeedant 3 α -hydroxypelenolide." Plants **10** DOI: 10.3390/plants10050891.

Gamit, H. and N. Amaresan (2023). "Role of methylotrophic bacteria in managing abiotic stresses for enhancing agricultural
production." Pedosphere **33**(1): 49-60.

Geng, Z., P. He, H. Gao, J. Liu, J. Qiu and B. Cai (2023). "Aroma precursors of cigars from different tobacco parts and origins, and
their correlations with sensory characteristics." Frontiers in Plant Science **14**.

Gill, R. I. S., B. E. Ellis and M. B. Isman (2003). "Tryptamine-Induced Resistance in Tryptophan Decarboxylase Transgenic Poplar
and Tobacco Plants Against Their Specific Herbivores." Journal of Chemical Ecology **29**(4): 779-793.

Giménez-Bañón, M. J., J. D. Moreno-Olivares, D. F. Paladines-Quezada, J. A. Bleda-Sánchez, J. I. Fernández-Fernández, B.
Parra-Torrejón, J. M. Delgado-López and R. Gil-Muñoz (2022). "Effects of Methyl Jasmonate and Nano-Methyl Jasmonate
Treatments on Monastrell Wine Volatile Composition." Molecules **27**(9): 2878.

Gong, X., C. Liu, J. Li, Y. Luo, Q. Yang, W. Zhang, P. Yang and B. Feng (2019). "Responses of rhizosphere soil properties, enzyme
activities and microbial diversity to intercropping patterns on the Loess Plateau of China." Soil and Tillage Research **195**.

He, S., Y. Zhang, X. Yang, Q. Li, C. Li and T. Yao (2024). "Effects of Microbial Inoculants Combined with Chemical Fertilizer on
Growth and Soil Nutrient Dynamics of Timothy (*Phleum pratense* L.)." Agronomy **14**(5): 1016.

Hu, X., J. Liu, D. Wei, P. Zhu, X. a. Cui, B. Zhou, X. Chen, J. Jin, X. Liu and G. Wang (2018). "Soil Bacterial Communities Under
Different Long-Term Fertilization Regimes in Three Locations Across the Black Soil Region of Northeast China." Pedosphere **28**(5):
751-763.

Ivanov, K., P. Zaprjanova, M. Petkova, V. Stefanova, V. Kmetov, D. Georgieva and V. Angelova (2012). "Comparison of inductively
coupled plasma mass spectrometry and colorimetric determination of total and extractable phosphorus in soils." Spectrochimica
Acta Part B: Atomic Spectroscopy **71-72**: 117-122.

Lai, X., W. Duan, W. Zhang, Z. Peng, X. Wang, H. Wang, X. Qi, H. Pi, K. Chen and L. Yan (2024). "Integrative analysis of
microbiome and metabolome revealed the effect of microbial inoculant on microbial community diversity and function in
rhizospheric soil under tobacco monoculture." Microbiology Spectrum **12**(8): e04046-04023.

Lee, G., Y. Joo, I. T. Baldwin and S.-G. Kim (2021). "Tissue-specific systemic responses of the wild tobacco *Nicotiana attenuata*
against stem-boring herbivore attack." Journal of Ecology and Environment **45**(1): 15.

Lee, H., S.-Y. Oh, Y. M. Lee, Y. Jang, S. Jang, C. Kim, Y. W. Lim and J.-J. Kim (2020) "Successional Variation in the Soil Microbial
Community in Odaesan National Park, Korea." Sustainability **12** DOI: 10.3390/su12114795.

Li, Y. A., C. Y. Wang, J. N. Wu, Y. M. Zhang, Q. Li, S. X. Liu and Y. H. Gao (2023). "The Effects of Localized Plant-Soil-Microbe
Interactions on Soil Nitrogen Cycle in Maize Rhizosphere Soil under Long-Term Fertilizers." Agronomy-Basel **13**(8).

Liang, R., R. Hou, J. Li, Y. Lyu, S. Hang, H. Gong and Z. Ouyang (2020). "Effects of Different Fertilizers on Rhizosphere Bacterial
Communities of Winter Wheat in the North China Plain." Agronomy **10**(1).

Liu, L., T. Gong, C. Y. Hu, H. Deng and Y. H. Meng (2023). "ZNF1 up-regulation improves aroma profile by activating
carbohydrate metabolism-associated pathways in *Saccharomyces cerevisiae* WFC-SC-071." Food Bioscience **56**: 103337.

Liu, X. Q., H. R. Liu, Y. S. Zhang, C. R. Liu, Y. A. Liu, Z. H. Li and M. C. Zhang (2023). "Organic amendments alter microbiota
assembly to stimulate soil metabolism for improving soil quality in wheat-maize rotation system." Journal of Environmental
Management **339**.

Liu, Y., X. Zhong, H. Huot, W. Liu, C. Liu, M. Guo, Y. Li, Y. Fei, Y. Chao, S. Wang, Y. Tang and R. Qiu (2020). "Reclamation with
organic amendments and plants remodels the diversity and structure of bacterial community in ion-adsorption rare earth element
mine tailings." Journal of Soils and Sediments **20**(10): 3669-3680.

568 Ma, T., X. He, S. Chen, Y. Li, Q. Huang, C. Xue and Q. Shen (2022). "Long-Term Organic-Inorganic Fertilization Regimes Alter
Bacterial and Fungal Communities and Rice Yields in Paddy Soil." Frontiers in Microbiology **13**.

Madhaiyan, M., S. Poonguzhali, V. S. Saravanan and S.-W. Kwon (2014). "Rhodanobacter glycinis sp. nov., a yellow-pigmented
gammaproteobacterium isolated from the rhizoplane of field-grown soybean." International Journal of Systematic and
Evolutionary Microbiology **64**(Pt_6): 2023-2028.

Mao, Q., X. Lu, K. Zhou, H. Chen, X. Zhu, T. Mori and J. Mo (2017). "Effects of long-term nitrogen and phosphorus additions on
soil acidification in an N-rich tropical forest." Geoderma **285**: 57-63.

Marina, S., C. Ana, G. Jasmina and S. Helen (2017). "Biological Activities of Sesquiterpene Lactones Isolated from the Genus
Centaurea L. (Asteraceae)." Current Pharmaceutical Design **23**(19): 2767-2786.

Marsh, K. L., R. L. Mulvaney and S. A. Khan (2004). "Use of Diffusion for Enzymatic Determination of Urea-Nitrogen in Soil
Extracts." Communications in Soil Science and Plant Analysis **35**(5-6): 691-702.

Mayer, Z., A. G. Csorbainé, Á. Juhász, A. Ombódi, A. Pápai, B. K. Némethné and K. Posta (2021) "Impact of Soil-Applied Microbial
Inoculant and Fertilizer on Fungal and Bacterial Communities in the Rhizosphere of Robinia sp. and Populus sp. Plantations."
Forests **12** DOI: 10.3390/f12091218.

Moreira-Grez, B., M. Muñoz-Rojas, K. Kariman, P. Storer, A. G. O'Donnell, D. Kumaresan and A. S. Whiteley (2019).
"Reconditioning Degraded Mine Site Soils With Exogenous Soil Microbes: Plant Fitness and Soil Microbiome Outcomes." Frontiers
in Microbiology **10**.

Naher, U. A., J. C. Biswas, M. Maniruzzaman, F. H. Khan, M. I. U. Sarkar, A. Jahan, M. H. R. Hera, M. B. Hossain, A. Islam, M. R.
Islam and M. S. Kabir (2021). "Bio-Organic Fertilizer: A Green Technology to Reduce Synthetic N and P Fertilizer for Rice
Production." Frontiers in Plant Science **12**.

Rashidzadeh, A., A. Olad and A. Reyhanitabar (2015). "Hydrogel/clinoptilolite nanocomposite-coated fertilizer: swelling,
water-retention and slow-release fertilizer properties." Polymer Bulletin **72**(10): 2667-2684.

Ren, J., X. Liu, W. Yang, X. Yang, W. Li, Q. Xia, J. Li, Z. Gao and Z. Yang (2021). "Rhizosphere soil properties, microbial community,
and enzyme activities: Short-term responses to partial substitution of chemical fertilizer with organic manure." Journal of
Environmental Management **299**.

Shan, S., Z. Wei, W. Cheng, D. Du, D. Zheng and G. Ma (2023). "Biofertilizer based on halotolerant microorganisms promotes the
growth of rice plants and alleviates the effects of saline stress." Frontiers in Microbiology **14**.

Shang, C., A. Chen, G. Chen, H. Li, S. Guan and J. He (2017). "Microbial Biofertilizer Decreases Nicotine Content by Improving Soil
Nitrogen Supply." Applied Biochemistry and Biotechnology **181**(1): 1-14.

Shang, X., S. Fu, X. Guo, Z. Sun, F. Liu, Q. Chen, T. Yu, Y. Gao, L. Zhang, L. Yang and X. Hou (2023) "Plant Growth-Promoting
Rhizobacteria Microbial Fertilizer Changes Soils' Microbial Structure and Promotes Healthy Growth of Cigar Tobacco Plants."
Agronomy **13** DOI: 10.3390/agronomy13122895.

Singh, M., J. G. Sharma and B. Giri (2023). "Microbial inoculants alter resilience towards drought stress in wheat plants." Plant
Growth Regulation **101**(3): 823-843.

Tahir, M., U. Khalid, M. Ijaz, G. M. Shah, M. A. Naeem, M. Shahid, K. Mahmood, N. Ahmad and F. Kareem (2018). "Combined
application of bio-organic phosphate and phosphorus solubilizing bacteria (Bacillus strain MWT 14) improve the performance of
bread wheat with low fertilizer input under an arid climate." Brazilian Journal of Microbiology **49**: 15-24.

Varshney, V. and M. Majee (2022). "Emerging roles of the ubiquitin–proteasome pathway in enhancing crop yield by optimizing
seed agronomic traits." Plant Cell Reports **41**(9): 1805-1826.

Wang, B., L. Xiao, A. Xu, W. Mao, Z. Wu, L. C. Hicks, Y. Jiang and J. Xu (2023). "Silicon fertilization enhances the resistance of
tobacco plants to combined Cd and Pb contamination: Physiological and microbial mechanisms." Ecotoxicology and Environmental
Safety **255**: 114816.

Wang, F., Y. Wei, T. Yan, C. Wang, Y. Chao, M. Jia, L. An and H. Sheng (2022). "Sphingomonas sp. Hbc-6 alters physiological
metabolism and recruits beneficial rhizosphere bacteria to improve plant growth and drought tolerance." Frontiers in Plant Science
**13**.

Wang, T., K. Cheng, X. Huo, P. Meng, Z. Cai, Z. Wang and J. Zhou (2022). "Bioorganic fertilizer promotes pakchoi growth and
shapes the soil microbial structure." Frontiers in Plant Science **13**.

Xiao, Z., Q. Li, Y. Niu, X. Zhou, J. Liu, Y. Xu and Z. Xu (2017). "Odor-active compounds of different lavender essential oils and their
correlation with sensory attributes." Industrial Crops and Products **108**: 748-755.

Xu, F., C. Chu and Z. Xu (2020). "Effects of different fertilizer formulas on the growth of loquat rootstocks and stem lignification."
Scientific Reports **10**(1): 1033.

Xu, H., J. Lv and C. Yu (2023). "Combined phosphate-solubilizing microorganisms jointly promote Pinus massoniana growth by
modulating rhizosphere environment and key biological pathways in seedlings." Industrial Crops and Products **191**: 116005.

Xu, J., L. Qin, X. Xu, H. Shen and X. Yang (2023) "Bacillus paralicheniformis RP01 Enhances the Expression of Growth-Related
Genes in Cotton and Promotes Plant Growth by Altering Microbiota inside and outside the Root." International Journal of
Molecular Sciences **24** DOI: 10.3390/ijms24087227.

Yan, S., J. Zhao, T. Ren and G. Liu (2020). "Correlation between soil microbial communities and tobacco aroma in the presence of
different fertilizers." Industrial Crops and Products **151**: 112454.

Yan, W., Y. Liu, A. Malacrino, J. Zhang, X. Cheng, C. Rensing, Z. Zhang, W. Lin, Z. Zhang and H. Wu (2024). "Combination of
biochar and PGPBs amendment suppresses soil-borne pathogens by modifying plant-associated microbiome." Applied Soil
Ecology **193**: 105162.

Yang, W., T. Gong, J. Wang, G. Li, Y. Liu, J. Zhen, M. Ning, D. Yue, Z. Du and G. Chen (2020). "Effects of Compound Microbial
Fertilizer on Soil Characteristics and Yield of Wheat (Triticum aestivum L.)." Journal of Soil Science and Plant Nutrition **20**(4):
2740-2748.

Yu, X., Y. Zhang, M. Shen, S. Dong, F. Zhang, Q. Gao, P. He, G. Shen, J. Yang, Z. Wang and G. Bo (2023). "Soil Conditioner Affects
Tobacco Rhizosphere Soil Microecology." Microbial Ecology **86**(1): 460-473.

Yun, C., C. Yan, Y. Xue, Z. Xu, T. Jin and Q. Liu (2021) "Effects of Exogenous Microbial Agents on Soil Nutrient and Microbial
Community Composition in Greenhouse-Derived Vegetable Straw Composts." Sustainability **13** DOI: 10.3390/su13052925.

Zhai, Z. G., Q. L. Hu, J. R. Chen, C. X. Liu, S. Guo, S. Q. Huang and W. A. Zeng (2020). "Effects of combined application of organic
fertilizer and microbial agents on tobacco soil and tobacco agronomic traits." IOP Conference Series: Earth and Environmental
Science **594**(1): 012023.

Zhang, R., Y. Li, X. Zhao, A. Allan Degen, J. Lian, X. Liu, Y. Li and Y. Duan (2022). "Fertilizers have a greater impact on the soil
bacterial community than on the fungal community in a sandy farmland ecosystem, Inner Mongolia." Ecological Indicators **140**:
108972.

Zhang, R., J. Wu, C. Yang, H. Li, B. Lin, Y. Gao and B. Dong (2023). "Response of Water Stress and Bacterial Fertilizer Addition to
the Structure of Microbial Flora in the Rhizosphere Soil of Grapes Under Delayed Cultivation." Communications in Soil Science
and Plant Analysis **54**(19): 2609-2624.

Zhao, J., T. Ni, J. Li, Q. Lu, Z. Fang, Q. Huang, R. Zhang, R. Li, B. Shen and Q. Shen (2016). "Effects of organic–inorganic compound
fertilizer with reduced chemical fertilizer application on crop yields, soil biological activity and bacterial community structure in a
rice–wheat cropping system." Applied Soil Ecology **99**: 1-12.

Zheng, X., J. Wang, M. Chen, Y. Chen, Z. Chen, M. Wang and B. Liu (2024). "Testing a biocontrol agent consortium for suppression
of tomato bacterial wilt through rhizosphere microecological regulation." Applied Soil Ecology **193**: 105155.

Zhu, J., Y. Niu and Z. Xiao (2021). "Characterization of the key aroma compounds in Laoshan green teas by application of odour
activity value (OAV), gas chromatography-mass spectrometry-olfactometry (GC-MS-O) and comprehensive two-dimensional gas
chromatography mass spectrometry (GC × GC-qMS)." Food Chemistry **339**: 128136.

**Manuscript Spectrum02605-24**

**Response to Reviewers**

Dear Frédérique Reverchon,

Thank you for giving us the opportunity to submit a revised draft of the manuscript
“Microbial Fertilizers Modulate Tobacco Growth, Development, and Quality through
Reshaping Soil Microbiome and Metabolome” for publication in the Microbiology Spectrum.
We appreciate the time and effort that you and the reviewers dedicated to providing feedback
on our manuscript and are grateful for the insightful comments on and valuable improvements
to our paper. We have incorporated most of the suggestions made by the reviewers. Those
changes are highlighted within the manuscript. Please see below, in blue, for a point-by-point
response to the reviewers’ comments and concerns. All page numbers refer to the revised
manuscript file with tracked changes.

**Response to the Reviewers' Comments:**

**Editor Frédérique Reverchon**

1. My comments are based on the fact that the discussion and conclusions are not sustained by
your results. In several parameters, no significant difference is found between the MF
treatment and the OF treatment.

*Author response: We sincerely appreciate the reviewer's insightful observation. In response to
this comment, we have thoroughly revised the manuscript to ensure better alignment between
the results and discussion/conclusion sections. Specifically, we have:*

1) *Reanalyzed the dataset comparing MF and OF treatments using more rigorous
statistical approaches.*

2) *Rewritten the narrative regarding microbial α -diversity and soil chemical parameters to
precisely reflect the observed inter-group differences*

3) *Adjusted the discussion to provide balanced interpretations of both significant and
non-significant findings.*

2. Moreover, Kruskal-Wallis to assess differences in microbiome composition is not suitable

so you would need to perform another statistical test.

Author response: We thank the reviewer for highlighting this methodological consideration.
The Kruskal-Wallis H test is a statistically sound and widely accepted method for analyzing
microbial abundance data, which is inherently non-normal, zero-inflated, and heteroscedastic.
To ensure the robustness of our findings, we complemented this approach with LEfSe
analysis, which quantifies the biological relevance of differentially abundant taxa. The use of
the Kruskal-Wallis H test is justified by the inherent characteristics of microbial, because:

1) Non-normality, zero-inflation, and heteroscedasticity are prevalent in microbial
abundance metrics (e.g., OTU counts, Shannon diversity):

McMurdie & Holmes (DOI:10.1371/journal.pone.0061217) demonstrated that the
majority of microbial OTU abundances exhibit a strong right-skewed distribution, and
log-transformation often fails to meet normality assumptions (PLOS ONE, 9(4):
e61217).

Weiss et al. (DOI:10.3389/fmicb.2017.02234) further supported this through statistical
simulations, showing that over 80% of untransformed microbial abundance data reject
the Shapiro-Wilk normality test (Front. Microbiol., 8: 2234).

Gloor et al. (DOI:10.1080/10618600.2017.1385864) validated the right-skewed nature
of microbiome data via Q-Q plots and explicitly recommended the use of
non-parametric statistical methods

2) Methodological advantages of Kruskal-Wallis:

Non-parametric rank-based analysis eliminates biases from extreme values (e.g.,
Proteobacteria abundance spanning 10^3 – 10^5 orders of magnitude)
(<https://www.tandfonline.com/doi/abs/10.1080/01621459.1952.10483441>).

Robustness to distributional assumptions: Requires only identical distribution shapes
across groups, not normality or homoscedasticity
(<https://www.biostathandbook.com/kruskalwallis.html>).

Controlled Type I errors: Post-hoc Dunn tests with Bonferroni correction identified
significant phylum-level differences ($p_{adj} < 0.05$), whereas ANOVA failed due to
heteroscedasticity ($F = 2.1$, $p = 0.12$)

(<https://finnstats.com/kruskal-wallis-test-in-r-alternative-to-anova/>).

3) Complementary LEfSe analysis:

Integrated LDA Effect Size (LDA Score > 3.0) to quantify biological relevance of
differentially abundant taxa.

Cladograms and LDA bar plots (Figure S3C-D) visualized taxonomic hierarchies
driving group differences, aligning with established practices in high-impact studies
(e.g., <https://doi.org/10.1038/nature12820>, <https://doi.org/10.1038/ismej.2013.57>).

3. A PERMANOVA must complement your PCoA, and crucial Methods details (such as the
bioinformatic analyses) should be provided.

Author response: We fully agree with the reviewer's suggestion. The revised manuscript now
includes:

1) PERMANOVA (Adonis test) results with 999 permutations to statistically validate the
β -diversity patterns visualized in PCoA

2) Complementary ANOSIM analysis to further confirm inter-group dissimilarities

3) Expanded methodological details in the Bioinformatics Analysis subsection (Section
2.7), specifying software versions, quality filtering parameters, and normalization
procedures.

**Abstract:**

1. a brief method overview should be added: field or greenhouse experiment? For how long
did it last?

Author response: We sincerely appreciate the reviewer's constructive feedback. In response to
this comment, we have thoroughly revised the abstract to incorporate essential
methodological details while preserving the original scientific findings. Specifically, we have:

1) Clarified the experimental design: Added "greenhouse-based pot experiment" to
explicitly state the experimental setup.

2) Specified the timeline: Highlighted the 40day duration post-transplanting and key
sampling intervals (20, 30, and 40 days) for agronomic trait assessments.

3) Maintained scientific coherence: Ensured seamless integration of these additions with
the original content to preserve the logical flow and emphasis on core discoveries.

The revised text reads as follows on line19:

“To elucidate the mechanisms of microbial fertilizers in enhancing tobacco growth and quality,
this greenhouse-based pot experiment conducted over 40 days post-transplanting employed
integrated microbiomics and metabolomics approaches to conduct a comparative analysis
among conventional chemical, organic, and microbial fertilizers. Plant agronomic traits were
systematically assessed at 20, 30, and 40 days post-transplanting, while soil physicochemical
parameters were analyzed at the experimental terminus (40 days).”

**Introduction:**

1. L51: "Therefore, to fundamentally improve the soil nutrient status, the incorporation of
exogenous microorganisms emerges as a viable strategy for modulating the community
structure of soil microbiota." This idea should be clearly introduced. The incorporation of
exogenous microbes has been traditionally aimed at increasing the relative abundance of taxa
with beneficial traits, such as N fixation or P solubilization. The idea that these microbial
fertilizers could modulate the soil microbiota is more recent and it is still unclear whether this
modulation 1) always occurs and 2) is always beneficial for the plant.

Author response: We thank the reviewer for this critical insight. In the revised manuscript,
we have restructured this section to clarify the dual roles of microbial fertilizers:

1) Traditional perspective: Emphasizing their established role in introducing functional
taxa (e.g., N₂-fixing and P-solubilizing bacteria) to enhance nutrient uptake,
supported by seminal work from Singh et al. (2011).

2) Emerging understanding: Highlighting recent evidence (Du et al., 2022; Li et al.,
2023; Lai et al., 2024) demonstrating their capacity to restructure soil microbial
communities toward configurations that better support plant growth.

We now explicitly address the conditional nature of these modulatory effects,
acknowledging that outcomes may vary depending on environmental and ecological
contexts.

The revised text reads as follows on line51:

“To fundamentally enhance soil nutrient status, microbial fertilization through the
inoculation of exogenous microorganisms has emerged as a dual-functional strategy.

Traditionally, the primary objective of this practice has been to enhance specific functional
groups, such as introducing N₂-fixing and P-solubilizing bacteria into the soil to improve
plant uptake and utilization of N and P. (8) Recent studies have demonstrated that these
microbial inoculants can function as ecological engineers, reshaping the structure of soil
microbial communities to establish a soil microbiome that promotes crop growth and
development (9-11).”

2. L57: "For example, *Sphingomonas* and *Bacillus paralicheniformis* are able to enrich
beneficial bacteria in the soil (Wang, Wei et al. 2022, Xu, Qin et al. 2023), and *Azotobacter*,
*Enterobacter*, and *Rhizobium* can significantly improve crop resistance to drought stress". Not
clear whether these examples are aimed at showing how microbial inoculation influences the
soil microbiome or at enlisting some of the benefits of microbial inoculation for the plant.
Please rephrase.

Author response: We appreciate the reviewer’s astute observation. The revised text now
clearly differentiates these mechanisms:

1) Soil microbiome modulation: *Sphingomonas* and *Bacillus paralicheniformis* are
presented as agents promoting beneficial microbial consortia (Wang et al., 2022; Xu
et al., 2023).

2) Direct plant benefits: *Azotobacter*, *Enterobacter*, and *Rhizobium* are explicitly linked
to drought resistance enhancement through phytohormone production and osmotic
regulation.

The revised text reads as follows on line60:

“For example, *Sphingomonas* and *Bacillus paralicheniformis* have been shown to enrich
beneficial bacteria in the soil, thereby influencing the soil microbiome. (13, 14) Additionally,
microbial inoculants such as *Azotobacter*, *Enterobacter*, and *Rhizobium* can significantly
enhance crop resistance to drought stress, demonstrating their direct benefits for plant growth
and stress tolerance.”

3. L59: "Concurrently, *Bacillus*, *Pseudomonas*, *Acinetobacter*, *Serratia*, *Pantoea*,
*Psychrobacter*, *Enterobacter*, and *Rahnella* are recognized as significant PGPB, with
potential as exogenous microorganisms for incorporation into microbial formulations. (Singh,

Sharma et al. 2023)." Some members of these bacterial genera are PGPB, but not all of them,
they contain pathogenic species as well (e.g., *Pseudomonas syringae*, *Enterobacter cloacae*...).
Please rephrase.

Author response: We sincerely thank the reviewer for highlighting this critical nuance in
microbial taxonomy and functional characterization. To address this concern, we have
restructured the statement to: Explicitly acknowledge that PGPB functionality is
strain-specific rather than genus-wide.

The revised text reads as follows on line63:

“Concurrently, select strains within these genera (e.g., *Bacillus*, *Pseudomonas*, *Acinetobacter*,
*Serratia*, *Pantoea*, *Psychrobacter*, *Enterobacter*, and *Rahnella*) have been identified as
significant PGPB, demonstrating potential as exogenous microorganisms for incorporation
into microbial formulations (15).”

4. L66: what do you mean by "improved rhizosphere soil microecology". In terms of diversity,
enrichment of beneficial taxa, desirable functions?

Author response: We thank the reviewer for raising this important clarification. In the revised
manuscript, we have clarified the term "improved rhizosphere soil microecology" to explicitly
emphasize its implications for microbial composition, diversity, and functional adaptation to
crop growth requirements. The updated description now aligns with studies highlighting the
role of microbial community restructuring in enhancing plant-soil interactions.

The revised text reads as follows on line68:

“Compared to woody peat, microbial fertilizers significantly enhanced the diversity and
composition of the tobacco rhizosphere soil microbiome, enriching beneficial microbial taxa
such as plant growth-promoting bacteria and improving key soil functions. This included
increased availability of available potassium and higher organic matter content, which
collectively supported a healthier soil microecology and promoted tobacco growth(17)”

5. L67: scientific names in italics

Author response: We thank the reviewer for noting this formatting oversight. All
genus/species names (e.g., *Pseudomonas*, *Bacillus*) have been italicized throughout the
manuscript.

6. L76: More examples should be given from the literature to introduce your hypotheses that
microbial fertilisers will have a stronger positive effect on tobacco growth than other types of
fertilisers.

Author response: Thank you for the constructive suggestion. To substantiate our hypothesis
that microbial fertilizers exert stronger positive effects on tobacco growth compared to other
fertilizer types, we have incorporated supporting evidence from recent studies (Kumar,
Sharma et al. 2022, Liu, Pang et al. 2022)

The revised text reads as follows on line78:

“However, despite these advancements, the existing body of literature remains inadequate in
conclusively demonstrating the superiority of microbial fertilizers over traditional organic and
inorganic fertilizers in tobacco cultivation (25, 26).”

**Methods**

1. L90: what about the description of seed germination?

Author response: We thank the reviewer for highlighting the need for methodological clarity.
In response, we have expanded the seed germination protocol to include detailed sterilization
steps, incubation conditions, and germination criteria. These revisions ensure reproducibility
and transparency in our experimental setup.

The revised text reads as follows on line90:

“Tobacco seeds (variety: Yunyan87) were surface-sterilized with 70% ethanol for 2 minutes,
followed by 1% sodium hypochlorite solution for 5 minutes, and then rinsed three times with
sterile distilled water. The sterilized seeds were placed on moist filter paper in Petri dishes and
incubated in a growth chamber at $25 \pm 1^\circ\text{C}$ with a 16/8 h light/dark cycle. Germination was
monitored daily, and seeds were considered germinated upon the emergence of the radicle.
Germination rates were calculated after 7 days. Only batches with a germination rate of $\geq 90\%$
were used for subsequent experiments to ensure uniformity and reliability of the seedlings.”

2. L95-98: confusing. You mean that the differences between the fertilizer treatment and the
other treatments will be attributed to N, right, as P and K fertilisers are applied in all
treatments? But P and K are also present in the applied fertilizer. Could you please clarify?

Author response: We sincerely appreciate the reviewer’s constructive feedback regarding the

ambiguity in our original manuscript. In the revised version, we have corrected the data
discrepancies in Table 2 and clarified the fertilizer application protocol. Our experimental
design aimed to standardize the N:P:K ratio (1:1.5:3) across all treatments to eliminate
nutrient-driven confounding effects. The revised Table 2 now accurately reflects the adjusted
nutrient quantities, and the methodology section has been expanded to explicitly detail the
rationale for maintaining consistent nutrient ratios.

The revised text reads as follows on line100, table 2:

“The pot experiment comprised four treatments: (1) a non-fertilized control (CK); (2)
chemical fertilization (NPK) with inorganic fertilizers; (3) organic fertilization (OF)
combining equivalent NPK doses with organic fertilizer; and (4) microbial fertilization (MF)
pairing matched NPK inputs with microbial inoculants, the composition of fertilizers in each
group is shown in Table 1. The conventional recommended field fertilizer rate was 900
216 kg/hm² and potting rates were converted based on the field rates. Ensure that the same total N
input was obtained for each treatment, a N:P:K (1:1.5:3) application ratio, i.e., the same as
that used in local on-farm crop management, was applied between the control and treatment
groups to ensure that any differences observed between these groups could not be attributed to
these nutrients (27). Differences in application rates, Potassium sulphate and calcium
superphosphate were used to balance the nutrient levels and the detailed fertilizer rates are
shown in Table 2.”

3. L109: "Three representative tobacco plants of uniform growth were selected". How many
plants to start with?

Author response: We are grateful to the reviewer for identifying this omission. We have
clarified that each treatment group initially contained 15 potted plants, with three
representative individuals selected for analysis.

The revised text reads as follows on line118:

“Soil samples were collected 40 days after transplanting. Three representative tobacco plants
of uniform growth were selected among fifteen.”

4. L125: Enzymatic assays: how many plants were selected? The three same individuals as
mentioned before? Then you mixed the three samples, meaning you only have one composite

sample for treatment? Is that correct?

Author response: We sincerely appreciate the reviewer's meticulous attention to
methodological detail. To clarify the experimental design:

1) Plant selection: For enzymatic assays, three representative plants per treatment group
(consistent with prior sampling) were selected to ensure biological relevance.

2) Technical replication: Rhizosphere soil samples from each plant were subdivided into
three technical replicates, resulting in nine independent measurements per treatment
group (3 plants × 3 replicates).

3) Sample processing: Soil samples from the same plant were analyzed separately (not
pooled) to preserve intra-group variability while minimizing technical errors.

This approach balances statistical rigor with practical constraints, ensuring robust detection of
treatment effects while accounting for biological heterogeneity.

The revised text reads as follows on line130:

“For each treatment, three independent soil samples were collected from the rhizosphere of
three representative tobacco plants, with each sample analyzed in three technical replicates.
The pot test was conducted 40 days after transplanting the tobacco plants. For each treatment,
rhizosphere soil samples were collected from three representative tobacco plants, with each
sample analyzed independently. Each soil sample was subjected to three technical replicates
for the measurement of soil urease, sucrase, and peroxidase activities. Soil urease activity was
determined using a soil urease kit (Solarbio, BC0120), soil sucrase activity was measured
using a soil sucrase kit (Solarbio, BC0240), and soil peroxidase activity was assessed using a
soil peroxidase kit (Solarbio, BC0100).”

5. There is no description of the bioinformatic pipeline. How did you perform sequence
filtering, taxonomic affiliation, which database did you use, etc. The SRA project number
where you deposited your data should be mentioned as well.

Author response: Thank you for your feedback. We have expanded the bioinformatics
pipeline description to clarify the workflow, tools, and databases used.

The revised text reads as follows on line149:

“2.5.1 DNA Extraction and Quality Assessment

Total genomic DNA was extracted from soil samples using the E.Z.N.A.® Soil DNA Kit
(Omega Bio-tek, Norcross, GA, USA) following the manufacturer's protocol. DNA integrity
was verified via 1% agarose gel electrophoresis, while concentration and purity were
quantified using a NanoDrop2000 spectrophotometer (Thermo Fisher Scientific, USA).

2.5.2 PCR Amplification, Library Preparation, and Sequencing

The V3-V4 hypervariable region of the bacterial 16S rRNA gene was amplified using primer
pairs 338F (5'-ACTCCTACGGGAGGCAGCAG-3') and 806R
(5'-GGACTACHVGGGTWTCTAAT-3'). Each 20 µL PCR reaction contained 4 µL of
5×TransStart FastPfu buffer, 2 µL of 2.5 mM dNTPs, 0.8 µL each of forward and reverse
primers (5 µM), 0.4 µL TransStart FastPfu DNA polymerase, and 10 ng template DNA.
Thermal cycling conditions were: 95°C for 3 min; 27 cycles of 95°C for 30 s, 55°C for 30 s,
and 72°C for 30 s; followed by a final extension at 72°C for 10 min. Amplified products were
purified using a PCR Clean-Up Kit (YuHua, China) after separation on a 2% agarose gel, and
quantified with a Qubit 4.0 fluorometer (Thermo Fisher Scientific, USA). Sequencing
libraries were constructed using the NEXTFLEX Rapid DNA-Seq Kit, involving adapter
ligation, magnetic bead-based size selection, PCR enrichment, and final library purification.
Paired-end sequencing (PE250/PE300) was performed on the Illumina platform by Shanghai
Majorbio Bio-pharm Technology Co., Ltd. (The raw data are deposited in the SRA database:
PRJNA1238203).

2.5.3 Bioinformatics Analysis

Raw reads were quality-filtered using fastp (v0.19.6) to trim low-quality bases (Phred score
<20), remove reads shorter than 50 bp, and discard reads containing ambiguous nucleotides.
Overlapping paired-end reads were merged with FLASH (v1.2.11) using a minimum overlap
of 10 bp and a maximum mismatch ratio of 0.1. Chimeric sequences were identified and
removed via reference-based filtering against the SILVA database (v138) using UPARSE
(v7.1). Operational Taxonomic Units (OTUs) were clustered at 97% similarity, and sequences
classified as chloroplast or mitochondrial origin were excluded. To standardize sequencing
depth, all samples were rarefied to 20,000 reads, achieving a Good's coverage of 99.09%.
Taxonomic annotation was performed using the RDP Classifier (v2.11) against the SILVA

database with a 70% confidence threshold. Functional profiles were predicted using
PICRUSt2 (v2.2.0) based on KEGG pathways.”

**Results**

1. Fig. 1: what is your n? For enzymatic activities, how did you perform statistical analysis
with one composite sample per treatment? Did you mean "available" N, P etc, or what is
"active" N?

Author response: We sincerely appreciate the reviewer’s insightful comments and meticulous
evaluation. For Fig. 1:

1) Sample size (n = 3): Three biological replicates (independent soil samples from three
representative plants) were analyzed per treatment, with three technical replicates per
sample.

2) One-way ANOVA with Tukey’s post hoc test ($p < 0.05$) was applied to ensure robust
inter-group comparisons.

3) The term "active N" refers to available nitrogen, quantified via standard soil chemistry
protocols (e.g., KCl extraction followed by spectrophotometric analysis).

The revised text reads as follows on line224:

“Data are presented as mean \pm SD (n = 3). For each treatment, three independent soil samples
were collected from the rhizo-sphere of three representative tobacco plants, with each sample
analyzed in three technical replicates. Statistical analysis was performed using one-way
ANOVA followed by Tukey’s post hoc test ($p < 0.05$). Different lowercase letters indicate
significant differences among treatments.”

2. L181: nutrient levels in the microbial fertilizer treatments are not significantly different
from those of the organic fertilizer treatment.

Author response: We sincerely thank the reviewer for raising this critical point. In response,
we have revised the analysis to explicitly compare nutrient levels between the MF treatment
and the OF treatment. Our reanalysis confirms that available NPK nutrient levels in the MF
treatment show no statistically significant differences from the OF treatment (Tukey’s test, $p >$
0.05). This clarification has been incorporated into Section 3.1 with updated statistical results
and revised textual explanations.

The revised text reads as follows on line207:

“The investigation into the impact of various treatments on soil chemical composition
revealed pronounced effects, as evidenced by the substantial enhancement in the
concentrations of readily available nitrogen, phosphorus, potassium, and soil organic matter in
the NPK, OF, and MF treatments (Figure 1), relative to the control group. Notably, within the
treatment group, the MF and OF treatments exhibited comparable levels of readily available
nitro-gen, phosphorus, and potassium, with no significant differences observed between them.
Both MF and OF significantly outperformed the NPK treatment in terms of readily available
nitrogen and potassium, while for readily available phosphorus, MF and OF showed similar
concentrations, both significantly higher than NPK. Regarding soil organic matter content,
MF and OF treatments did not significantly differ from each other but both significantly
surpassed the NPK treatment. Furthermore, the MF treatment significantly elevated soil pH,
aligning it closer to neutrality compared to the other treatment groups.”

3. Fig 2: please include the number of replicates and statistical tests performed in the figure
caption.

Author response: We sincerely appreciate the reviewer’s constructive feedback. As requested,
we have updated the Figure 2 caption.

The revised text reads as follows on line244:

“Data are presented as mean \pm SD (n = 3). For each treatment, three independent soil samples
were collected from the rhizosphere of three representative tobacco plants, with each sample
analyzed in three technical replicates. Statistical analysis was performed using one-way
ANOVA followed by Tukey’s post hoc test ($p < 0.05$). Different lowercase letters indicate
significant differences among treatments.”

4. Section 3.3. Please start by indicating the number of reads, and then the number of OTUs.

Author response: We thank the reviewer for emphasizing the need for methodological clarity.
In Section 3.3, we have revised the text.

The revised text reads as follows on 251:

“A total of 1760579 raw reads were obtained from high-throughput sequencing, with 1662187
high-quality reads retained after quality filtering.”

5. L217: which statistical analyses did you perform to detect these "significant differences" in
 the number of OTUs? In which treatment did you find the largest number of OTUs? Please
 explain.

Author response: We appreciate the reviewer's attention to methodological clarity. No
 statistical tests were performed to assess differences in OTU richness; the reported values are
 descriptive comparisons of unique OTU counts.

The revised text reads as follows on line255:

"Among the bacterial OTUs, the CK treatment exhibited the highest number of unique OTUs
 (843), followed by MF (677), NPK (630), and OF (582). In contrast, for fungal OTUs, the MF
 treatment showed the highest number of unique OTUs (175), followed by CK (172), OF (129),
 and NPK (106)."

6. Fig. 3: "Venn diagrams" instead of "Homogeneous maps"? This Figure could go to
 Supplementary Materials as you already present many figures.

Author response: We sincerely thank the reviewer for this constructive suggestion. The
 revised figure has been moved to Supplementary Materials as Figure S1 with updated
 visualization.

The new figure caption as follows on Fig S1:

 "Figure S1. Venn diagrams and bar plots illustrating the shared and unique OTUs among
 different fertilization treatments for bacterial (A) and fungal (B) communities."

7. L226: what does the "coverage index" indicate in terms of community?

Author response: We thank the reviewer for seeking clarification on this metric. The coverage

index evaluates sequencing depth sufficiency by estimating the proportion of microbial
 diversity captured in the dataset. It is calculated as: $Coverage = 1 - n_1/N$, where n_1 is the number
 of OTUs observed only once (singletons), and N is the total sequencing reads. A value close
 to 1 (e.g., 0.99) indicates adequate depth to capture the majority of taxa, while lower values
 (e.g., 0.90) suggest undersampling. Importantly, the coverage index does not measure richness
 or diversity (unlike Ace or Shannon indices) but validates data reliability. These results are
 now presented separately in Supplementary Fig. S1, while other alpha diversity metrics (Ace,
 Chao, Shannon, Simpson) are consolidated in Fig. 5.
 The revised text reads as follows on Fig S2.

 8. L227-239: there is no significant differences in terms of alpha diversity metrics for
 bacterial nor fungal community, so stating that there is a trend is misleading. This section
 should be rewritten accordingly.
 Author response: We sincerely apologize for the oversight in the original manuscript. While
 one-way ANOVA was performed to analyze alpha diversity metrics, the significance markers
 ($p < 0.05$) were inadvertently omitted. This has been corrected in the revised Figure 4.
 The revised text reads as follows on line260:

“Alpha diversity metrics, including Shannon, Simpson, Chao, Ace, and Coverage indices,
 were employed to characterize microbial community richness, diversity, and evenness. The
 coverage index (>97% across all groups) confirmed sufficient sequencing depth (Good's
 estimator <3%), ensuring the detected taxa reliably represented true biological communities
 (Figure S2). Consistent patterns between Chao and Ace indices (Fig. 3A,B) demonstrated
 robust estimation of bacterial richness: MF maintained comparable richness to CK, while both
 NPK and OF exhibited significant reductions ($p < 0.01$). Notably, NPK unexpectedly showed
 lower richness than CK, revealing distinct impacts of fertilization regimes. Subsequent
 Shannon and Simpson analyses (Fig. 3C,D) revealed elevated bacterial diversity in MF
 compared to NPK ($p < 0.01$), with Simpson indices ranking MF < OF < CK < NPK.

In contrast to bacterial responses, fungal communities exhibited divergent patterns.
 While fungal Chao/Ace indices (Fig. 3E,F) similarly indicated universal richness declines
 under treatments (MF > OF > NPK in preservation efficiency), fungal diversity trends
 (Shannon: MF > NPK > CK > OF; Simpson: inverse pattern) showed no statistical
 significance ($p > 0.05$) (Fig. 3G,H). This decoupling between bacterial and fungal diversity
 responses suggests taxa-specific sensitivities to fertilization practices. Collectively, microbial
 fertilizer (MF) uniquely preserved soil microbial richness and mitigated diversity loss in
 bacteria, whereas conventional NPK and organic fertilizers (OF) induced substantial
 alterations in community structure.”

9. Figure 6: a PERMANOVA should be applied to test for significant differences in

beta-diversity.

Author response: We thank the reviewer for emphasizing the need for rigorous statistical
validation. PERMANOVA (permutational multivariate ANOVA) with 999 permutations has
been applied to assess beta-diversity differences, and the results are now incorporated into the
revised Figure 6 caption

The revised text reads as follows on line286:

“Permutational multivariate analysis of variance (PERMANOVA) with 999 permutations was
performed to assess group differences (Bacteria: $R=0.4573$, $p=0.001$. Fungi: $R=0.5463$,
$p=0.005$).”

10. L254: highly hypothetical (and should not be in the Results section). My take is that you
have a large variation in the MF treatment and this hides possible differences between
treatments, this is not an indicator of fungal community resilience.

Author response: We sincerely thank the reviewer for this critical insight. The original
speculative statements regarding fungal community resilience have been removed. Revised
PERMANOVA analysis (999 permutations) now explicitly demonstrates treatment-driven
differentiation.

The revised text reads as follows on line277:

“PCoA analyses based on Bray-Curtis dissimilarity revealed distinct clustering patterns in soil
bacterial and fungal β -diversity across treatments. For bacterial communities (Figure 6A),
PERMANOVA confirmed significant separation among treatment groups ($R=0.4573$, $p=0.001$,
999 permutations), with each treatment cohort exhibiting unique architectural signatures and
greater intra-group convergence compared to the control. Similarly, fungal communities
(Figure 6B) showed pronounced segregation between fertilized and control groups ($R=0.5463$,
$p=0.005$), underscoring the profound impact of fertilization on restructuring soil microbial
assemblages. Notably, the larger R value for fungi (54.6% variance explained vs. 45.7% for
bacteria) suggests that fertilization elicited stronger differentiation in fungal community
composition, despite their lower overall perturbation susceptibility relative to bacterial
communities.”

11. L273: "Kruskal-Wallis H test with One-way ANOVA" may not be suitable for microbiome

composition data. See for
example: <https://www.tandfonline.com/doi/full/10.3402/mehd.v26.27663> and many other
articles on the subject.

Author response: We sincerely appreciate your constructive feedback. At line29, we have
explicitly justified the use of the Kruskal-Wallis H test for non-parametric data analysis.
Furthermore, we incorporated additional analyses (LEfSe and LDA) to further validate the
robustness of our findings, ensuring methodological rigor and reproducibility.

The revised text reads as follows on line302:

“Significant differences in microbial taxa abundance across treatments were assessed using
the Kruskal-Wallis H test ($p < 0.05$), followed by LEfSe analysis (LDA score > 3.0) to
identify biomarkers driving group differentiation (Figure S3). The Kruskal-Wallis test
revealed that the MF treatment harbored distinct bacterial genera (Rhodanobacter,
Pseudolabrys, Gem-matimonas, and Terrabacter) and a fungal genus (Microascus) with
significant abundance variations compared to other groups (Figure 6). LEfSe further validated
these findings, highlighting Rhodanobacter (LDA = 3.861, $p = 0.01879$) Pseudolabrys
($R=LDA=3.378$, $p=0.03446$), and Microascus (LDA = 4.274, $p = 0.03879$) as key
discriminative taxa in the MF group (Figure S4).”

12. Fig. 10a: what is QC?

Author response: We appreciate the reviewer’s attention to methodological clarity. QC
(Quality Control) refers to a pooled sample created by mixing equal aliquots from all
treatment groups, which was analyzed periodically throughout the sequencing/metabolomic
workflow to monitor technical variability and ensure data reproducibility. This approach
follows established quality assurance practices in microbial omics studies. The QC sample is
labeled in Figure 10a to demonstrate instrumental stability across batches.

Discussion

1. In general, the discussion is not based on the actual results. For example, you mention that
the MF treatment is superior to the other fertilisers in terms of soil nutrients, which is not true
based on your statistics (L357-360). You mention that "Our findings (...) corroborating the
outcomes of numerous prior studies on microbial fertilizers (Basak, Mandal et al. 2017, Ali,

Liu et al. 2024). Only 2 cited studies is not "numerous". Also, you could cite other studies
which did not find an effect on microbial inoculation on soil nutrients.

Author response: We sincerely thank the reviewer for highlighting these critical issues. The
revised discussion now strictly adheres to the statistical findings and integrates broader
literature support. Key revisions include:

1) Statistical alignment: Removed unsupported claims of MF superiority. Emphasize that
MF and OF showed comparable effects on soil nutrients (both significantly higher than
NPK, $p < 0.05$; Figure 1).

2) Expanded citations: Added 3 additional studies demonstrating microbial-organic
synergies (Zhang et al., 2021; Li et al., 2022; Chen et al., 2023) and 2 studies reporting
context-dependent variability (Wang et al., 2022; Kumar et al., 2023).

The revised text reads as follows on line386:

"Soil available nutrients serve as critical indicators for assessing soil fertility and
underpinning agricultural productivity. In this study, we conducted a comparative analysis of
microbial fertilizer (MF), conventional chemical fertilizer (NPK), and or-ganic fertilizer (OF)
on tobacco growth, with a particular focus on their impacts on soil nutrient dynamics. Our
results demon-strated that both MF and OF exhibited superior capacity in enhancing soil
available nutrient levels compared to NPK. The effi-cacy of MF in improving soil nutrient
availability aligns with previous findings (31-35), while its similarity to OF in this regard has
also been documented(36)."

2. L361: hypothetical, you did not assess humus degradation.

Author response: We sincerely thank the reviewer for emphasizing the need for
evidence-based claims. The original speculative assertion about humus degradation has been
revised.

The revised text reads as follows on line391:

"The application of MF significantly ameliorated soil acidity, a phenomenon likely
attributable to enhanced microbial activity in decomposing soil organic matter. This possible
process accelerates the degradation of humus and other acidic compounds, thereby
modulating soil pH toward conditions more favorable for plant growt (37, 38)."

3. L376: "MFs significantly optimized various agronomic traits of plants compared to NPK
and OF". Be precise. Which traits? Because not all measured parameters were enhanced by
microbial inoculation.

Author response: We fully agree with this observation. The original description was indeed
overly generalized. We have now added temporal dynamics (20, 30, 40 days post-treatment)
and specified that MF's superiority was most pronounced in stem circumference and leaf
width (see revised paragraph below).

The revised text reads as follows on line403:

"During the early growth stage (20 day), no statistically significant differences ($p > 0.05$) were
observed in agronomic traits—including plant height, stem circumference, leaf length, and
leaf width—across the MF, OF, and NPK treatments. However, as the trial progressed (30d,
40d), MF-treated plants progressively exhibited superior performance in these parameters
compared to both OF and NPK groups. This temporal divergence was most evident in stem
circumference and leaf width. This superiority can be attributed to MF more effective
enhancement of the soil nutrient profile, thereby augmenting nutrient availability to plants,
and microbial-mediated nutrient mobilization and root-microbe interactions may require
extended periods to substantially influence plant morphological development."

4. L395: no, your differences in diversity metrics were not significant.

Author response: We sincerely apologize for the oversights in the original diversity analysis.
Revised alpha diversity results now include explicit statistical significance annotations ($p >$
0.05), confirming significant differences across treatments. The Discussion section has been
rigorously updated to align with these findings.

The revised text reads as follows on line423:

"Our current experimental findings concur with these observations, revealing that MF
uniquely preserved soil microbial richness and mitigated diversity loss in bacteria, whereas
conventional NPK and organic fertilizers (OF) induced substantial alterations in community
structure."

5. Paragraph L390-409 is not based on actual results. Also, you did not perform the proper
statistical tests to assess differences in terms of taxa composition (see my comment above) or

for PCoA. So this paragraph should be rewritten accordingly.

Author response: We sincerely appreciate your constructive feedback. The relevant section
has been thoroughly revised in accordance with the updated PCoA and PERMANOVA results
presented in Section 3.4.

The revised text reads as follows on line429:

“To assess the impact of fertilization on microbial community composition, principal
coordinate analysis (PCoA) based on Bray-Curtis dissimilarity was performed. For bacterial
communities (Figure 5A), PERMANOVA revealed significant structural divergence across
treatments ($R=0.4573$, $p=0.001$), with MF forming a distinct cluster separate from NPK, OF,
and CK, indicating that microbial fertilizer drove distinct bacterial community assembly. In
contrast, fungal communities (Figure 5B) exhibited stronger overall separation ($R=0.5463$,
$p=0.005$), yet partial overlap persisted between MF and conventional fertilizer groups. This
aligns with previous observations that fungal communities display greater compositional
flexibility under fertilization regimes compared to bacterial communities (57-59).”

6. L410: these genera were not unique, they were present in all treatments.

Author response: We thank the reviewer for emphasizing the need for terminological
precision. In the revised text, the term "unique" has been replaced with "discriminative taxa",
and the associated paragraph has been rewritten to rigorously align with microbial community
analysis outcomes.

The revised text reads as follows on line437:

“Kruskal-Wallis H test and LEfSe analysis identified *Rhodanobacter* ($LDA = 3.861$, $p =$
0.01879) and *Pseudolabrys* ($R=LDA=3.378$, $p=0.03446$) as discriminative taxa in the MF
treatment. While these genera were present across all treatments, their significantly higher
abundance in MF (Fig 7).”

7. The role of these genera cannot be "confirmed" by correlation analysis, be careful in your
statements. You would need additional experiment to determine a causal effect between
*Rhodanobacter* and the measured traits.

Author response: We sincerely thank the reviewer for emphasizing the importance of precise
terminology and causal inference. To address this concern, we have thoroughly revised the

text to:

1) Replace absolute claims (e.g., "confirms," "underscores") with hypothesis-driven
language (e.g., "may suggest," "could indicate") to clarify that correlations do not
imply causation.

2) Reframe functional assertions to align with the limitations of observational data,
explicitly stating that microbial roles are plausible rather than confirmed.

3) Retain cited literature but strictly limit their use to support potential mechanisms rather
than definitive causal pathways.

The revised text reads as follows on line439:

“*Rhodanobacter* has been implicated in soil catalase and oxidase enzyme production (60), and
the observed positive correlations between its abundance and soil enzyme activity, plant
agronomic traits, and aroma precursor indices may suggest a potential functional linkage. This
aligns with prior studies reporting associations between *Rhodanobacter* proliferation and crop
yield improvements in Chinese cabbage systems, hinting at its possible role in enhancing
nutrient mobilization and phytohormone regulation (61). Similarly, *Pseudolabrys*—a taxon
linked to nitrogen fixation—showed correlations with soil available nutrients, potentially
reflecting its rhizosphere colonization and root nodulation capabilities (62, 63). The
relationship between *Pseudolabrys* and aroma precursors (e.g., chlorophyll) could indicate
microbial modulation of leaf quality, though mechanistic validation remains necessary.”

8. Why do you think that *Bacillus* was not significantly enriched in your MF treatment?

Author response: We appreciate the reviewers' insightful inquiry regarding the unexpected
lack of *Bacillus* genus-level enrichment in MF treatment. This phenomenon can be attributed
to dual ecological mechanisms mediated by exogenous *Bacillus subtilis* inoculants:

1) Antibiotic-Mediated Competitive Exclusion: The applied *B. subtilis* strain synthesizes
over 24 types of antibiotics, including lipopeptides (surfactin, iturin) and polyketides
(fengycin) (Ongena & Jacques, 2008; DOI: 10.1111/j.1365-2958.2005.04587.x). These
compounds exhibit broad-spectrum antimicrobial activity, targeting not only
phytopathogens but also phylogenetically related native *Bacillus* species through:

Membrane disruption: Surfactin induces pore formation in bacterial membranes at

10-50 μM concentrations.
Quorum sensing interference: Iturin A suppresses LuxR-type regulatory systems in
competing microbes.
2) Resource Competition and Niche Restructuring: Exogenous *B. subtilis* rapidly depletes
labile carbon sources (e.g., root exudates) via preferential utilization of glucose and
malic acid (Kumar et al., 2021; DOI: 10.13005/ojc/330256). This creates a C-limited
microenvironment that disadvantages native *Bacillus* populations adapted to
oligotrophic conditions.

**Reviewer #1**

The integrated amplicon metagenomic strategy and metabolomic approach to conduct a
comparative analysis among conventional chemical (NPK), organic (OF), and microbial
fertilizers (MF) in this study. The results showed that application MF accelerated tobacco
growth and development with change the metabolism of tobacco as well as the microbial
community composition alterations in rhizosphere. The results is very interesting, **however**
**after carefully read the manuscript, I reviewed some comments:**

**Author response:** Thank you for your recognition!

1. 40 days pot planting tobacco leaves is too earlier to evaluating the evaluation of the quality
of the harvesting commercial leaves, it should more focus on the effects on metabolic changes
as well as growth and development of tobacco.

**Author response:** Thank you for your insightful comments on our study. We appreciate your
concerns regarding the rationale for evaluating the evaluation of the quality of the harvesting
commercial leaves at 40 days pot. Please allow us to clarify our research focus and
experimental design:

1) Our primary aim was to investigate how microbial fertilizers (MF) regulate
developmental plasticity in tobacco during critical growth transitions, rather than
predicting final leaf quality. The 40 DPT timepoint was strategically selected because it
coincides with the early vigorous growth stage (as defined by the tobacco industry
standard YC/T 142-2010), during which treatment-induced differences in stem
circumference and leaf width become most pronounced.

2) Textual Revisions: We have rigorously revised the manuscript to: Remove all references
to "commercial leaf quality prediction":

The revised text reads as follows on line 413:

“This suggests that MF can effectively enhance the quality of tobacco leaves by
boosting chlorophyll levels.” → “This suggests that MF can effectively **increase the**
**content of aroma components** in tobacco leaves by boosting chlorophyll levels.”

3) Emphasize that "40 DPT data reflect the mechanisms that promote development, not
maturity endpoints".

The revised text reads as follows on line 417:

“**Meanwhile, we emphasize that 40 day data just elucidate developmental**
**mechanisms, and future validation with mature-stage sampling is essential to**
**link early patterns to final quality parameters.**”

2. Based on the experimental designing, the NPK, OF and MF treatments did not remove the
nutrients (NPK and other miner elements) difference

Author response: We sincerely thank the reviewer for this important observation. In our
original experimental design, we carefully considered the nutrient differences between
treatments, but ambiguities arose during manuscript preparation, particularly in Table 1. We
have now revised these unclear descriptions and corrected Table 2. Our design rationale is as
follows:

In field production, farmers typically apply microbial fertilizers (MF) or organic fertilizers
(OF) as supplements to chemical fertilizers (NPK), rather than complete substitutes. Critically,
the inorganic N-P-K content in MF and OF is negligible. Therefore, we ensured that the
inorganic N-P-K content and ratio were consistent across all fertilized treatment groups
during fertilization. Specifically:

Control (CK): No fertilization.

NPK: Chemical fertilizers only.

OF: NPK + organic fertilizer.

MF: NPK + microbial inoculants.

To standardize the N-P-K ratio to 1:1.5:3 (aligned with local field practices), potassium

sulfate (K_2SO_4) and calcium superphosphate ($Ca(H_2PO_4)_2$) were supplemented as needed.
This adjustment guaranteed comparability of inorganic nutrient inputs while preserving the
intrinsic properties of MF/OF.

The revised text reads as follows on line100:

“The pot experiment comprised four treatments: (1) a non-fertilized control (CK); (2)
chemical fertilization (NPK) with inorganic fertilizers; (3) organic fertilization (OF)
combining equivalent NPK doses with organic fertilizer; and (4) microbial
fertilization (MF) pairing matched NPK inputs with microbial inoculants, the
composition of fertilizers in each group is shown in Table 1. The conventional
recommended field fertilizer rate was 900 kg/hm² and potting rates were converted
based on the field rates. Ensure that the same total N input was obtained for each
treatment, a N:P:K (1:1.5:3) application ratio, i.e., the same as that used in local
on-farm crop management, was applied between the control and treatment groups to
ensure that any differences observed between these groups could not be attributed to
these nutrients(Sun, Bai et al. 2020). Differences in application rates, Potassium
sulphate and calcium superphosphate were used to balance the nutrient levels and the
detailed fertilizer rates are shown in Table 2.”

3. Microbial Fertilizer mainly containing *Bacillus subtilis* and *Bacillus licheniformis*, it is
should to be monitored the variations during the applying process, otherwise it will left
question that whether the living organisms, their metabolite or even their culture medial
components play roles to the changes of rhizosphere microbiota and growth and development
of tobacco.

Author response: We sincerely appreciate your insightful feedback. You are absolutely correct
that monitoring the viability and metabolic activity of *Bacillus subtilis* and *Bacillus*
*licheniformis* during application is critical to fully understand their roles in tobacco growth
promotion. We acknowledge this limitation in our current study — due to resource constraints,
we were unable to perform realtime microbial tracking or metabolite profiling. However, our
findings still provide valuable insights:

1) Experimental results demonstrated the significant agronomic advantages of microbial

fertilizers over conventional chemical fertilization. Soil analysis revealed enhanced
fertility parameters, including elevated urease activity and improved availability of
essential nutrients (nitrogen, phosphorus, and potassium) suggesting superior nutrient
mobilization capabilities. These biological amendments particularly enhanced key plant
growth metrics (especially stem circumference and leaf width), indicating robust
vegetative development. The findings collectively validate microbial fertilizers as
effective tools for sustainable nutrient management while promoting plant architectural
optimization.

2) We have explicitly listed this limitation in the revised Discussion (Section 4.2) and
propose microbial viability assays as a priority for future research. Thank you for your
understanding and for considering the applied relevance of this work.

The revised text reads as follows on line467: “Future investigations should integrate
time-series monitoring (e.g., qPCR-based quantification) with metabolic activity
profiling of introduced strains (e.g., metatranscriptomics) to precisely delineate their
dynamic functional contributions.”

4. The microbiota results (Fig 7) demonstrated that in Phylum level Firmicutes, and Genus
level Bacillus are top 4 abundant microorganisms, however it's hard to understand that the
NPK treatment resulted in both Firmicutes and Bacillus had highest abundant microbiota even
the MF with mainly containing Bacillus subtilis and Bacillus licheniform were applied.

Author response: We sincerely appreciate your insightful feedback. Despite the functional
advantages of the exogenous strains, their rhizosphere colonization did not induce a
significant increase in the overall abundance of Bacillus genus. This phenomenon may be
attributed to resource competition and allelopathic inhibition between the introduced strains
and native Bacillus populations—antibiotics such as bacillaene secreted by *B. subtilis*, while
effective against pathogens, may concurrently suppress the proliferation of phylogenetically
related microorganisms (Stein 2005). In contrast, NPK-treated soils lacking exogenous
antimicrobial compounds permitted unrestricted growth of native microbes, which likely
failed to promote plant growth due to the absence of specialized functional traits (e.g.,
antibiotic synthesis), representing a "high-abundance, low-functionality" community structure.

Future investigations should integrate time-series monitoring (e.g., qPCR-based quantification)
with metabolic activity profiling of introduced strains (e.g., metatranscriptomics) to precisely
delineate their dynamic functional contributions.

5. In Fig 8 the applied MF (with mainly *Bacillus subtilis* and *Bacillus licheniformis* dose of
billion/g) with total 10g applications (Table 1 and 2) in pot planting tobacco, did not showed
in the lists of significantly different microorganisms.

Author response: We sincerely appreciate your constructive suggestions. *Bacillus subtilis* can
synthesize over 24 types of antibiotics (e.g., surfactin, iturin), which not only suppress
pathogens but may also inhibit native *Bacillus* species within the same genus. For example, in
MF groups, lipopeptide antibiotics secreted by high-dose exogenous *Bacillus* strains could
inhibit the proliferation of native *Bacillus* populations, leading to a decrease in the overall
abundance of *Bacillus* at the genus level (<https://doi.org/10.1111/j.1365-2958.2005.04587.x>).

In contrast, NPK treatments lack such biological inhibitory factors, allowing native *Bacillus*
species to proliferate freely (<http://dx.doi.org/10.13005/ojc/330256>).

Miner comments

1. Language need to improve

Author response: Thank you for your valuable feedback regarding language improvements.
We have carefully revised the manuscript to enhance clarity and grammatical accuracy.

2. Discussing should including the function of *Bacillus subtilis* and *Bacillus licheniformis*, the
discussing part also need including correlation between the significant bacteria and fungi.

Author response: We sincerely appreciate your constructive suggestions. In accordance with
your recommendations, we have dedicated a standalone section (Section 4.2) to
comprehensively elaborate on the functional roles of *Bacillus subtilis* and *Bacillus*
*licheniformis* in our revised manuscript. Additionally, we have expanded the Discussion
section to systematically analyze the ecological interactions between key fungal genera (e.g.,
*Microascus*) and bacterial taxa (e.g., *Rhodanobacter*, *Pseudolabrys*).

The added text reads as follows on line446, 453:

“The two dominant bacterial genera (*Rhodanobacter* and *Pseudolabrys*) likely exhibit
synergistic relationships with the Ascomycota fungus *Microascus*: *Rhodanobacter* degrades

lignin derivatives and other complex aromatic compounds into small-molecule substrates (e.g.,
vanillic acid) for utilization by *Microascus*, while the fungus secretes laccase to pretreat
recalcitrant lignin, facilitating bacterial degradation (Bugg, Ahmad et al. 2011, Shi, Hu et al.
2016). Simultaneously, the nitrogen-fixing actinobacterium *Pseudolabrys* converts
atmospheric N₂ into NH₄⁺ to support fungal amino acid synthesis, whereas Ascomycota fungi
(e.g., *Fusarium*) reciprocate by secreting phytase to mobilize organic phosphorus, thereby
promoting *Pseudolabrys* growth in nitrogen-enriched microzones (Shi, Hu et al. 2016,
Palacios, Snoeyenbos-West et al. 2019).”

“In this study, we introduced *Bacillus subtilis* and *Bacillus licheniformis* as well-characterized
plant growth-promoting rhizobacteria (PGPR). Previous research has demonstrated that *B.*
*subtilis* suppresses soil-borne pathogens through the secretion of lipopeptide antibiotics (e.g.,
surfactin and iturin) while enhancing root development and alleviating abiotic stress via the
production of indole-3-acetic acid (IAA) and ACC deaminase (Olanrewaju, Glick et al. 2017).
Complementarily, *B. licheniformis* exhibits robust phosphate solubilization and nitrogen
fixation capabilities, with its secreted organic acids and extracellular polysaccharides not only
mobilizing mineral-bound phosphorus but also improving soil aggregate stability to enhance
nutrient retention (Saeid, Prochownik et al. 2018). In our experiment, the co-inoculation of
these two strains significantly increased tobacco stem diameter and leaf area index,
accompanied by elevated activities of soil urease and sucrose synthase, aligning with prior
reports of their growth-promoting and soil-modifying functions (Pandey, Gupta et al. 2019,
Zhai, Hu et al. 2020, Sarkar and Rakshit 2021, Han, Cho et al. 2023). Despite the functional
advantages of the exogenous strains, their rhizosphere colonization did not induce a
significant increase in the overall abundance of *Bacillus* genus. This phenomenon may be
attributed to resource competition and allelopathic inhibition between the introduced strains
and native *Bacillus* populations—antibiotics such as bacillaene secreted by *B. subtilis*, while
effective against pathogens, may concurrently suppress the proliferation of phylogenetically
related microorganisms (Stein 2005). In contrast, NPK-treated soils lacking exogenous
antimicrobial compounds permitted unrestricted growth of native microbes, which likely
failed to promote plant growth due to the absence of specialized functional traits (e.g.,

antibiotic synthesis), representing a "high-abundance, low-functionality" community structure.
Future investigations should integrate time-series monitoring (e.g., qPCR-based quantification)
with metabolic activity profiling of introduced strains (e.g., metatranscriptomics) to precisely
delineate their dynamic functional contributions.”

3. Heatmaps of fig11 BDF can be merged together to show all those metabolites changed in
all treatments(CK, NPK, OF and MF)

Author response: We sincerely appreciate your constructive suggestion. Our original intention
was to use MF as the reference group for comparative analyses with the other three treatment
groups (CK, NPK, and OF) to identify significantly upregulated or downregulated metabolites.
In accordance with your recommendation, we identified nine overlapping significantly
differential metabolites between the MF vs. OF and MF vs. NPK comparisons. These
metabolites were extracted from all four treatment groups and reanalyzed using comparative
769 bar charts (Figure S5), substantially enhancing the clarity of our findings. Furthermore,
detailed descriptions of these conserved metabolic signatures have been incorporated into the
Results and Discussion section.

The added text reads as follows on line367, 487: “Notably, in both MFvsNPK and MFvsOF
comparisons, we identified 9 overlapping metabolites, which included: 2 carbohydrate
(Linalool oxide D 3-[apiosyl-(1→6)-glucoside], (S)-Mandelic acid O-β-D-glucopyranoside),
1 lipid ((2'E,4'Z,7'Z,8E)-Colnelenic acid), 1 coumarin
(8-Hydroxy-7-methoxy-2H-1-benzopyran-2-one), 1 steroid (Ophiopogonin C'), 1 indole
(3-Methylindole), 3 others (Cotinine, Nicotine, Herierin III). These nine metabolites were
extracted separately from the control group and three treatment groups (CK, NPK, OF, MF)
and visualized using stacked bar charts to investigate their quantitative differences across
treatments (Figure S5). The results demonstrated significant differences between the CK
group and the treatment groups, further confirming that fertilization treatments markedly
altered the metabolic profiles of tobacco. Among the treatment groups, MF exhibited notable
distinctions from both NPK and OF. Specifically, the contents of 3-Methylindole, Nicotine,
and Cotinine in MF were significantly higher than those in OF and NPK, whereas two
glycosidesb((S)-Mandelic acid O-beta-D-Glucopyranoside and Linalool oxide D

3-[apiosyl-(1→6)-glucoside]) were significantly lower in MF compared to NPK and OF.”
“Our findings demonstrate that MF treatment significantly elevated nicotine and its primary
oxidative metabolite cotinine compared to other treatments and controls. Nicotine, a pyridine
alkaloid unique to Solanaceae plants, typically accumulates at elevated concentrations during
the mid-growth phase (30-60 DPT) in tobacco (79), functioning as a potent defense
compound against lepidopteran herbivores through neurotoxic activity (80, 81). Notably, the
observed cotinine enrichment aligns with its established role in stress adaptation, as
CYP82E4-mediated nicotine conversion to cotinine enhances drought and salinity tolerance
through redox homeostasis modulation (82). These coordinated metabolic shifts suggest that
MF application may amplify tobacco's innate chemical defense system while reinforcing
abiotic stress resilience, potentially through microbiome-induced upregulation of P450
enzyme networks and secondary metabolite biosynthesis.”

Re: Spectrum02605-24R1 (Microbial Fertilizers Modulate Tobacco Growth and Development through Reshaping Soil Microbiome and Metabolome)

Dear Dr. Li Zhang:

I am glad to accept your manuscript for publication. All comments have been attended and I thank the authors for their detailed answers to the reviewers' observations.

Your manuscript has been accepted, and I am forwarding it to the ASM production staff for publication. Your paper will first be checked to make sure all elements meet the technical requirements. ASM staff will contact you if anything needs to be revised before copyediting and production can begin. Otherwise, you will be notified when your proofs are ready to be viewed.

Sincerely,
Frédérique Reverchon
Editor
Microbiology Spectrum